# A conserved role for centriolar satellites in translation of centrosomal and ciliary proteins

Claudia Pachinger[1,2] , Jeroen Dobbelaere[1] , Cornelia Rumpf-Kienzl[1] , Shiviya Raina[1,2] , Júlia Garcia-Baucells[1,2] , Marina Sarantseva[1] , Andrea Brauneis[1] , and Alexander Dammermann[1]

**Centriolar satellites are cytoplasmic particles found in the vicinity of centrosomes and cilia whose specific functional contribution has long been unclear. Here, we identify Combover as the *Drosophila* ortholog of the main scaffolding component of satellites, PCM1. Like PCM1, Combover localizes to cytoplasmic foci containing centrosomal proteins and its depletion or mutation results in centrosomal and ciliary phenotypes. Strikingly, however, the concentration of satellites near centrosomes and cilia is not a conserved feature, nor do Combover foci display directed movement. Proximity interaction analysis revealed not only centrosomal and ciliary proteins, but also RNA-binding proteins and proteins involved in quality control. Further work in *Drosophila* and vertebrate cells found satellites to be associated with centrosomal and ciliary mRNAs, as well as evidence for protein synthesis occurring directly at satellites. Given that PCM1 depletion does not affect overall protein levels, we propose that satellites instead promote the coordinate synthesis of centrosomal and ciliary proteins, thereby facilitating the formation of protein complexes.**

## Introduction

Centriolar satellites are nonmembranous particles concentrated in the vicinity of centrosomes and the ciliary base in vertebrate cells (Devi et al., 2021; Tischer et al., 2021). Work over the past two decades has linked numerous proteins to satellites, including proteins otherwise localized to centrioles, the pericentriolar material (PCM), or cilia (Gheiratmand et al., 2019; Gupta et al., 2015; Quarantotti et al., 2019). However, just a single protein, pericentriolar material 1 (PCM1), has been found to be essential for their formation, suggesting that it acts as their main scaffolding component (Dammermann and Merdes, 2002). PCM1 depletion not only eliminates satellites but also impairs the accumulation of centrosomal and ciliary proteins, thereby affecting processes from centriole assembly to centrosomal microtubule anchorage and ciliary trafficking (Dammermann and Merdes, 2002; Kim et al., 2004; Kodani et al., 2015).

While centriolar satellites have been linked to nearly every aspect of centrosome and cilium biology, their specific functional contribution has been difficult to define. Dynein motor–dependent movement of satellites to centrosomes (Kubo et al., 1999) has led to the idea of their functioning as transport modules delivering proteins for centrosome and cilium biogenesis (Dammermann and Merdes, 2002) (Fig. 1 A). However, while some PCM1 particles indeed undergo directed movement (Kubo et al., 1999), others display purely diffusive motion (Conkar et al., 2019). At the same time, satellite localization is highly variable, responding to a variety of environmental factors and cellular stresses (Baron et al., 1994; Villumsen et al., 2013), as well as cell cycle state, with marked disassembly in mitotic cells (Dammermann and Merdes, 2002). These changes may, however, be driven not only by motor-mediated movement, but also by liquid–liquid phase separation/dissolution of PCM1 condensates (Rai et al., 2018; Zhao et al., 2021).

In addition to protein trafficking, satellites have also been linked to autophagy and cellular proteostasis (Joachim et al., 2017; Prosser et al., 2022; Tang et al., 2013). Consistent with such a role, the phenotypes associated with their perturbation are highly pleiotropic, yet not as severe as those observed for many of their centrosomal and ciliary clients. For example, mouse mutants in *Plk4/Sak* (Hudson et al., 2001), *C2cd3* (Hoover et al., 2008), and *Pcnt* (Chen et al., 2014), all satellite clients, exhibit fully penetrant prenatal lethality. In contrast, *Pcm1* mutants are born at Mendelian frequencies, albeit with significant postnatal lethality and exhibiting a variety of defects indicative of ciliary dysfunction (Hall et al., 2023). Centrosome

[1]Max Perutz Labs, Vienna Biocenter (VBC), University of Vienna, Vienna, Austria;   [2]Vienna BioCenter PhD Program, Doctoral School of the University of Vienna and Medical University of Vienna, Vienna, Austria.

Correspondence to Alexander Dammermann: alex.dammermann@univie.ac.at

J. Dobbelaere's current affiliation is Institute of Science and Technology Austria (ISTA), Klosterneuburg, Austria.

Figure 1. **Satellite scaffolding component PCM1 is conserved beyond vertebrates. (A)** Schematic representation of the transport model of centriolar satellite (magenta) function as mediators of dynein-dependent recruitment of centrosomal and ciliary client proteins (blue) for centrosome/cilium biogenesis. **(B)** Reciprocal BLAST analysis reveals the presence of PCM1 orthologs across opisthokonts, correlating with the reported presence of centriole-organized centrosomes and cilia (Azimzadeh, 2014; Grell and Benwitz, 1981). **(C)** Multiple sequence alignment of conserved C terminus (part of pfam15717) of selected PCM1 orthologs. Note that *Drosophila* Combover (CMB) is highly divergent. **(D)** Overexpression but not RNAi-mediated depletion or mutation of *Cmb* results in a PCP phenotype, seen by misalignment of bristles in the fly notum. Tissue-specific RNAi was performed using the Pannier-GAL4 driver, with the PCP effector Fritz (Strutt and Warrington, 2008) as a positive control. See also Fig. S1.

composition is also largely unaffected following loss of PCM1 (Gheiratmand et al., 2019; Quarantotti et al., 2019). These results are inconsistent with an essential role of satellites in protein delivery, but perfectly in line with a potential role of satellites in refinement of the centrosomal/ciliary protein complement.

In summary, much remains to be discovered about how centriolar satellites become enriched around centrosomes and cilia, and the degree to which this localization contributes to their function. Here, we re-examine satellite function, building on the identification of a *Drosophila* ortholog of PCM1. Defining features conserved across metazoan evolution, we identify a role of satellites in facilitating the synthesis of centrosomal and ciliary proteins, a paradigm shift we suggest reconciles many of these disparate observations.

## Results

### The satellite scaffolding component PCM1 is conserved beyond vertebrates

While invertebrate model organisms including *Caenorhabditis elegans* and *Drosophila melanogaster* have made significant contributions to our understanding of centrosome and cilium biogenesis, work on satellites has so far focused exclusively on their role in vertebrates and in particular vertebrate cultured cells. Indeed, centriolar satellites have long been thought to be unique to vertebrates. Yet, a computational study using a refined sequence alignment algorithm optimized for coiled-coil proteins identified putative orthologs of PCM1 beyond vertebrates including in *C. elegans* and *Drosophila* (Kuhn et al., 2014), suggesting the phylogenetic distribution of PCM1 and satellites may

be wider than currently appreciated. Our reciprocal BLAST analysis confirmed this identification and revealed further orthologs across opisthokonts, including in choanoflagellates and ciliated fungi (Blastocladiales, chytrids), though not in higher fungi, in which centrioles and cilia have been secondarily lost (Fig. 1 B; see also Fig. S1 A). PCM1 orthologs could also not be identified in representatives of the parasitic phylum Nematomorpha (Cunha et al., 2023), which lack most ciliary genes. In contrast, PCM1 is conserved in the planarian *Schmidtea mediterranea*, which lacks centrosomes but retains centrioles and cilia (Azimzadeh et al., 2012). This pattern of inheritance and loss is consistent with a functional association with centriole-based structures conserved across >1,000 million years (Parfrey et al., 2011) of opisthokont evolution.

We initially came across the *Drosophila* ortholog of PCM1, Combover, or CMB (Fagan et al., 2014), in the course of establishing TurboID in flies as a proximity interactor of the centriolar structural component SAS-4 in S2 cells (Fig. S1 B). Primary sequence homology to the human protein, largely restricted in insects to the more conserved C terminus, is very low (Fig. 1 C), with identification of the *Drosophila* ortholog requiring the use of other less divergent insect species as intermediates (see Materials and methods). We were therefore interested to determine to what extent the functions ascribed to vertebrate PCM1 are conserved in *Drosophila*. Unlike the putative *C. elegans* ortholog, CMB is not entirely uncharacterized, having previously been linked to planar cell polarity (PCP), with the overexpression of the protein resulting in the formation of multiple hair cells in the wing (Fagan et al., 2014), a phenotype associated with perturbation of PCP effector genes such as Fritz (*Frtz*), Fuzzy (*Fy*), and Inturned (*In*) (Strutt and Warrington, 2008). We observed similar defects in the orientation of bristles upon the overexpression of CMB in the notum of the fly, another signature PCP effector phenotype (Fig. 1 D; and Fig. S1, C and D). However, neither acute depletion by tissue-specific RNAi nor complete loss of the protein in *Cmb* deletion mutants (Fagan et al., 2014) resulted in any observable PCP phenotype. The functional significance of this link therefore remains unclear.

### *Drosophila* PCM1/CMB is required for proper centrosome and cilium function

While *Cmb* mutants appear morphologically wild-type, behaviorally they are clearly not. The first noticeable defect is abnormal wing posture, indicative of impaired mechanosensation involving the chordotonal neurons, a ciliated cell type in the fly (Tuthill and Wilson, 2016) (Fig. 2 A and Fig. S2 A). Consistent with this, negative geotaxis in adult flies (Enjolras et al., 2012) was found to be strongly impaired (Fig. 2 B). As previously reported (Steinhauer et al., 2019), mutants were also fully male (though not female) infertile (Fig. 2 C and Fig. S2 B). Both negative geotaxis and male infertility could be reproduced by tissue-specific RNAi and rescued by introduction of a GFP transgene, confirming specificity of the mutant phenotype and functionality of the GFP transgene (Fig. 2, B and C). Neither phenotype is shared by PCP genes (Fig. 2, B and C), but both are highly reminiscent of what is observed following perturbation of centriolar or ciliary components (Dobbelaere et al., 2020, 2023). A

closer examination of cilium morphology in the chordotonal neurons responsible for mechanosensation in the animals' legs by DIC and immunofluorescence microscopy, as well as ultrastructural analysis, revealed only minor defects (Fig. S2, C–E). In contrast, spermatogenesis was more markedly affected, with a dissection of the testes of adult males revealing weakly motile sperm incapable of reaching the seminal vesicle (Fig. S2 F). Transmission electron microscopy showed a significant proportion of broken axonemes and missing axonemal microtubule doublets (Fig. 2 D). Cysts furthermore contained fewer than 64 flagella, indicative of a failure of axoneme extension or prior cell division defects (Fig. 2 E). Examining the process of spermatogenesis by staining isolated testes to visualize nuclear morphology and the actin cytoskeleton revealed no apparent defects at early stages of differentiation until the formation of actin cones during individualization, a process that was highly defective in *Cmb* mutants (Fig. S2 G; see also Steinhauer et al., 2019).

*Cmb* mutants, then, display clear defects in cilium biogenesis and function. However, ciliary defects cannot explain the fully penetrant parental-effect embryonic lethality of *Cmb* mutants. Thus, while homozygous mutant mothers are viable if uncoordinated, the offspring of these mothers and heterozygous males (homozygous males being unable to mate and fertilize oocytes) exhibit 50% lethality, with survivors invariably found to be among those 50% expected to carry a wild-type allele (Fig. 2 F). Examination of syncytial-stage embryos resulting from such a cross-expressing marker for centrioles (ASL, green) and chromosomes (Histone H2A, red) revealed a high degree of chromosome missegregation almost invariably followed by nuclear fallout (12 out of 15 missegregation events, Fig. 2, G and H), a quality control mechanism that internalizes faulty nuclei to the embryo interior to prevent them from contributing to further development (Sullivan et al., 1993). Centrioles and centrosomes are known to be dispensable for later stages of development and morphogenesis in the fly (Basto et al., 2006). Yet, acentrosomal somatic cell divisions are not entirely normal (Poulton et al., 2014) and centrosomes are essential in early embryogenesis, which in flies (Stevens et al., 2007; Varmark et al., 2007) as in vertebrates (Hudson et al., 2001; Yabe et al., 2007) depends on centrosomal microtubule–organizing center activity to sustain spindle assembly during their rapid mitotic cell division cycles. Time-lapse imaging of ASL revealed no evidence for centriole duplication defects in *Cmb* mutants, nor was mitotic PCM assembly significantly affected based on analysis of the PCM scaffolding component CNN (Fig. 2, H and I). However, this does not exclude potential defects in recruitment of PCM client proteins such as D-PLP, whose misexpression has been linked to similar phenotypes (Fang and Lerit, 2022). In summary, then, CMB is dispensable for PCP, but like PCM1 in vertebrates (Hall et al., 2023) is required for proper centrosome and cilium function and hence organismal viability and fertility.

### Satellites are conserved in *Drosophila* but do not concentrate near centrosomes or cilia

A defining characteristic of centriolar satellites in vertebrates is their concentration near centrosomes and the base of cilia, as

Figure 2. **Combover is required for ciliogenesis and proper cell division. (A)** Schematic of ciliated tissues used for phenotypic analysis. Primary cilia are found in sensory bristles and chordotonal neurons, while motile cilia/flagella are found in testes (Jana et al., 2016). **(B)** Climbing assay used to assess defects in mechanosensation. *Cmb* mutant flies are severely uncoordinated, a phenotype rescued by the expression of CMB-GFP. PCP flies (*Fy* RNAi) show no detectable phenotype. Error bars are the mean ± SD. N > 10 flies per condition. A Kruskal–Wallis test with Dunn's multiple comparisons test was performed; **P < 0.01, ****P < 0.0001. **(C)** Male fertility scored by crossing individual males with WT virgins and assessing the number of offspring. *Cmb* RNAi/mutant flies are fully male infertile. Error bars are the mean ± SD. N = 3 single males each crossed to four virgin females per condition. A Kruskal–Wallis test with Dunn's multiple comparisons test was performed; *P < 0.05, **P < 0.01, ***P < 0.001. **(D and E)** TEM analysis of sperm axonemes in control and *Cmb* mutant testes. Cross-sectional views reveal missing axonemal doublets and fragmented axonemes (D), as well as overall lower number of cilia (number per cyst <64, E). Error bars are the mean ± SD. N = 38 cysts (control), 38 cysts (*Cmb* mutant). Student's *t* test was used to assess statistical significance; *P < 0.05. **(F)** Embryonic viability test shows lethality in 50% of offspring of *Cmb* mutant females with heterozygous mutant males (viability could not be assessed for homozygous males due to their failure to mate and fertilize oocytes). Error bars are the mean ± SD. N = 2 virgin females crossed to one male per condition. Student's *t* test was used to

assess statistical significance; **P < 0.01. **(G)** Schematic of *Drosophila* syncytial-stage embryo showing synchronous nuclear divisions occurring close to the egg surface. **(H)** Live imaging of control and *Cmb* mutant early embryos (nuclear cycle 12) expressing the centriole marker ASL-GFP and H2A-RFP. Time shown in min:s. *Cmb* mutants show lagging chromosomes and increased frequency of nuclear fallout (orange dashed circle). Error bars are the mean ± SD. *N* = 5 control, 6 *Cmb* mutant embryos. Student's *t* test was used to assess statistical significance; *P < 0.05. **(I)** Live imaging of control and *Cmb* mutant early embryos (nuclear cycle 12) expressing the centrosome marker RFP-CNN. Time shown in min:s. *Cmb* mutants show defects in centrosome separation subsequent to nuclear fallout (orange dashed circle) but not PCM recruitment. CNN centrosome intensity was measured at NEBD in nuclear cycle 12. Dots represent average intensity of all centrosomes in a single embryo. Error bars are the mean ± SD. *N* = 8 control, 6 *Cmb* mutant embryos. A Mann–Whitney test used to assess statistical significance. Scale bars, 100 nm (D), 500 nm (E), 10 μm (H and I). See also Fig. S2. NEBD, nuclear envelope breakdown; TEM, transmission electron microscopy.

well as other noncentrosomal MTOCs in differentiated cells (Odabasi et al., 2020) (Fig. 3 A). This behavior is consistent with their originally ascribed role in delivery of proteins required for centrosome and cilium function. In stark contrast, while *Drosophila* CMB forms cytoplasmic foci like vertebrate PCM1, these foci show no discernible enrichment near centrosomes or cilia in S2 cultured cells or in any of the fly tissues examined (Fig. 3, B and C), although CMB is occasionally observed at centrosomes in a small fraction of cells (Fig. S3 A). This is true for both rescuing GFP transgene and endogenous CMB, detected using a polyclonal antibody raised against the fly protein (Fig. S3 B). There is furthermore little sign of directed trafficking, with CMB particles moving largely by diffusion or entirely stationary (Fig. 3, D–H).

If CMB is not enriched at centrosomes, how could it have been identified as a proximity interactor of SAS-4 or contribute to centrosome and cilium function? Centriolar satellites have not previously been reported in *Drosophila*. However, it is worth bearing in mind that satellites in vertebrates were originally identified as particles found concentrated in the vicinity of centrosomes (Kubo and Tsukita, 2003), a property CMB foci do not share. Furthermore, aside from PCM1 most satellite proteins primarily localize to centrosomes or cilia, with only a minor population on satellites, such that new constituents are usually identified by colocalization with PCM1 (Dammermann and Merdes, 2002; Gupta et al., 2015). With this in mind, we re-evaluated centrosomal protein localization in *Drosophila* S2 cells and primary spermatocytes using a panel of antibodies to centriolar and PCM proteins in conjunction with CMB. While the majority of each of these proteins indeed localizes to centrosomes, in many cases faint cytoplasmic foci could be identified and these foci colocalized with CMB to an extent similar to what has been reported for satellite clients in vertebrates (Fig. 3 I and Fig. S3, C–E) (Gheiratmand et al., 2019). One notable exception is the microtubule nucleator γ-tubulin, which is one of the few centrosomal proteins not found on vertebrate satellites and only weakly and indirectly affected in satellite perturbations (Dammermann and Merdes, 2002; Gheiratmand et al., 2019; Gupta et al., 2015; Quarantotti et al., 2019). Interestingly, *Drosophila* CMB colocalized with PCM1 on centriolar satellites when expressed in vertebrate cells (Fig. S3 F), although perhaps unsurprisingly given the low degree of sequence homology, we were unable to detect any functional rescue upon PCM1 depletion (Fig. S3 G). Collectively, these results suggest that satellites are conserved in *Drosophila*, although they do not concentrate in the vicinity of centrosomes or cilia.

## Satellites are associated with cytoplasmic protein synthesis in *Drosophila*

In an effort to better understand satellite function in *Drosophila*, we performed proximity interaction analysis on CMB by

conventional, direct, TurboID (Branon et al., 2018) in S2 cells (Fig. 4 A) and indirect, GFP nanobody–targeted, TurboID (Holzer et al., 2022) in fly testes (Fig. 4 B). For the former analysis, we further fractionated cellular extracts to distinguish between interactions occurring in the general cytoplasm and (detergent-insoluble) cytoskeleton. No marked differences between the two cellular contexts were observed, although significantly more interactions were detected in the cytoplasm (Fig. S4, A and B). However, both preparations yielded numerous centrosomal protein proximity interactors and other proteins previously linked to PCM1 in vertebrates, such as the ubiquitin ligase MIB1 (Villumsen et al., 2013) and the deubiquitinase CYLD (Douanne et al., 2019). Overall, there was a striking degree of overlap with the proteome of the centrosome–cilium interface previously defined in vertebrates, with 122 of the 412 CMB proximity interactors conserved in humans (30%) also found among the high-confidence interactors identified by BioID performed on 58 centriole, satellite, and ciliary transition zone proteins (Gupta et al., 2015) (Fig. 4 C), and a lesser but still significant overlap with the PCM1/centriolar satellite proteome defined by Gheiratmand et al. (2019), Quarantotti et al. (2019) (Fig. S4 C). Largely missing were ciliary proteins, S2 cells being a nonciliated cell type. Such proteins were, however, identified in fly testes, including components of the ciliary motility machinery and axonemal dynein assembly factors (Fig. 4, B and C). IFT and BBS proteins do not contribute to cilium biogenesis in *Drosophila* sperm (Dobbelaere et al., 2023; Han et al., 2003) and were not detected.

The identification of centrosomal and ciliary protein proximity interactors also in the fly was perhaps to be expected given previous work on satellites in vertebrates. More surprising was the presence, in both flies and vertebrates, of mRNA-binding proteins, translation initiation factors, and components of the protein quality control machinery (Fig. 4, A and B). Indeed, 37–40% of the CMB interactome conserved in humans was also found in the cytosolic RNA interactome defined by Youn et al. (2018), including components of the 43S preinitiation complex, the most enriched gene ontology (GO) term in our MS samples (Fig. 4, D–F). While there are no obvious RNA-binding motifs in either CMB or its vertebrate counterpart, PCM1 likewise has been repeatedly identified in proteome-wide screens for RNA-binding proteins (e.g., Trendel et al., 2019; see RBP2GO database [Caudron-Herger et al., 2021] for a full listing: https://RBP2GO.DKFZ.de), and 33–55% of the PCM1/centriolar satellite proteome is shared with the cytosolic RNA interactome of Youn et al. (2018) (Fig. S4 D). Might satellites be associated with protein synthesis? To address this question, we performed a puromycin incorporation assay (Fig. 4 G). Puromycin, a structural analog of

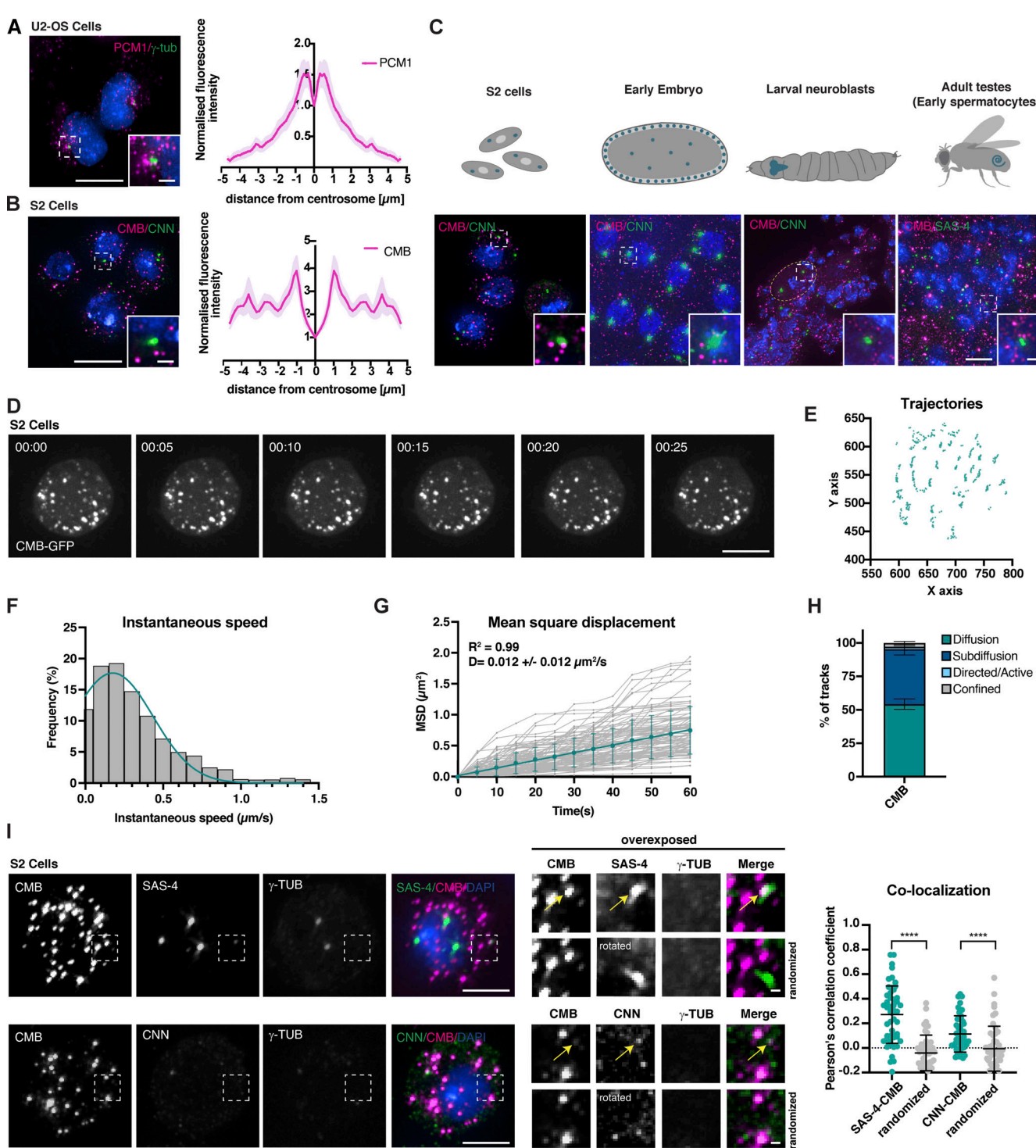

Figure 3. **Centriolar satellites are conserved in *Drosophila* but do not concentrate near centrosomes or cilia, nor move in a directed manner. (A)** PCM1 concentrates in the vicinity of centrosomes (marked with γ-tubulin) in vertebrate cells. The radial profile of the normalized distribution of PCM1 shows the mean ± SEM. *N* = 27 centrosomes. **(B)** CMB localization in *Drosophila* S2 cells. Unlike PCM1, CMB does not concentrate near the centrosome (marked with CNN), but is found throughout the cytoplasm. The radial profile shows the mean ± SEM. *N* = 39 centrosomes. **(C)** Top: Schematics showing cells and tissues used to examine CMB localization/dynamics. Bottom: Immunofluorescence micrographs show CMB localizing to cytoplasmic foci in all tissues examined. SAS-4/CNN was used to visualize centrioles/centrosomes. **(D)** Time-lapse recording of *Drosophila* S2 cell expressing CMB-GFP. Time is shown in min:s. CMB particles display little movement. **(E)** Corresponding trajectory analysis showing the position of individual particles over time. **(F)** Histogram plotting frequency of instantaneous speed (μm/s) of all particles tracked in eight different cells. *N* = 140 particles. The majority of particles moves at slow speeds (0.2–0.3 μm/s), consistent with diffusion. **(G)** MSD as a function of time. Trajectories with at least 11 frames from eight cells were analyzed. All tracks analyzed are shown in gray (*N* = 99 tracks). Weighted MSD (mean ± SD) of all diffusive particles follows a linear fit (turquoise line), reflecting the overall Brownian diffusion. **(H)** Stacked column plot (mean ± SEM) showing all tracks analyzed in G assigned to different categories as described in Materials and methods. The majority of tracks analyzed display the Brownian diffusion (54.2% ±3.9) or subdiffusion/anomalous diffusion (41.1% ±4.3). Only a minor fraction displays directed/active

(1.8% ±0.7) or confined movement (2.9 % ±1.1). **(I)** Centriolar (SAS-4) and centrosomal (CNN) proteins colocalize with CMB on cytoplasmic foci, although immunofluorescence signal is weak compared with that at centrosomes (marked with γ-tubulin). Colocalization of CMB with SAS-4 and CNN was assessed by Pearson's correlation coefficient. Randomized control has one of the channels rotated 90°. Centrosomes were excluded from analysis. Error bars are the mean ± SD. N = 50 cells per condition. A Mann–Whitney test was used to test statistical significance; ****P < 0.0001. Scale bars, 10 μm (A–C), 5 μm (D and I), 1 μm (A–C, insets, I, magnified views). See also Fig. S3. MSD, mean squared displacement.

tyrosyl-tRNA, blocks translation by incorporating into the C terminus of elongating polypeptide chains and triggering their release from ribosomes. Puromycin antibodies can be used to detect those polypeptide chains and, at low doses of puromycin and short incubation times, identify sites of translation (David et al., 2012). Performing this assay in S2 cells revealed clear labeling of CMB foci with puromycin (15% of CMB foci labeled with Puro, Fig. 4 H). Pretreatment of cells with cycloheximide, which binds the ribosome and blocks eEF2-mediated translation elongation (Obrig et al., 1971), abolished puromycin signal, confirming specificity. Consistent with a role of satellites in translation, single-molecule fluorescence in situ hybridization (smFISH) showed the mRNA for SAS-4 to be frequently associated with CMB (Fig. S4, E–G). CMB satellite foci, then, are associated with nascent protein and hence are potentially sites of protein synthesis, at least in *Drosophila*. Could the same be true in vertebrates?

### Satellites are sites of translation of centrosomal and ciliary proteins in vertebrate cells

The puromycin incorporation assay in our hands revealed no distinct cytoplasmic foci when performed in vertebrate cultured cells, potentially due to higher rates of global protein synthesis or the rapid diffusion of nascent polypeptides away from the ribosome even at short incubation times (Enam et al., 2020). However, we were able to detect nascent PCNT, a PCM scaffolding component and satellite client protein (Dammermann and Merdes, 2002), using the puromycylation proximity ligation assay (Puro-PLA), which combines puromycylation with proximity-dependent ligation (Fig. 5 A, [tom Dieck et al., 2015]), and found this signal to be closely associated with PCM1 (Fig. 5 B). Interestingly, longer incubations with high-dose puromycin to stop translation resulted in a striking loss of cytoplasmic signal for most centrosomal and ciliary proteins, while centrosomal signal remained (Fig. 5 C). Similar results were obtained with cycloheximide, again without a concomitant increase in centrosomal signal ruling out protein redistribution from satellites to centrosomes (Fig. S5 A). The satellite population of most centrosomal and ciliary proteins is therefore newly synthesized. Based on the persistence of PCM1 signal, satellites themselves are not lost, although they were now found dispersed throughout the cell (for an explanation of this phenomenon, see below). Similarly unaffected were the ubiquitin ligase MIB1 and OFD1, a centriolar protein previously linked to protein synthesis and turnover (Iaconis et al., 2017; Morleo et al., 2021), which continued to colocalize with PCM1 (Fig. S5 B). This perturbation therefore appears to discriminate between centrosomal and ciliary clients and the machinery potentially involved in their synthesis (MIB1, OFD1).

What the above results do not show is that translation of centrosomal and ciliary clients actually occurs at satellites, as opposed to satellites representing a way station in the eventual translocation of a newly synthesized protein to centrosomes. To distinguish between these possibilities, we performed smFISH to visualize the mRNA for PCNT, a PCM scaffolding component and satellite client protein (Dammermann and Merdes, 2002). As seen in Fig. 5 D and Fig. S5 C, *PCNT* mRNA localized in close proximity to both its cognate protein and PCM1, indicating that PCNT is translated at or in the immediate vicinity of satellites (see Fig. S5 D for a control of smFISH signal specificity). We also observed satellite association of the mRNA for CEP290, a satellite client and component of the ciliary transition zone (Kim et al., 2008) (Fig. S5 E), though not for RANBP10, an unrelated housekeeping protein expressed at similar levels to PCNT and CEP290 (Hounkpe et al., 2021) (Fig. S5 F). The close proximity of *PCNT* mRNA to PCM1 remained in cells treated with puromycin and therefore devoid of the cytoplasmic PCNT protein, indicating that PCM1 associates with *PCNT* mRNA, not (or not only) protein (Fig. 5 D). PCM1-containing satellites consequently act as sites where centrosomal and ciliary mRNAs concentrate and are potentially coordinately translated within the same satellite particle. The previously documented loss of satellite signal for centrosomal and ciliary clients following PCM1 depletion (Dammermann and Merdes, 2002; Kim et al., 2004; Kodani et al., 2015; see Fig. 5 A and Fig. S5 G) therefore appears to reflect a failure to synthesize new proteins in situ on satellites, not merely to concentrate them there in preparation for transit to centrosomes. Interestingly, putative components of the satellite protein synthesis machinery are differentially affected by PCM1 depletion, with MIB1 dispersing throughout the cytoplasm, while OFD1 continues to localize to centrosomes while losing its cytoplasmic population (Fig. S5 G). These results are consistent with PCM1 acting as the main scaffolding component of cytoplasmic translation factories specifically for centrosomal and ciliary proteins.

### Cotranslational targeting explains centrosomal enrichment of centriolar satellites in vertebrate cells

This left one key question: Why do satellites concentrate in the vicinity of centrosomes and the base of cilia in vertebrates? We hypothesized that this might be due to the previously reported phenomenon of cotranslational targeting of certain centrosomal proteins including PCNT, which is recruited to centrosomes along with its mRNA at the onset of mitosis in a translation-, microtubule-, and dynein-dependent manner (Lerit, 2022; Safieddine et al., 2021; Sepulveda et al., 2018). *PCNT* mRNA highly concentrates at centrosomes in prophase/prometaphase, but then disperses in metaphase/anaphase (Sepulveda et al., 2018), precisely the same pattern observed for PCM1-

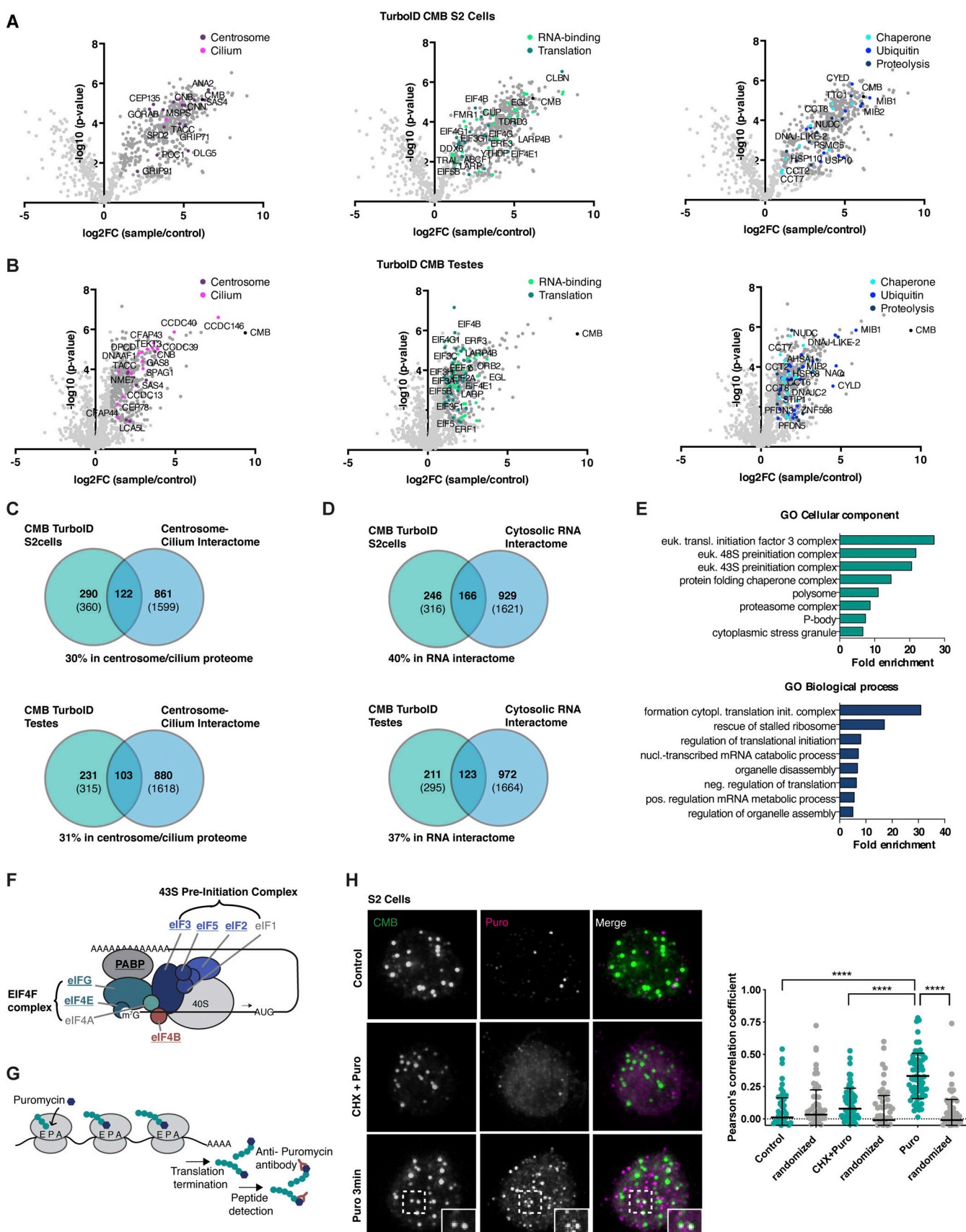

Figure 4. **CMB is associated with protein synthesis in *Drosophila*. (A and B)** Direct TurboID of CMB in S2 cells (cytosolic fraction, A) and indirect, GFP nanobody–targeted TurboID in fly testes (B) identify centrosomal (magenta) and ciliary proteins (pink), RNA-binding proteins (light green), and proteins

involved in translation (dark green), chaperone-mediated protein folding (light blue), ubiquitination (blue), and proteolysis (dark blue). Volcano plots of $-\log_{10}$ P values against $\log_2$ fold change (sample/control). Significantly enriched proteins ($\log_2$ enrichment >1, P <0.05) are indicated in dark gray, with proteins of the above functional categories highlighted in color. See also Table S2 B and D. **(C)** Venn diagrams showing the overlap between CMB S2 cell and testis TurboID interactomes and human centrosome/cilium proteome defined by Gupta et al. (2015). Comparison of those proteins conserved between human and flies. Numbers in parentheses are total number in each dataset. See also Table S2 E. **(D)** Venn diagrams showing an overlap between CMB S2 cell and testis TurboID interactomes and cytosolic RNA interactomes defined by Youn et al. (2018). Comparison of those proteins conserved between human and flies. Numbers in parentheses are total number in each dataset. See also Table S2 E. **(E)** GO enrichment analysis performed on human orthologs of the CMB TurboID testis dataset. The top eight terms and their fold enrichments are shown for the GO categories cellular component and biological process. **(F)** Schematic of eukaryotic translation initiation (Jackson et al., 2010). The 43S-preinitiation complex is recruited to the mRNA by the EIF4F complex through interaction of EIF4G with eIF3. Components identified in the CMB interactome are highlighted in bold. **(G)** Schematic of the puromycin labeling assay. Puromycin mimics tyrosyl-tRNAs and binds the ribosomal acceptor site, blocking translation. Nascent peptide chains labeled with puromycin (puromycylated) are released into the cytoplasm and can be detected using antibodies against puromycin. **(H)** Puromycin labeling performed in *Drosophila* S2 cells. Puromycin labels CMB foci in the cytoplasm after brief incubation with puromycin. No signal is detected in control cells or cells pretreated with cycloheximide before the addition of puromycin. Co-localization of CMB and puromycin label was quantified by Pearson's correlation coefficient analysis. To exclude random colocalization, distribution was compared with randomized controls (see Materials and methods). Error bars are the mean ± SD. N = 70 cells per condition. A Kruskal–Wallis test followed by Dunn's multiple comparisons test was used to assess statistical significance; ****$P < 0.0001$. Scale bars, 5 μm (H), 1 μm (H, insets). See also Fig. S4.

containing satellites (Dammermann and Merdes, 2002). PCM1 depletion eliminates *PCNT* mRNA enrichment without affecting total cellular mRNA abundance as assessed in interphase (Fig. 6, B and C). In the original transport model of PCM1 function, this would reflect the role of satellites in moving nascent PCNT protein and hence indirectly also its mRNA to centrosomes. Alternatively, cotranslational protein transport of PCNT, with or without direct involvement of the satellite machinery, would drive their accumulation at centrosomes. If so, loss of PCNT should affect PCM1 localization in mitosis, while in the original model, PCM1 localization would be unaffected. The former is what we observed: PCM1 no longer concentrates at centrosomes in prophase/prometaphase of *PCNT* mutant cells (Watanabe et al., 2020), while its interphase localization is unaffected (Fig. 6 D). Importantly, despite reported defects in mitotic PCM organization (Haren et al., 2009; Lawo et al., 2012) PCNT depletion does not impair assembly and organization of the mitotic spindle (Watanabe et al., 2020) and clear foci of γ-tubulin remain (Fig. 6 D). Microtubule trafficking should therefore be largely unperturbed. The previously reported movement of satellites along microtubules (Kubo et al., 1999) and their concentration at centrosomes in vertebrates are therefore a result of cotranslational targeting of certain client proteins, independent of their function in translation itself. PCNT is likely not the only satellite client that is cotranslationally targeted. The dispersal of satellites following puromycin treatment in interphase cells (see above) indicates that their concentration near centrosomes at this stage is likewise a result of their function in translation of one or more centrosomal clients, while such cotranslational targeting may be a less prominent feature of their *Drosophila* counterparts.

## Discussion

Since their discovery more than 30 years ago, the specific functional contribution of centriolar satellites to centrosome and cilium assembly has continued to excite researchers. Their dynamic localization pattern has inspired a variety of hypotheses, of which the most prominent remains the transport model originally put forward in the early 2000s (Dammermann and Merdes, 2002), according to which satellites are involved in

microtubule- and dynein-dependent transport of centrosomal and ciliary cargo. Here, we demonstrate that PCM1 and centriolar satellites are conserved outside of vertebrates. Characterizing their role in *Drosophila* reveals not only similarities but also differences in their dynamics, composition, and function. Like vertebrate PCM1 (Hall et al., 2023), depletion or mutation of *Drosophila* CMB results in centrosomal and ciliary phenotypes. This, together with a pattern of inheritance and loss across opisthokonts matching that of centriole-based structures, is consistent with a functional association of satellites with centrosomes and cilia conserved across evolution. Proximity interaction mapping in both S2 cultured cells and whole flies likewise reveals a set of conserved set of interactors (Gheiratmand et al., 2019; Gupta et al., 2015; Quarantotti et al., 2019). These include not only centrosomal and ciliary proteins, but also proteins like MIB1, CYLD, and OFD1 linked to translational initiation and protein quality control (Douanne et al., 2019; Iaconis et al., 2017; Morleo et al., 2021; Villumsen et al., 2013; Wang et al., 2016). This finding, along with the evident lack of centrosomal enrichment and directed movement of satellites in the fly, led us to examine an alternate hypothesis of satellites as sites of local translation. Consistent with this hypothesis, satellite foci in both *Drosophila* and vertebrates are associated with nascent protein, and PCM1 colocalizes with centrosomal and ciliary mRNA independent of translation.

In our revised view of satellites as hubs of translation (Fig. 6 E), movement can and does occur as a result of cotranslational targeting of client proteins, at least in vertebrates. It is unclear whether the core satellite machinery is directly involved in mediating that movement. What is clear is that inhibiting protein synthesis or removing particular satellite client proteins impairs satellite centrosomal localization. When viewed from this perspective, both their highly variable localization patterns in different cell types, cell cycle stages, and environmental conditions (Baron et al., 1994; Dammermann and Merdes, 2002; Villumsen et al., 2013) begin to make sense. Depending on the levels of synthesis (and nature) of different client proteins, satellites may concentrate in the vicinity of centrosomes, as they do in prophase/prometaphase, or be scattered throughout the cytoplasm. Thus, the reported dispersal of satellites in response to cellular stresses (Villumsen et al., 2013) is to be expected given

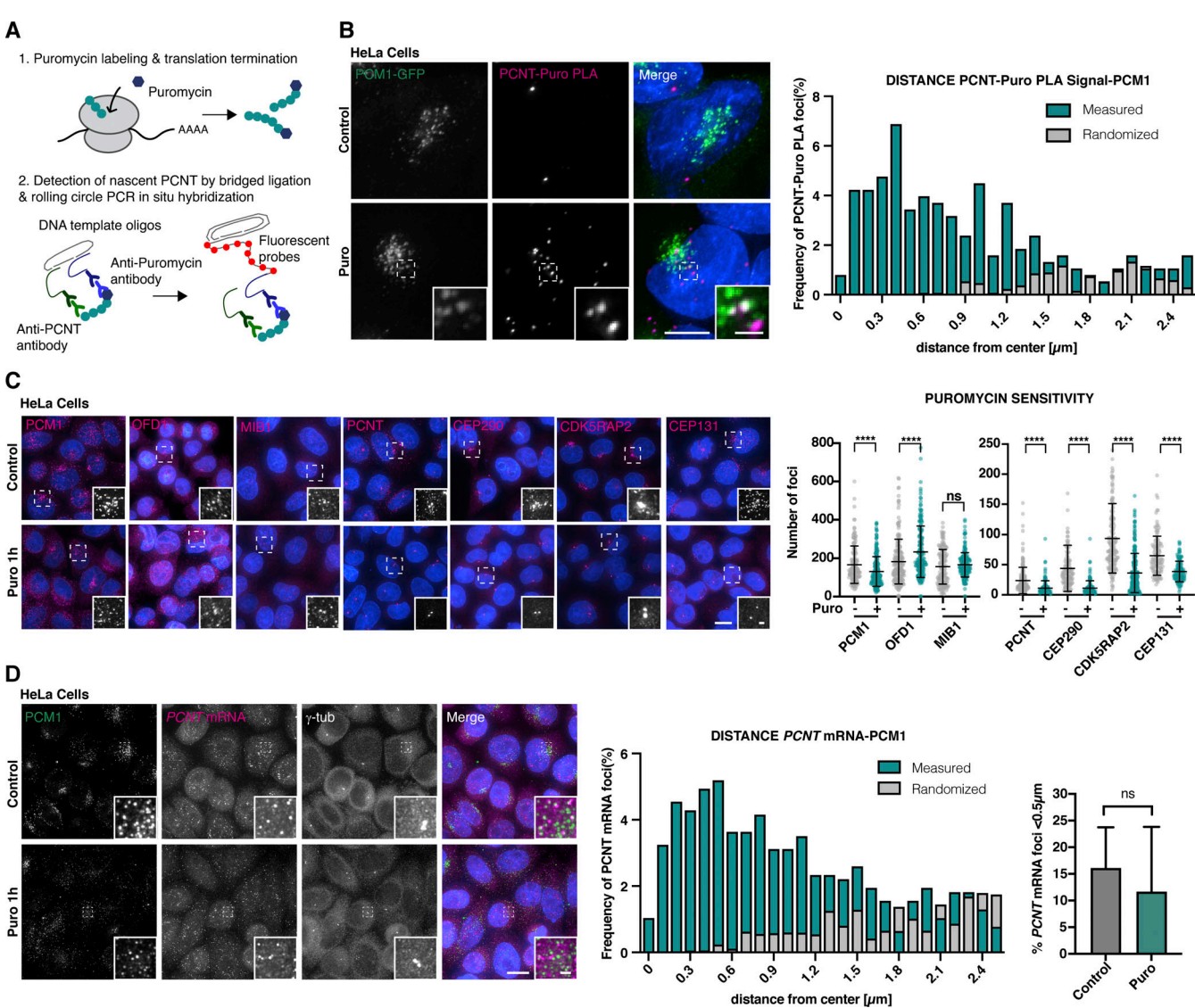

Figure 5. **Centriolar satellites are sites of translation of centrosomal and ciliary proteins in vertebrate cells. (A)** Schematic of the Puro-PLA. Puro-PLA combines puromycin labeling with proximity-dependent ligation to make labeling specific for a particular protein of interest, here PCNT. **(B)** Puro-PLA reveals nascent PCNT in proximity to centriolar satellites visualized using PCM1-GFP. To exclude random colocalization/proximity, distribution was compared with randomized controls (see Materials and methods). N = 109 cells. **(C)** Immunofluorescence micrographs and quantitation of centriolar satellite protein signal in control and puromycin-treated HeLa cells. Centrosomal and ciliary proteins PCNT, CEP290, CDK5RAP2, and CEP131 are significantly depleted of their cytoplasmic localization upon puromycin treatment, whereas their centrosome localization persists (centrosomes identified using anti-γ-tubulin coundmarker, not shown), suggesting the former represents newly synthesized protein. In contrast, foci of PCM1, OFD1, and MIB1 remain, although now dispersed throughout the cytoplasm. >100 cells were analyzed per condition. Mean ± SD are displayed. A Mann–Whitney test was used to assess statistical significance; ****P < 0.0001. **(D)** smFISH combined with immunofluorescence microscopy in control and puromycin-treated HeLa cells. *PCNT* mRNA localizes in the vicinity of PCM1 independently of ongoing translation. To exclude random colocalization/proximity, distribution was compared with randomized controls (see Materials and methods). Error bars are the mean ± SD. N = 77 cells (control), 102 cells (puromycin). Scale bars, 5 µm (B and D), 10 µm (C), 1 µm (insets). See also Fig. S5.

the attendant shutdown of protein synthesis. The association of satellites with mRNAs encoding centrosomal and ciliary proteins is likely to also modulate the reported liquid–liquid phase separation behavior of satellites (Rai et al., 2018; Zhao et al., 2021). RNA is known to buffer phase separation behavior in the case of other membraneless organelles (Maharana et al., 2018) and may act here to prevent the merger of satellites into large aggregates as occurs in certain PCM1 perturbations (Dammermann and Merdes, 2002; Kubo and Tsukita, 2003). Such aggregation is

known to perturb PCM1 function (Dammermann and Merdes, 2002) and is likely to be disease-relevant given the link of PCM1-containing satellites to ciliopathies and neurodegenerative disorders (Keryer et al., 2011; Lopes et al., 2011).

The lack of any evident concentration of satellites in the vicinity of centrosomes or cilia in *Drosophila* shows that their function in translation is independent of their localization and therefore separable from their potential role in cotranslational protein trafficking and centrosomal mRNA localization in

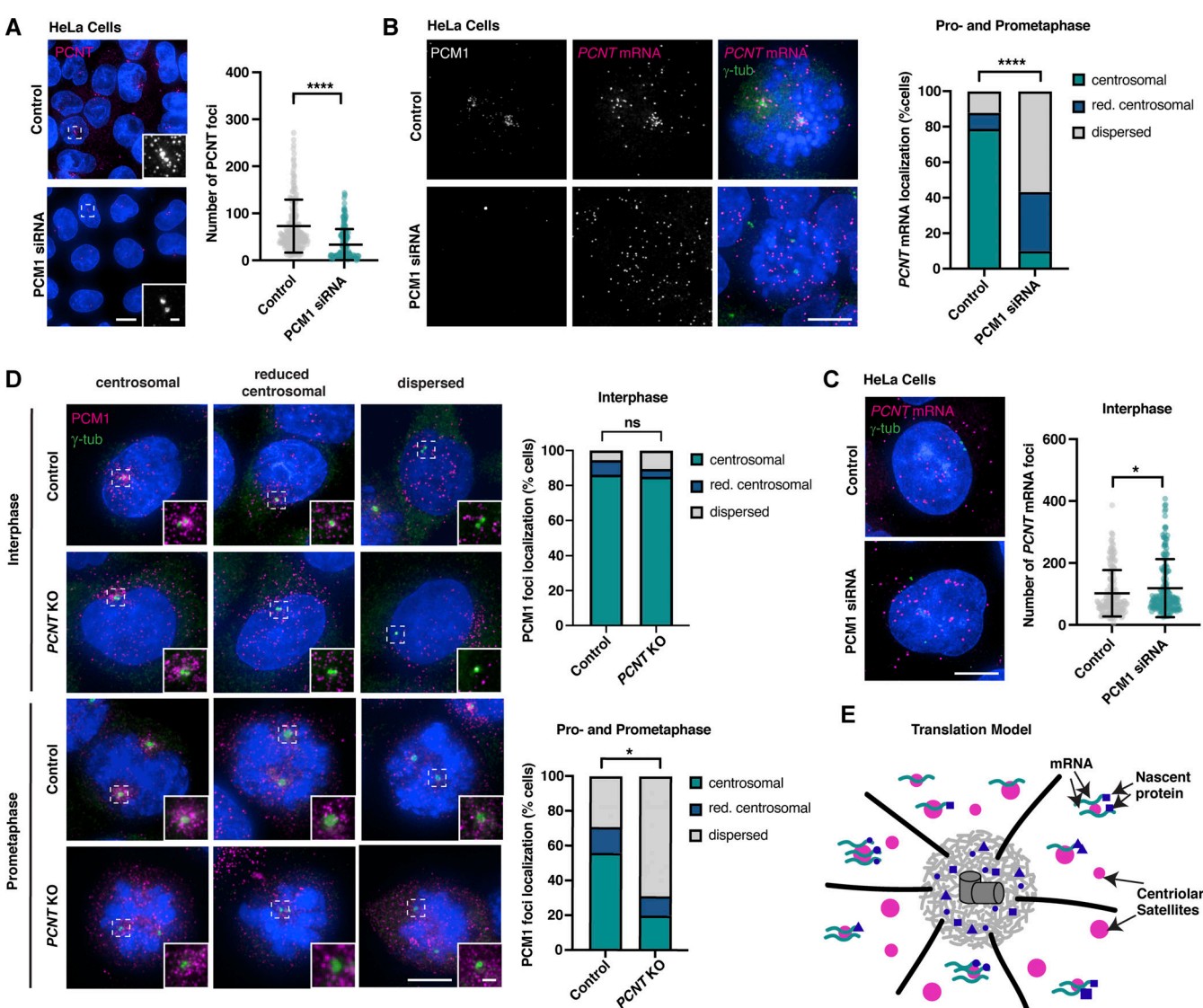

Figure 6. **Centriolar satellite distribution reflects cotranslational transport of certain centrosomal/ciliary clients. (A)** Immunofluorescence micrographs and quantification of cytoplasmic PCNT signal in control and PCM1 siRNA-treated cells. Depletion of PCM1 leads to a significant decrease in the number of PCNT foci. Error bars are the mean ± SD. $N$ = 211 control cells and 171 PCM1 siRNA cells. Data were analyzed using the Mann–Whitney test; ****$P$ < 0.0001. **(B and C)** smFISH combined with immunofluorescence microscopy in control cells and cells depleted of PCM1. PCM1 depletion impairs *PCNT* mRNA localization around the centrosome in prometaphase (B), reflecting a loss of cotranslational targeting. However, the total number of *PCNT* mRNA foci assessed in interphase remains almost unchanged (C). Error bars in (C) are the mean ± SD. $N$ = 33 prometaphase-stage and 189 interphase cells (control), 30 and 196 cells (PCM1 siRNA). A Mann–Whitney test was used to assess statistical significance; ****$P$ < 0.0001, *$P$ < 0.05. **(D)** Immunofluorescence micrographs of PCM1 in control and *PCNT* KO cells. Loss of PCNT does not impact PCM1 localization in interphase but compromises its concentration near centrosomes in pro- and prometaphase. $N$ = 154 prometaphase and 348 interphase cells (control), 161 and 355 cells (*PCNT* KO). Localization groups were compared using the Mann–Whitney test, *$P$ < 0.05. **(E)** Revised model of centriolar satellite function: centriolar satellites are sites of translation of centrosomal and ciliary proteins. Their concentration close to centrosomes in vertebrates is a byproduct of cotranslational targeting of certain centrosomal/ciliary clients including PCNT. Scale bars, 10 µm (A), 5 µm (B–D), 1 µm (insets).

vertebrates (Lerit, 2022; Safieddine et al., 2021; Sepulveda et al., 2018). In this respect, protein synthesis at satellites differs from other reported instances of local translation, such as in the developing *Drosophila* oocyte or in the axons and dendrites of vertebrate neurons, where it is thought to facilitate protein delivery to distant cellular locations or avoid inappropriate protein interactions en route (Bourke et al., 2023). We suggest that the primary role of satellites is instead to facilitate the coordinate synthesis of members of the same multiprotein complexes for

subsequent delivery to centrosomes and cilia, another proposed benefit of local translation (Chouaib et al., 2020; Morales-Polanco et al., 2022). The enrichment of chaperones and proteins involved in quality control on satellites is consistent with that hypothesis. It is important to note that general protein synthesis of centrosomal and ciliary proteins still occurs in PCM1 perturbations, with protein abundance reported to be only marginally affected, both globally and at centrosomes (Quarantotti et al., 2019). Such a role in refinement of

centrosomal/ciliary protein expression is consistent with the rather minor yet highly pleiotropic consequences of PCM1 perturbation (Gheiratmand et al., 2019; Hall et al., 2023; Quarantotti et al., 2019), a long-standing conundrum in the field. While satellites appear to be solely concerned with synthesis of centrosomal and ciliary proteins, how selectivity is achieved in this case, as in other instances of local translation, remains unclear, but may involve primary sequence or secondary structure motifs ("zip codes" [Martin and Ephrussi, 2009; Mayr, 2017]) present in centrosomal/ciliary mRNAs. Ribosome heterogeneity (Shi and Barna, 2015) may also play a role. We may not be returning to the extreme of "one gene–one ribosome–one protein" (Crick, 1958), but cellular protein synthesis appears to be much more organized than commonly appreciated. A better understanding of the role of proteins like PCM1 is clearly key to understanding how cells orchestrate the coordinate synthesis of the components that make up their various constituent structures.

## Materials and methods

### *Drosophila melanogaster* stocks and husbandry

*Cmb* mutants and flies overexpressing CMB-RA under GAL4/UAS control (Fagan et al., 2014) were a gift from Andreas Jenny (Albert Einstein College of Medicine, New York, NY, USA), flies expressing RFP-CNN (Conduit et al., 2010), H2A-RFP (Pandey et al., 2005), and ASL-GFP (Blachon et al., 2008) from Jordan Raff (University of Oxford, Oxford, UK), and Bam-GAL4 (Chen and McKearin, 2003) and Pnr-GAL4 (Calleja et al., 2000) driver lines from Helen White-Cooper (Cardiff University, Cardiff, UK) and Jürgen Knoblich (IMBA, Vienna, Austria), respectively. Flies expressing GFP nanobody–TurboID under the *eggless* promoter (derived from an NLS-containing construct generated by the laboratory of Julius Brennecke [IMBA, Vienna, Austria] [Baumgartner et al., 2022]) were generated by integration into the attP40 landing site on chromosome 2 by the IMBA Fly Facility. All other fly strains were obtained from the Vienna *Drosophila* Resource Center (VDRC) or Bloomington *Drosophila* Stock Center (BDSC) or generated by genetic crosses as detailed in Table 1. Further information on strains is available on FlyBase (https://flybase.org), as well as on the websites of the two stock centers (BDSC, https://bdsc.indiana.edu, and VDRC, https://stockcenter.vdrc.at/control/main). Flies were maintained on standard media at 25°C. W[1118] flies were used as controls for all experiments. For RNAi, UAS-hairpin RNAi males and GAL4 driver line females were crossed in standard vials and allowed to mate for 4 days at 25°C, then shifted to 29°C to induce GAL4 expression.

### Insect and vertebrate cell culture

*Drosophila* Schneider S2 cells were obtained from Life Technologies and cultured in Schneider's medium containing 10% FBS, penicillin (50 U/ml), and streptomycin (50 μg/ml). Cells were kept at 25°C at atmospheric $CO_2$ and passaged every 3–4 days.

For proximity interaction analysis of SAS-4 by BioID in S2 cells, a construct expressing myc-BirA*(R118G) under the copper-inducible metallothionein promoter (pMt-myc-BirA*)

Table 1.  **Reagents and tools table**

| Reagent/resource | Reference or source | Identifier or catalog number |
|---|---|---|
| Experimental models | | |
| Cell lines | | |
| *D. melanogaster*: S2 cells | Life Technologies | Cat# R69007, RRID: CVCL_Z232 |
| Human: HeLa cells | ATCC | Cat# CCL-2, RRID: CVCL_0030 |
| Human: HeLa *PCNT* KO cells | Watanabe et al. (2020) | ODCL0025 |
| Human: U2-OS cells | ATCC | Cat# HTB-96, RRID: CVCL_0042 |
| Organisms/strains | | |
| *D. melanogaster*: Asl RNAi/P{TRiP.HMS01453}attP2 | BDSC | BDSC ID: 35039, RRID: BDSC_35039 |
| *D. melanogaster*: Asl-GFP & H2A-RFP; CmbΔ/P{His2Av-mRFP1}; P{UAS.asl-asl-GFP.B}; CmbΔ/Tmb6 SbTb | This study | N/A |
| *D. melanogaster*: Asl-GFP & H2A-RFP/P{His2Av-mRFP1}; P{UAS.asl-asl-GFP.B} | This study | N/A |
| *D. melanogaster*: Asl-GFP/P{UAS.asl-asl-GFP.B} | Blachon et al. (2008) | FlyBase ID: FBtp0040947 |
| *D. melanogaster*: Bam-GAL4 & Nanos-GAL4/P{GAL4-nos.NGT}40; P{bam-GAL4:VP16} | Dobbelaere et al. (2023) | N/A |
| *D. melanogaster*: Cmb OE/P{UAS-cmb.RA} | Fagan et al. (2014) | FlyBase ID: FBtp0095481 |
| *D. melanogaster*: Cmb RNAi 1/P{KK103563}VIE-260B | VDRC | VDRC ID: 109767, RRID: Flybase_FBst0481420 |
| *D. melanogaster*: Cmb RNAi 2/P{GD7077}v18666 | VDRC | VDRC ID: 18666, RRID: Flybase_FBst0453208 |
| *D. melanogaster*: Cmb-GFP rescue/PBac{fTRG01063.sfGFP-TVPTBF}VK00002; CmbΔ | This study | N/A |
| *D. melanogaster*: Cmb-GFP/PBac{fTRG01063.sfGFP-TVPTBF}VK00002 | VDRC | VDRC ID: 318739, RRID: Flybase_FBst0491914 |
| *D. melanogaster*: CmbΔ/CmbΔ/Tm6 SbTb | Fagan et al. (2014) | FlyBase ID: FBal0298847 |
| *D. melanogaster*: Cnn RNAi/P{GD1726}v44526 | VDRC | VDRC ID: 44526, RRID: Flybase_FBst0465625 |
| *D. melanogaster*: Control RNAi/P{TRiP.HMS00045}attP2 | BDSC | BDSC ID: 33644, RRID: BDSC_33644 |
| *D. melanogaster*: Deficiency 1/w[1118]; Df(3L)BSC614/TM6C, cu[1] Sb[1] | BDSC | BDSC ID: 25689, RRID: BDSC_25689 |
| *D. melanogaster*: Deficiency 2/w[1118]; Df(3L)BSC737/TM6C, Sb[1] cu[1] | BDSC | BDSC ID: 26835, RRID: BDSC_26835 |

Table 1.  **Reagents and tools table** (*Continued*)

| Reagent/resource | Reference or source | Identifier or catalog number |
|---|---|---|
| *D. melanogaster*: Elav-GAL4/P{GawB}elav[C155] | BDSC | BDSC ID: 458, RRID: BDSC_458 |
| *D. melanogaster*: Frtz RNAi/P{GD9437}v40088 | VDRC | VDRC ID: 40088, RRID: Flybase_FBst0463379 |
| *D. melanogaster*: GFP nanobody–TurboID/P{egg-TurboID.vhhGFP.3xHA} | This study | N/A |
| *D. melanogaster*: H2A-RFP/P{His2Av-mRFP1} P{UAS.asl-asl-GFP.B} | Pandey et al. (2005) | FlyBase ID: FBtp0056035 |
| *D. melanogaster*: In RNAi/P{GD10892}v27252 | VDRC | VDRC ID: 27252, RRID: Flybase_FBst0456836 |
| *D. melanogaster*: Plp RNAi/P{GD9625}v20667 | VDRC | VDRC ID: 20667 |
| *D. melanogaster*: Pnr-GAL4/P{GawB}pnr[MD237] | Calleja et al. (2000) | FlyBase ID: FBti0004011 |
| *D. melanogaster*: RFP-Cnn; CmbΔ/P{Ubi-p63E-cnn.RFP}; CmbΔ/Tmb6 SbTb | This study | N/A |
| *D. melanogaster*: RFP-Cnn/P{Ubi-p63E-cnn.RFP} | Conduit et al. (2010) | FlyBase ID: FBtp0056991 |
| *D. melanogaster*: Sas-4 RNAi/P{GD6852}v17975 | VDRC | VDRC ID: 17975, RRID: Flybase_FBst0452943 |
| *D. melanogaster*: Spd-2 RNAi/P{KK110116}VIE-260B | VDRC | VDRC ID: 101882, RRID: Flybase_FBst0473755 |
| *D. melanogaster*: Wild-type/w[1118] | BDSC | BDSC ID: 6326, RRID: BDSC_6326 |
| Recombinant DNA | | |
| Blasticidin selection vector: pCoBlast | Thermo Fisher Scientific | Cat# K515001 |
| GST expression vector: pGEX-6P-1 Cmb (aa1-150) | This study | N/A |
| BioID control vector: pMt-myc-BirA* | This study | N/A |
| BioID SAS-4 vector: pMt-Sas-4-myc-BirA* | This study | N/A |
| S2 cell expression vector: pMt-V5-6xHis B | Invitrogen | Cat# V412020 |
| CMB-GFP vector: pMt-V5-6xHis-Cmb-GFP | This study | N/A |
| TurboID CMB vector: pMt-V5-TurboID-Cmb | This study | N/A |
| TurboID control vector: pMt-V5-TurboID-GW | Norbert Perrimon | N/A |
| Indirect TurboID fly expression vector: pUASz-TurboID-vhhGFP-3xHA | This study | N/A |
| Indirect TurboID expression vector (+NLS): pUASz-TurboID-vhhGFP-3xHA-NLS | Baumgartner et al. (2022) | N/A |

Table 1.  **Reagents and tools table** (*Continued*)

| Reagent/resource | Reference or source | Identifier or catalog number |
|---|---|---|
| Gateway vector: pDONR/Zeo | Thermo Fisher Scientific | Cat# 12535035 |
| BioID source vector: pcDNA3.1 myc-BirA* | Kyle Roux | N/A |
| CMB-GFP vector: pcDNA3.1 Cmb-GFP | This study | N/A |
| PCM1 GFP vector: pEGFP-N-PCM1-GFP | This study | N/A |
| Antibodies | | |
| Donkey anti-goat IgG (H + L) Cy2-conjugated antibody | Jackson ImmunoResearch | Cat# 705-225-147, RRID: AB_2307341 |
| Donkey anti-mouse IgG (H + L) Cy2-conjugated antibody | Jackson ImmunoResearch | Cat# 715-225-150, RRID: AB_2340826 |
| Donkey anti-mouse IgG (H + L) Cy3-conjugated antibody | Jackson ImmunoResearch | Cat# 715-165-150, RRID: AB_2340813 |
| Donkey anti-mouse IgG (H + L) Cy5-conjugated antibody | Jackson ImmunoResearch | Cat# 715-175-150, RRID: AB_2340819 |
| Donkey anti-rabbit IgG (H + L) Cy2-conjugated antibody | Jackson ImmunoResearch | Cat# 711-225-152, RRID: AB_2340612 |
| Donkey anti-rabbit IgG (H + L) Cy3-conjugated antibody | Jackson ImmunoResearch | Cat# 711-165-152, RRID: AB_2307443 |
| Donkey anti-rabbit IgG (H + L) Cy5-conjugated antibody | Jackson ImmunoResearch | Cat# 711-175-152, RRID: AB_2340607 |
| Goat polyclonal anti-GFP antibody | Dammermann et al. (2004) | N/A |
| Mouse monoclonal anti-dmNompC antibody | Liang et al. (2011) | N/A |
| Mouse monoclonal anti-gamma tubulin antibody | Sigma-Aldrich | Cat# T6557, RRID: AB_477584 |
| Mouse monoclonal anti-hsPCM1 antibody | Santa Cruz Biotechnology | Cat# sc-398365, RRID: AB_2827155 |
| Mouse monoclonal anti-puromycin | Merck Millipore | Cat# MABE343, RRID: AB_2566826 |
| Rabbit polyclonal anti-dmANA1 antibody | Conduit et al. (2010) | N/A |
| Rabbit polyclonal anti-dmANA2 antibody | Cabral et al. (2019) | N/A |
| Rabbit polyclonal anti-dmCEP97-N-ter antibody | Dobbelaere et al. (2020) | N/A |
| Rabbit polyclonal anti-dmCMB antibody | This study | N/A |
| Rabbit polyclonal anti-dmCNN antibody | Dobbelaere et al. (2008) | N/A |
| Rabbit polyclonal anti-dmCP110 antibody | Franz et al. (2013) | N/A |
| Rabbit polyclonal anti-dmSAS-4 antibody | Basto et al. (2006) | N/A |
| Rabbit polyclonal anti-dmSPD-2 antibody | Cabral et al. (2019) | N/A |

| Reagent/resource | Reference or source | Identifier or catalog number |
|---|---|---|
| Rabbit polyclonal anti-hsCDK5RAP2 antibody | Merck Millipore | Cat# 06-1398, RRID: AB_11203651 |
| Rabbit polyclonal anti-hsCEP131/AZI1 antibody | Abcam | Cat# ab84864, RRID: AB_1859791 |
| Rabbit polyclonal anti-hsCEP290 antibody | Novus Biologicals | Cat# NB100-86991, RRID: AB_1201171 |
| Rabbit polyclonal anti-hsMIB1 antibody | Sigma-Aldrich | Cat# M5948; RRID: AB_1841007 |
| Rabbit polyclonal anti-hsOFD1 antibody | Sigma-Aldrich | Cat# HPA031103, RRID: AB_10602188 |
| Rabbit polyclonal anti-hsPCM1 antibody | Dammermann and Merdes (2002) | N/A |
| Rabbit polyclonal anti-hsPCNT antibody | Abcam | Cat# ab4448, RRID: AB_304461 |
| Oligonucleotides and other sequence-based reagents | | |
| siRNA targeting sequence: PCM-1 #1: 5'-GGGCUCUAA ACGUGCCUCC-3' | Dammermann and Merdes (2002) | N/A |
| siRNA targeting sequence: PCM-1 #2: 5'-UCAGCUUCG UGAUUCUCAG-3' | Dammermann and Merdes (2002) | N/A |
| RNA FISH probes: see Table S3 | Sigma-Aldrich/ Biosearch Technologies | N/A |
| smiFISH secondary probe: 5'-CACTGAGTCCAGCTC GAAACTTAGGAGG-3' | Biosearch Technologies | N/A |
| Chemicals, enzymes, and other reagents | | |
| Agar 100 resin | Agar Scientific | Cat# AGR1031; CAS: 25038–04-4 |
| UltraPure BSA, Nonacetylated | Invitrogen | Cat# AM2618 |
| Cycloheximide | Sigma-Aldrich | Cat# C1988; CAS: 66-81-9 |
| 25% glutaraldehyde, EM grade | Agar Scientific | Cat# AGR1020; CAS: 111-30-8 |
| Hoechst 33258 | Thermo Fisher Scientific | Cat# H1398; CAS: 23491–45-4 |
| Lysyl endopeptidase (LysC) | FUJIFILM Wako Chemicals | Cat# 129-02541 |
| Osmium tetroxide 4% solution | Electron Microscopy Sciences | Cat# 19170; CAS: 20816–12-0 |
| Paraformaldehyde | Roth | Cat# O335.1; CAS: 30525-89-4 |
| 32% paraformaldehyde, EM grade | Electron Microscopy Sciences | Cat# 15714 |
| Phalloidin/Alexa Fluor 568 | Thermo Fisher Scientific | Cat# A12380 |
| Pierce Streptavidin Magnetic Beads | Thermo Scientific Pierce | Cat# 88817 |

| Reagent/resource | Reference or source | Identifier or catalog number |
|---|---|---|
| Polyethylenimine | Polysciences | Cat# 23966; CAS: 26913-06-4 |
| Puromycin dihydrochloride | Sigma-Aldrich | Cat# P8833 |
| Reynolds' lead citrate | Delta Microscopies | Cat# 11300; CAS: 512–26-5 |
| Trypsin Gold Mass Spec Grade | Promega | Cat# V5280 |
| Uranyl acetate | Science Services | Cat# E22400; CAS: 541–09-3 |
| Vanadyl ribonucleoside complexes solution | Sigma-Aldrich | Cat# 94742 |
| Vectashield Plus | Vector Laboratories | Cat# H-1900 |
| Critical commercial assays | | |
| Duolink In Situ Detection Reagents Red | Sigma-Aldrich | Cat# DUO92008 |
| Duolink In Situ PLA Probe Anti-Mouse MINUS | Sigma-Aldrich | Cat# DUO92004 |
| Duolink In Situ PLA Probe Anti-Rabbit PLUS | Sigma-Aldrich | Cat# DUO92002 |
| Duolink In Situ Wash Buffers, Fluorescence | Sigma-Aldrich | Cat# DUO82049 |
| Lipofectamine RNAiMAX | Invitrogen | Cat# 13778100 |
| TransIT Insect Transfection reagent | Mirus Bio | Cat# MIR 6104 |
| Software | | |
| Fiji v 2.0.0 | NIH | https://imagej.net/software/fiji/ |
| FragPipe v 20.0 | University of Michigan Medical School | https://github.com/Nesvilab/FragPipe |
| GraphPad Prism 10 | GraphPad | https://www.graphpad.com/scientific-software/prism |
| HMMER v 3.4 | Eddy (2011) | https://hmmer.org/ |
| Ilastik v 1.4.0 | Berg et al. (2019) | https://www.ilastik.org |
| IonQuant v 1.9.8 | Yu et al. (2021) | https://github.com/Nesvilab/IonQuant |
| Jalview v 2.11.1.4 | Waterhouse et al. (2009) | https://www.jalview.org/ |
| MAFFT v 7 | Katoh et al. (2019) | https://mafft.cbrc.jp/alignment/server/ |
| MATLAB v R2023a | MathWorks | https://www.mathworks.com |
| MSFragger v 3.8 | Kong et al. (2017) | https://github.com/Nesvilab/MSFragger |
| MsReport v 0.0.23 | Max Perutz Labs Mass Spectrometry Facility | N/A |
| NCBI BLAST+ v 2.13.0 | Altschul et al. (1997) | https://ftp.ncbi.nlm.nih.gov/blast/executables/blast+/LATEST/ |

**Table 1.  Reagents and tools table (Continued)**

| Reagent/resource | Reference or source | Identifier or catalog number |
|---|---|---|
| Philosopher v 5.0.0 | da Veiga Leprevost et al. (2020) | https://github.com/Nesvilab/philosopher |

was generated by inserting the myc-BirA* coding sequence (derived from a plasmid obtained from Kyle Roux) into the pMt-V5-6xHisB vector backbone (Invitrogen). The coding sequence of *Sas-4* was then inserted into that construct by PCR and restriction cloning. For CMB TurboID, the coding sequence of *Cmb*-RA was cloned into the pDONR/Zeo entry vector (Invitrogen) and then transferred into the pMT-TurboID-V5 vector (a gift from Norbert Perrimon [Harvard Medical School, Boston, MA, USA]) by Gateway cloning. To monitor CMB dynamics in S2 cells, the coding sequence of *Cmb*-RA was cloned into pMt-V5-6xHisB by Gibson cloning. Clonal cell lines expressing GFP/biotin ligase fusions were generated using TransIT transfection reagent (Mirus) by cotransfection of 0.6 µg/ml expression plasmid with 0.06 µg/ml pCoBlast selection plasmid (Invitrogen). Cells were grown for 3–5 days before selection with 25 µg/ml blasticidin. GFP/biotin ligase expression was induced by the addition of 500/100 µM of CuSO4 for 24 h before imaging/harvesting. Puromycin labeling was carried out as described in Hao et al. (2021). Cells were treated with 10 µg/ml puromycin for different incubation times (30 s, 1, 3, 5, 10 min). Control cells were treated with DMSO for the same period of time or pretreated with 10 µg/ml cycloheximide for 30 min prior to the addition of puromycin. After treatment, cells were immediately fixed in –20°C methanol and processed for immunofluorescence as described below.

HeLa CCL-2 cervical carcinoma cells and their derivative, *PCNT* mutant cell line ODCL0025 (Watanabe et al., 2020), were a gift from Karen Oegema (UC San Diego, San Diego, CA, USA) and cultured in Dulbecco's Modified Eagle's Medium (DMEM) containing 10% FBS, 25 mM HEPES (Gibco), and nonessential amino acids (Gibco). Human osteosarcoma U2-OS HTB-96 cells are from the American Type Culture Collection and were cultured in DMEM containing 10% FBS, penicillin (100 U/ml), and streptomycin (0.1 mg/ml). Cells were grown at 37°C in 5% CO2 and passaged every 2–3 days. All cell lines were regularly tested for *Mycoplasma* contamination.

To monitor PCM1 dynamics in vertebrate cells, the coding sequence of PCM1 was cloned into pEGFP-N by Gibson cloning and introduced into HeLa cells by transient transfection using polyethylenimine (PEI, Polysciences) and cells were analyzed 48 h after transfection. RNAi-mediated depletion of PCM1 was performed by siRNA using a mixture of two oligonucleotides (Dammermann and Merdes, 2002; see Table 1), purchased from Sigma-Aldrich and introduced into cells by Lipofectamine RNAiMAX-mediated transfection following the manufacturer's instructions. Phenotypes were assessed 48 h after transfection. To examine CMB localization and potential to rescue depletion of endogenous PCM1 in vertebrate cells, the coding sequence of Cmb including C-terminal GFP was subcloned into pcDNA3.1 for expression in mammalian cells. For colocalization analysis, transfection was performed using PEI as above and cells were analyzed 48 h after transfection. To examine cross-species rescue, HeLa cells were transfected with PCM1 siRNA using Lipofectamine RNAiMAX and again with Cmb-GFP using PEI 4 h later. Cells were fixed 48 h after siRNA transfection and stained using the immunofluorescence protocol optimized for staining with mouse PCM1 antibody described for single-molecule fluorescence in situ hybridization–immunofluorescence (smFISH-IF) below. To inhibit translation in vertebrate cells, cells were treated with 200 µg/ml cycloheximide for 20 min or 100 µg/ml puromycin for 30 min prior to fixation and staining.

## Identification of orthologs of centrosomal/ciliary genes across opisthokonts

PCM1 orthologs, as well as orthologs of centrosomal/ciliary genes across different phyla, were identified by reciprocal BLAST search (BLAST+ 2.13.0 [Camacho et al., 2009]) performed locally using the human protein as the starting point, with bidirectional best match at an E-value threshold cutoff of 0.1 as a simple but robust (Kristensen et al., 2011) method to infer orthology. Where direct comparisons failed to identify a clear ortholog, indirect searches were performed using less divergent related species as intermediates, as well as by applying hidden Markov models with HMMER (Eddy, 2011) using alignments of known representatives, constructed with MAFFT (Katoh et al., 2019). Conserved motifs within PCM1 were identified by Meme (https://meme-suite.org/meme/) and sequences aligned using MUSCLE within Jalview (https://www.jalview.org). The presence of centrosomes and cilia within fungi, Nematomorpha, and Platyhelminthes was inferred from the conservation of core centriolar (STIL/ANA2, SASS6/SAS-6, CENPJ/SAS-4, CEP135/BLD10), centrosomal (CDK5RAP2/CNN, CEP192/SPD-2), and ciliary (distal appendage, transition zone, IFT and BBS components, inner and outer dynein arm components, dynein assembly factors, nexins, N-DRC, radial spoke, and central apparatus components [Dobbelaere et al., 2023]) proteins, as well as literature reports (Azimzadeh et al., 2012; Grell and Benwitz, 1981).

### *Drosophila* behavioral assays

To examine fly coordination, 10 3-day-old adult males were collected in 8-cm graduated, flat-bottom tubes. Flies were left for 30 min to recover from the anesthetic and acclimatize, then banged to the bottom of the tube, and filmed climbing back upward. Videos were analyzed to establish the time at which all flies had crossed the halfway mark. The experiment was repeated three times per condition.

To assess male fertility, single 3-day-old males were crossed with four control virgin females in standard culture vials. After mating for 1 h, males were removed and females kept in the vial for an additional 24 h, and the number of offspring was counted prior to hatching. The experiment was repeated three times with 10 males per condition.

To assess embryonic viability, two virgin females and one male were put into cages with apple juice plates and allowed to lay eggs for 24 h at 25°C. Agar plates were transferred into a

humid chamber, and the number of hatched and unhatched embryos was determined after 24 and 48 h.

## *Drosophila* immunofluorescence staining

Rabbit polyclonal antibodies against the N terminus (amino acids 1–150) of *Drosophila* CMB were raised using a GST fusion as antigen and purified over the untagged cleaved antigen as described previously (Oegema et al., 2001). The following primary antibodies were used for immunofluorescence in *Drosophila*: mouse anti-puromycin (clone 12D10; Merck Millipore) at 1:200, rabbit anti-CMB antibody (this study) at 1:300, mouse anti-NompC (Liang et al., 2011), rabbit anti-ANA1 (Conduit et al., 2010), rabbit anti-CEP97 (Dobbelaere et al., 2020), and rabbit anti-SAS-4 (Basto et al., 2006), all at 1:500, rabbit anti-CNN (Dobbelaere et al., 2008) at 1:600, rabbit anti-CP110 (Franz et al., 2013) at 1:700, rabbit anti-SPD-2 (Cabral et al., 2019) at 1:800, rabbit anti-ANA-2 (Cabral et al., 2019) at 1:900, mouse anti-γ-tubulin (GTU88; Sigma-Aldrich) at 1:1,000. Secondary antibodies (Jackson ImmunoResearch) were used at 1:100.

For immunofluorescence staining in S2 cells, cells seeded on concanavalin A–coated coverslips were fixed in 4% formaldehyde for 15 min at room temperature (or –20°C methanol for 20 min in the case of puromycin labeling), permeabilized with PBS/0.2% Tween-20 for 15 min, and blocked for 20 min in 5% BSA in PBS/0.1% Triton X-100. Cells were incubated with primary antibodies in blocking solution for 1 h, washed three times for 5 min with PBS/0.1% Triton X-100, and incubated with secondary antibodies in blocking solution for 1 h. Following staining with 1 μg/ml Hoechst in PBS for 5 min, cells were washed three times for 5 min with PBS/0.1% Triton X-100 and once with PBS before being mounted in Vectashield.

For immunofluorescence staining in testes, adult males were dissected in ice-cold PBS, and the testes were transferred to a microscope slide, covered with a coverslip, and flash-frozen in liquid nitrogen. After recovering slides and removing the coverslip, samples were incubated for 5 min in –20°C methanol and 1–2 min in –20°C acetone. Samples were then rinsed with PBS, rehydrated in PBS/1% Triton X-100 for 10 min, and washed two times with PBS for 5 min, followed by blocking with 1% BSA in PBS for 45 min. Primary antibodies in blocking solution were then added and samples incubated overnight at 4°C. The next day, samples were washed two times with PBS for 5 min before incubation with secondary antibodies in blocking solution for 1 h. Finally, after washing twice with PBS for 5 min, samples were stained with 1 μg/ml Hoechst in PBS for 5 min and mounted in Vectashield.

For immunofluorescence staining of chordotonal neurons, leg chordotonal organs were dissected out of 36-h-old male pupae, placed between a microscope slide and coverslip, and gently pressed to remove them from the cuticle. Fixation and staining were performed as described for testes above.

For immunofluorescence staining of neuroblasts, brains from third-instar larvae were dissected in PBS and placed in 4% paraformaldehyde in PBS for 20 min, treated with 45% acetic acid for 15 s, followed by 60% acetic acid for 3 min, then covered with a coverslip, and squashed between coverslip and slide before flash-freezing in liquid nitrogen. After recovering slides and

removing the coverslip, samples were fixed in methanol for 5–8 min at –20°C, washed four times for 15 min with PBS/0.1% Triton X-100, and incubated in primary antibody in PBS/0.1% Triton X-100 overnight at 4°C. The next day, brains were washed three times for 5 min with PBS/0.1% Triton X-100 and incubated with secondary antibody overnight at room temperature for 3–4 h. Finally, samples were washed three times for 15 min with PBS/0.1% Triton X-100, stained with 1 μg/ml Hoechst in PBS for 5 min, and mounted in Vectashield.

For embryo immunofluorescence, flies were put into cages on apple juice plates with yeast paste 3 days before embryo collection. Early embryos (0–2 h) were collected from plates, washed and quickly treated with 5% bleach to dechorionate them, and fixed in heptane/4% paraformaldehyde in PBS for 2 min. Heptane was replaced by methanol, and samples were incubated for 1 min. Embryos were rehydrated with PBS/0.1% Triton X-100 for 15-min rotating, and blocked two times with 5% BSA in PBS for 30 min before incubation with primary antibody in blocking buffer overnight at 4°C. The next day, embryos were washed three times with PBS/0.1% Triton X-100 for 20 min and incubated on the rotator with secondary antibodies in blocking buffer for 1 h. Finally, embryos were washed in PBST, stained with 1 μg/ml Hoechst in PBS for 5 min, washed again, and mounted in Vectashield.

## *Drosophila* whole-mount testis staining

For Hoechst/phalloidin whole-mount testis staining, adult males were dissected in PBS and testes transferred to Nunc MicroWell MiniTrays containing PBS and fixed in 4% paraformaldehyde in PBS for 25 min. Testes were washed three times with PBS/0.1% Triton X-100 for 5 min and stained with 5 μg/ml Hoechst and 1:50 phalloidin/Alexa Fluor 568 in PBS overnight at 4°C. The next day, testes were washed three times with PBS, carefully separated from the rest of the abdomen, and placed in a drop of Vectashield on a multiwell slide and covered with a coverslip.

## Immunofluorescence staining in vertebrate cells

The following primary antibodies were used for immunofluorescence in vertebrate cells: rabbit anti-hsPCM1 (Dammermann and Merdes, 2002) and goat anti-GFP (Dammermann et al., 2004), both 1 μg/ml, rabbit anti-MIB1 (Sigma-Aldrich), 1:100, rabbit anti-AZI1 (Abcam), 1:250, mouse anti-hsPCM1 (G-6; Santa Cruz Biotechnology), 1:300, rabbit anti-CDK5RAP2 (Merck Millipore), rabbit anti-CEP290 (Novus Biologicals), and rabbit anti-PCNT (Abcam), all 1:400, rabbit anti-OFD1 (Sigma-Aldrich), 1:500, and mouse anti-γ-tubulin (GTU88; Sigma-Aldrich), 1:1,000. Secondary antibodies (Jackson ImmunoResearch) were used at 1:100.

For immunostaining in vertebrate cells, cells were fixed in –20°C methanol for 20 min, then washed, and rehydrated in PBS two times for 5 min. Permeabilization was done in PBS + 0.1% Tween-20 for 5 min, followed by blocking in 0.5% BSA in PBS/Tween-20 for 5 min. Cells were subsequently incubated with primary antibodies in blocking buffer for 40 min, and washed two times for 5 min with PBS and once with PBS/Tween-20. Cells were then incubated with secondary antibodies for 20 min and stained with 1 μg/ml Hoechst in PBS for 5 min. After

2 further washes with PBS (5 min each) and once with PBS/Tween-20, coverslips were mounted on slides with Vectashield. For immunofluorescence experiments including the mouse anti-hsPCM1 (G-6; Santa Cruz Biotechnology) antibody, we followed the protocol described in Chen et al. (2022).

## Puromycylation proximity ligation assay (Puro-PLA) in vertebrate cells

HeLa cells were seeded on coverslips and transfected with PCM1-GFP 48 h prior to the experiment. Cells were incubated with 2 µM puromycin in complete medium for 5 min (or untreated for controls) at 37°C, washed once with ice-cold PBS, and fixed with 4% wt/vol paraformaldehyde in PBS. Coverslips were washed two times with PBS and once with PBS + 0.1% 0.1% Triton X-100. Puro-PLA was performed according to the manufacturer's instructions (Duolink PLA protocol; Sigma-Aldrich). Briefly, cells were blocked with Duolink Blocking Solution for 60 min at 37°C in a humid chamber. Primary antibodies were diluted in Duolink Antibody Diluent (anti-puromycin 1:500, anti-PCNT 1:3,000) for 1 h at room temperature. Cells were washed twice with wash buffer A (2 × 5 min) at room temperature and incubated with Duolink In Situ PLA Probe Anti-Mouse PLUS and Anti-Rabbit MINUS probes diluted 1:5 in Duolink Antibody Diluent for 1 h at 37°C. After washing with wash buffer A (2 × 5 min), the Duolink In Situ Detection Reagents Red kit was used and samples were ligated using DNA ligase mixed in 1× ligation buffer for 30 min at 37°C. Next, cells were washed twice with wash buffer A and incubated with DNA polymerase in 1× amplification buffer (1:80 dilution) for 100 min at 37°C before washing twice with wash buffer B for 5 min and once with 0.01× wash buffer B for 1 min. Finally, cells were stained with goat anti-GFP antibody following the immunofluorescence protocol described above.

## Sequential single molecule fluorescence in situ hybridization and immunofluorescence staining (smFISH-IF) in vertebrate cells

Fluorescence in situ hybridization probes against the coding sequence of *CEP290* (transcript variant 1, NCBI accession no. NM_025114) and *PCNT* mRNA (transcript variant 1, NCBI accession no. NM_006031) were designed using Stellaris Probe Designer (https://www.biosearchtech.com/stellarisdesigner) and ordered from Biosearch Technologies conjugated with Quasar 570. Sequences can be found in Table 1. Probes were resuspended in TE buffer at 25 mM. The protocol for sequential smFISH-IF was adapted from Bayer et al. (2015), Trcek et al. (2017). Cells were fixed in 4% paraformaldehyde for 15 min, washed three times for 10 min with PBS/0.05% Triton X-100, and then incubated in warmed prehybridization buffer/FISH wash buffer (10% formamide, 2× SSC) at 37°C for 20 min before incubation in hybridization buffer (10% formamide, 10 mg/ml sheared salmon sperm DNA, 10% dextran sulfate, 10 mM vanadyl ribonucleoside complexes solution) containing 300 nM smFISH probe overnight at 37°C. The next day, cells were permeabilized and blocked with 1% BSA in 2× SSC, 1% Triton X-100 for 2 h (solution exchanged twice during incubation). Cells were then washed two times with 2× SSC and incubated with primary

antibody in 2× SSC and 1% Triton X-100 for 1 h, washed three times with IF wash buffer (2× SSC, 0.5% Triton X-100 in ddH2O), and incubated with secondary antibody solution in 2× SSC for 2–4 h at room temperature. Finally, cells were washed three times for 10 min with IF wash buffer, stained with 1 µg/ml Hoechst in PBS for 5 min, and mounted in Vectashield. In the case of *Drosophila* S2 cells, smFISH-IF was performed as described for vertebrate cells with the following modifications: cells were permeabilized and blocked with 1% RNase-free BSA in 2× SSC, 1% Triton X-100. Incubations with primary and secondary antibodies were performed in 2× SSC, 1% Triton X-100 containing 0.1% RNase-free BSA.

## Single-molecule inexpensive fluorescence in situ hybridization (smiFISH)

Primary unlabeled FISH probes against the coding sequence of human *RANBP10* (transcript variant 1, NCBI accession no. NM_020850) and *Drosophila Sas-4* mRNA (NCBI accession no. NM_141444) were designed using Stellaris Probe designer and ordered with FLAP-X sequence (5′-CCTCCTAAGTTTCGAGCTGGACTCAGTG-3′) (Tsanov et al., 2016) from Sigma-Aldrich. Sequences can be found in Table 1. Secondary probes containing the reverse complement of the FLAP-X sequence (5′-CACTGAGTCCAGCTCGAAACTTAGGAGG-3′) were ordered from Biosearch Technologies conjugated with Quasar-570 at both 5′ and 3′ ends. FLAP hybridization was performed according to Tsanov et al. (2016) and probes used for smFISH-IF as described above.

## Fixed imaging

For imaging of fixed samples (except whole-mount testis staining), 0.2-µm 3D wide-field datasets were acquired using either a 60× 1.42NA UPlanX Apochromat or a 100× 1.4NA UPlanS Apochromat lens on DeltaVision Ultra Epifluorescence Microscope equipped with a 4-megapixel sCMOS camera and 7-Color SSI module, computationally deconvolved using SoftWorx, and imported into Fiji for postacquisition processing. Exposure settings were held constant across conditions.

For whole-mount testis staining, samples were imaged on a Zeiss LSM 700 confocal laser scanning microscope equipped with 40× 1.3NA Plan Apochromat oil immersion objective. Image stacks of ~20 z-planes were acquired at 4-µm increments for each of the six testes. Maximum projections from 4 z-stacks were stitched together using ZEN software (Zeiss) to cover the whole testis.

## Live-cell imaging

For live-cell imaging of *Drosophila* syncytial-stage embryos, 0- to 2-h embryos were collected on apple juice agar plates, dechorionated on double-sided tape, immobilized on glass-bottom dishes coated with Scotch tape adhesive dissolved in heptane, and mounted with Voltalef oil. Embryos were imaged on a Yokogawa CSU-X1 spinning disk confocal mounted on a Zeiss Axio Observer Z1 inverted microscope equipped with a 63× 1.4NA Plan Apochromat lens, 100-mW 488-nm and 200-mW 561-nm solid-state lasers, and a Hamamatsu ImageEM X2 EM-CCD camera and controlled by VisiView software (Visitron Systems). 14 × 0.75 µm GFP/mCherry z-series were acquired at 15-s

intervals, with manual focus adjustment between intervals if needed, using low laser illumination to minimize photobleaching. Image stacks were imported into Fiji for postacquisition processing.

For live-cell imaging in S2 cells, GFP expression was induced with $CuSO_4$ 24 h before imaging. On the day of imaging, cells were plated on glass-bottom dishes and allowed to settle for at least 30 min. Imaging was performed on a Yokogawa CSU X1 spinning disk confocal mounted on a Zeiss Axio Observer Z1 inverted microscope equipped with a 100× 1.3NA EC Plan-Neofluar lens, 100-mW 488-nm solid-state laser, and an Evolve EM512 EMCCD camera (Photometrics) and controlled by VisiView software (Visitron Systems). 0.5-µm GFP z-series were acquired at 5-s intervals, using low laser illumination to minimize photobleaching. Image stacks were imported into Fiji for postacquisition processing.

For live-cell imaging of sperm motility, seminal vesicles of 3-day-old males were dissected in PBS and immediately transferred into a drop of Schneider medium on a microscope slide. The seminal vesicle was pierced with a sharp tungsten needle, mineral oil used to encircle the drop, a coverslip placed on top, and sperm motility imaged immediately using dark-field settings on a Zeiss Axio Imager Z2 microscope equipped with a 63× 1.4NA Plan Apochromat lens at 12 frames/s. Image sequences were imported into Fiji for postacquisition processing.

**Quantification of centrosome intensity in the early embryo**

Analysis of centrosome intensity in the early embryo was done as described in Alvarez-Rodrigo et al. (2019) with modifications described hereafter. Maximum intensity projections of z-stacks were generated and image sequences bleach-corrected using the ImageJ plugin "Bleach correction" (Miura, 2020) by applying the exponential fitting method. Centrosome intensity (RFP:CNN channel) was analyzed at nuclear envelope breakdown. A fixed-sized ROI (4.88 × 4.88 µm) was placed manually around the center of each centrosome and the mean intensity of this centrosome ROI computed using Fiji. For background quantifications, we analyzed mean intensities within the same-sized ROI placed in three cytoplasmic regions close to the centrosome, calculated the average, and subtracted this value from the centrosome mean intensity. For each embryo, all centrosomes that were fully captured within the z-stack were used for quantification and included in the statistical test. The final average mean intensity per embryo was calculated and plotted in GraphPad Prism.

**Colocalization analysis**

To assess colocalization of CMB/PCM1 with proteins of interest, the Fiji colocalization plugin JaCoP (Bolte and Cordelières, 2006) was used for both image channels. For this, a cytoplasmic region 3.5 × 3.5 µm in size was chosen at random away from the nucleus and centrosomes. In the case of *Drosophila* S2 cells, care was taken to avoid any of supernumerary centrosomes present in >50% of cells (Kwon et al., 2008). To assess colocalization at the centrosome, the ROI for both channels was positioned such that the centrosome was located not in the center but at one extreme of the ROI. The 3D

z-stack was cropped and segmented using the built-in threshold option. To exclude random colocalization, one of the channels was rotated by 90°. Pearson's correlation coefficient was calculated and used as an indicator of the degree of overlap. Any difference between experimental and randomized conditions was assessed by statistical testing.

**Quantification of the number of foci**

For quantification of the number of satellite or mRNA foci, maximum projections were generated in Fiji. Pixel-based segmentation of foci was carried out using Ilastik with the simple segmentation method. A custom-made Fiji macro was used to convert the background and signal image classes into 0 and 1 s, respectively. The segmented images were analyzed in MATLAB with a custom-made script (https://github.com/jugarbau/Pachinger-et-al.-2025). In brief, the maximum intensity projections of DAPI, centrosome, and foci channels were combined into a single image, with the user given the opportunity to adjust brightness and contrast to make the outline of the cells visible and the outline of individual cells manually delineated using the CROIEditor.m function (https://github.com/aether-lab/prana/blob/master/CROIEditor.m). Segmented images were postprocessed using the watershed algorithm and the area and intensity of foci quantified using regionprops. The cell background intensity was calculated, and results for intensities, the number of foci per cell, and foci size were imported into a.csv table and analyzed in Excel.

**Proximity analysis for mRNA and protein**

To assess proximity of mRNA and protein, the object-based Fiji plugin DiAna (Gilles et al., 2017) was used. While DiAna is capable of proximity measurements in 3D, a 2D measurement was chosen in order to take advantage of the "shuffle" function to test for nonrandom proximity. Deconvolved, single-plane images for the two respective channels (mRNA and protein) were loaded into Fiji and segmented using global thresholding and Gaussian filtering implemented in the plugin. Distance to the first nearest object (center–center distance) was calculated. To assess statistical significance, distances between pairs of proximal particles were compared with their counterparts in randomized images. For this purpose, a mask was applied using Fiji "create mask" to manually outline each cell (excluding the nucleus, mitotic, and apoptotic cells). In randomized conditions, the shuffle function was used to randomly distribute objects within the mask created for each cell and distances were calculated as described before.

**Quantification of satellite signal around centrosomes**

To determine the subcellular distribution of PCM1 and CMB relative to the centrosome, radial profiles were quantified. First, maximum intensity projections were created and thresholded, and the center of mass of each centrosome based on γ-tubulin signal was identified using the "analyze particles" function in Fiji. A circular ROI was placed around this point and the plugin Radial Profile Plot (https://imagej.net/ij/plugins/radial-profile.html) used to obtain a radial intensity profile of satellite foci

centered around each centrosome. Concentric rings and corresponding integrated intensities were quantified within 5 µm of the centrosome. For background quantifications, three smaller sized circular ROIs in distant cytoplasmic regions without foci were used to subtract from the mean intensity value. Intensities were then normalized to the central ring, and all intensities were combined, averaged, and mirrored to plot a single symmetric radial profile around the centrosome.

### Single-particle tracking and analysis of satellite dynamics

Time-lapse recordings of CMB foci were first exported and processed using Fiji to create maximum intensity projections. Segmentation of CMB foci was performed using Ilastik (v1.4.0-OX) (Berg et al., 2019) and the simple segmentation method (Geisler et al., 2023). Particles exceeding 700 nm in diameter or exceeding a mean fluorescence signal intensity twofold higher than average were excluded from analysis to avoid artifacts of overexpression. Single-particle tracking of CMB-GFP satellites in S2 cells was performed using TrackMate (v7.11.1) (Tinevez et al., 2017) with the following settings: Ilastik detector, simple linear assignment problem (LAP) tracker, maximum linking distance of 5 µm, maximum gap-closing distance of 5 µm, and maximum gap closing of two frames. The resulting spot and track features were exported and further analyzed in Excel. Instantaneous speed, average speed, and mean squared displacement were computed from coordinates of satellites from trajectory analysis. Tracks were further characterized using TraJClassifier (v0.8.1) (Wagner et al., 2017), a Fiji plugin that classifies trajectories into normal diffusion, subdiffusion, confined diffusion, and directed/active motion by a random forest approach. Parameters used were as follows: minimum track length of 11 frames, window size (positions) of 10, segment length of 10, resample rate and pixel size 0, and frame rate of 0.2 frames per second.

### Drosophila transmission electron microscopy

Transmission electron microscopy was carried out as previously described in Dobbelaere et al. (2020). For EM of chordotonal organs, legs from 36-h-old pupae were cut off with microscissors and fixed in 2% glutaraldehyde and 2% paraformaldehyde in 0.1 mol/l sodium phosphate buffer (pH 7.2) in a desiccator for 2 h at room temperature and then overnight on a rotator at 4°C. The next day, legs were rinsed with sodium phosphate buffer, postfixed in 2% osmium tetroxide in buffer on ice, dehydrated in graded series of acetone on ice, and subsequently embedded in Agar 100 resin. 70-nm sections were cut and poststained with 2% uranyl acetate and Reynolds' lead citrate. Sections were examined on a Morgagni 268D microscope (FEI) operated at 80 kV. Images were acquired with an 11-megapixel Morada CCD camera (Olympus-SIS).

For EM of testes, late pupal testes from third-instar larvae were dissected in PBS and fixed using 2.5% glutaraldehyde in 0.1 mol/l sodium phosphate buffer, pH 7.2, for 1 h at room temperature. Samples were then rinsed with sodium phosphate buffer, postfixed in 2% osmium tetroxide in ddH2O on ice, dehydrated in a graded series of acetone, and embedded in Agar 100 resin. 70-nm sections were then cut, processed, and imaged as described above.

### BioID and TurboID in Drosophila S2 cells

BioID and TurboID pulldown was performed as detailed in Roux et al. (2012), with minor modifications. Cells expressing pMt-myc-BirA*, pMt-SAS-4-myc-BirA*, pMt-V5-TurboID-GW, or pMt-V5-TurboID-CMB were incubated overnight with 50 µm biotin, washed three times with PBS, and lysed in ELB+ buffer (150 mM NaCl, 50 mM HEPES, pH 7.5, 5 mM EDTA, 0.3% NP-40, 6% glycerol) supplemented with Roche cOmplete Mini EDTA-free Protease Inhibitor Cocktail (1 tablet/10 ml lysis buffer), 1 mM PMSF, and 1 mM benzamidine. Lysates were clarified by brief centrifugation at 200 $g$ for 3 min at 4°C in a benchtop centrifuge before pelleting insoluble cellular material including centrosomes by centrifugation at 20,000 $g$ for 30 min at 4°C. To perform TurboID on the detergent-soluble cytoplasmic fraction, the supernatant was used directly for pulldown. For the detergent-insoluble cytoskeletal fraction, pellets were resuspended in 2% SDS, 1% β-mercaptoethanol in PBS and boiled for 30 min at 95°C with intermittent vortexing. Subsequently, the SDS concentration was reduced to 0.2% by dilution with PBS and Triton X-100 added to a final concentration of 2%. Samples were then sonicated by tip sonication (3 × 30 s pulses using a Bandelin Sonopuls GM70 sonicator at 60% continuous output, with brief cooling on ice between pulses), the SDS concentration was reduced to 0.1% SDS with PBS, and samples were sonicated once more for 30 s at 60% output before centrifuging at 20,000 $g$ for 30 min at 4°C and recovering the supernatant. Pierce Streptavidin-coated magnetic beads were equilibrated with PBS containing 0.1% SDS before incubating with the detergent-soluble cytoplasmic fraction or solubilized cytoskeletal fraction on a rotator overnight at 4°C. For CMB TurboID samples, beads were further acetylated to reduce background (Hollenstein et al., 2023) by incubation with a mixture of 190 µl 50 mM HEPES-NaOH, pH 7.8, containing 0.2% Tween-20 and 10 µl 100 mM Pierce Sulfo-NHS-Acetate for 1 h at room temperature for 1 h prior to incubation with lysate. Following the overnight incubation, unbound lysate was removed and beads were washed for 8 min each on a rotator at room temperature, first with 2% SDS in ddH2O, then 0.1% deoxycholic acid, 1% Triton X-100, 1 mM EDTA, 500 mM NaCl, and 50 mM HEPES, pH 7.5, and finally 0.5% deoxycholic acid, 0.5% NP-40, 1 mM EDTA, 500 mM LiCl, and 10 mM Tris, pH 7.5. Beads were then washed five times for 3 min with 50 mM Tris, pH 7.4, before being sent for on-bead protein digestion and mass spectrometry analysis.

### Nanobody–TurboID in Drosophila testes

Lysates were prepared as previously described for C. elegans in Holzer et al. (2022). Briefly, adult flies were washed three times with PBS, recovering flies by centrifugation for 3 min at 300 $g$, quickly ground in RIPA buffer (1% Triton X-100, 1 mM EDTA, 0.5% sodium deoxycholate, 0.1% SDS, 150 mM NaCl, and 50 mM Tris-HCl, pH 7.4) supplemented with Roche cOmplete Mini EDTA-free Protease Inhibitor Cocktail (1 tablet/10 ml lysis buffer), 1 mM PMSF, and 1 mM benzamidine using a mortar and pestle, and drop-frozen in liquid nitrogen. Frozen fly "popcorn" was further ground by cryogenic milling using a SPEX 6875 cryogenic mill (5 cycles, 1 min precool, 2 min run-time, 1 min cool-time; 12 cps). Lysates were thawed, and SDS and DTT were

added to a final concentration of 1% and 10 mM, respectively, boiled at 95°C for 5 min, and sonicated by tip sonication (two times for 1 min at 20% continuous output, with brief cooling on ice between pulses). Urea solution (8 M urea, 1% SDS, 50 mM Tris-Cl, 150 mM NaCl) was added to a final concentration of 2 M urea, and lysates were centrifuged at 100,000 $g$ for 45 min at 22°C. Lysates were desalted over Zeba spin desalting columns (7K MWCO; Thermo Fisher Scientific) to remove free biotin and incubated with Streptavidin magnetic beads by incubating on a rotator overnight. Following incubation, beads were washed twice with 150 mM NaCl, 1 mM EDTA, 2% SDS, 50 mM Tris-HCl, pH 7.4, once with 1× TBS buffer, pH 7.4, twice with 1 M KCl, 1 mM EDTA, 50 mM Tris-HCl, 0.1% Tween-20, pH 7.4, twice with 0.1 M Na$_2$CO$_3$, 0.1% Tween-20, pH 11.5, twice with 2 M urea, 10 mM Tris-HCl, 0.1% Tween-20, pH 8.0, and finally five times with 1× TBS buffer. Beads were finally resuspended in 1X TBS buffer and sent for on-bead protein digestion and mass spectrometry analysis.

## Sample preparation for mass spectrometry analysis

Beads were resuspended in 50 µl 1 M urea and 50 mM ammonium bicarbonate. Disulfide bonds were reduced with 2 µl of 250 mM DTT for 30 min at room temperature before adding 2 µl 500 mM iodoacetamide and incubating for 30 min at room temperature in the dark. Remaining iodoacetamide was quenched with 1 µl of 250 mM DTT for 10 min. Proteins were digested with 150 ng LysC (mass spectrometry grade, FUJIFILM Wako Chemicals) in 1.5 µl 50 mM ammonium bicarbonate at 25°C overnight. The supernatant without beads was digested with 150 ng trypsin (Trypsin Gold, Promega) in 1.5 µl 50 mM ammonium bicarbonate followed by incubation at 37°C for 5 h. The digest was stopped by the addition of trifluoroacetic acid to a final concentration of 0.5%, and the peptides were desalted using C18 StageTips (Rappsilber et al., 2007).

## Liquid chromatography separation coupled to mass spectrometry

Peptides were separated on an Ultimate 3000 RSLC nanoflow chromatography system (Thermo Fisher Scientific), using a precolumn for sample loading (Acclaim PepMap C18, 2 cm × 0.1 mm, 5 µm; Thermo Fisher Scientific), and a C18 analytical column (Acclaim PepMap C18, 50 cm × 0.75 mm, 2 µm; Thermo Fisher Scientific), applying a segmented linear gradient from 2 to 35% and finally 80% solvent B (80% acetonitrile, 0.1% formic acid; solvent A, 0.1% formic acid) at a flow rate of 230 nl/min over 120 min. The peptides eluted from the nano-liquid chromatography were analyzed by mass spectrometry as described below.

For SAS-4 S2 cell BioID, a Q Exactive HF Orbitrap mass spectrometer (Thermo Fisher Scientific) coupled to the column with a nanospray ion source using coated emitter tips (PepSep, MSWil) was used with the following settings: the mass spectrometer was operated in data-dependent acquisition (DDA) mode, and survey scans were obtained in a mass range of 380–1,650 m/z with lock mass activated, at a resolution of 120k at 200 m/z and an AGC target value of 3E6. The 10 most intense ions were selected with an isolation width of 2.0 m/z without

offset, and fragmented in the HCD cell at 27% collision energy, and the spectra were recorded for max. 250 ms at a target value of 1E5 and a resolution of 30k. Peptides with a charge of +2 to +6 were included for fragmentation, the peptide match and the exclude isotopes features were enabled, and selected precursors were dynamically excluded from repeated sampling for 30 s.

For CMB S2 cell TurboID, a Q Exactive HF-X Orbitrap mass spectrometer (Thermo Fisher Scientific) coupled to the column with a nanospray ion source using coated emitter tips (PepSep, MSWil) was used with the following settings: the mass spectrometer was operated in (DDA) mode, and survey scans were obtained in a mass range of 375–1,500 m/z with lock mass activated, at a resolution of 120k at 200 m/z and an AGC target value of 3E6. The eight most intense ions were selected with an isolation width of 1.6 m/z with offset 0.2 m/z, and fragmented in the HCD cell at 28% collision energy, and the spectra were recorded for max. 250 ms at a target value of 1E5 and a resolution of 30k. Peptides with a charge of +2 to +6 were included for fragmentation, the peptide match and the exclude isotopes features were enabled, and selected precursors were dynamically excluded from repeated sampling for 30 s.

For CMB testis TurboID, an Exploris 480 Orbitrap mass spectrometer (Thermo Fisher Scientific) coupled to the column with a FAIMS Pro ion source (Thermo Fisher Scientific) using coated emitter tips (PepSep, MSWil) was used with the following settings: the mass spectrometer was operated in DDA mode with two FAIMS compensation voltages (CV) set to –45 or –60 and 1.5-s cycle time per CV. The survey scans were obtained in a mass range of 350–1,500 m/z, at a resolution of 60k at 200 m/z and a normalized AGC target at 100%. The most intense ions were selected with an isolation width of 1.2 m/z and fragmented in the HCD cell at 28% collision energy, and the spectra were recorded for max. 100 ms at a normalized AGC target of 100% and a resolution of 15k. Peptides with a charge of +2 to +6 were included for fragmentation, the peptide match feature was set to preferred, the exclude isotope feature was enabled, and selected precursors were dynamically excluded from repeated sampling for 45 s.

## Mass spectrometry data analysis

The RAW MS data were analyzed with FragPipe (20.0), using MSFragger (3.8) (Kong et al., 2017), IonQuant (1.9.8) (Yu et al., 2021), and Philosopher (5.0.0) (da Veiga Leprevost et al., 2020). The default FragPipe workflow for label-free quantification (LFQ-MBR) was used, except "Normalize intensity across runs" was turned off. For SAS-4 BioID, the "MBR top runs" parameter was set to 1 to address batch measurements of sample replicates. Cleavage specificity was set to Trypsin/P, with two missed cleavages allowed. The protein FDR was set to 1%. A mass of 57.02146 (carbamidomethyl) was used as fixed cysteine modification; methionine oxidation and protein N-terminal acetylation were specified as variable modifications. MS2 spectra were searched against the *D. melanogaster* 1 protein per gene reference proteome from UniProt (Proteome ID: UP000000803, release 2023.03), concatenated with a database of 382 common laboratory contaminants (release 2023.03, https://github.com/maxperutzlabs-ms/perutz-ms-contaminants).

Computational analysis was performed using Python and the in-house–developed Python library MsReport (version 0.0.23). Only noncontaminant proteins identified with a minimum of two peptides and being quantified in at least two replicates of one experiment were considered for the analysis. LFQ protein intensities reported by FragPipe were $\log_2$-transformed and normalized across samples using the ModeNormalizer from MsReport. This method involves calculating $\log_2$ protein ratios for all pairs of samples and determining normalization factors based on the modes of all ratio distributions. For the data presented in the figures, missing values were imputed by deterministic lowest of detection after filtering out contaminants and proteins with less than two razor and unique peptides, an approach we previously found to yield superior results for centrosomal proteins that are frequently absent in one or all control samples but also not highly abundant in the experimental samples (Holzer et al., 2022). Also included in Table S2 is the more standard approach of imputation by drawing random values from a left-censored normal distribution modeled on the whole dataset (data mean shifted by –1.8 SD, width of distribution of 0.3 SD). Likely proximity interactors largely passed the significance threshold with both methods of imputation. These were defined as a $\log_2$ fold change of >1 and a P value in an unpaired $t$ test of <0.05. GraphPad Prism was used to prepare bar graphs and volcano plots using LFQ values imported from Microsoft Excel.

### Comparisons to published human datasets and Gene Ontology (GO) analysis

To compare the proteins significantly enriched in our CMB proximity interactome analyses in flies with those reported in published datasets of centrosomal, ciliary, and centriolar satellite proteins, as well as cytosolic mRNA-associated proteins in vertebrates (Gheiratmand et al., 2019; Gupta et al., 2015; Quarantotti et al., 2019; Youn et al., 2018), we first generated a proteome-wide matrix of all conserved proteins based on reciprocal BLAST analysis as described above, then performed a pairwise comparison between datasets using FlyBase IDs in *Drosophila* and Ensembl gene IDs in humans. Venn diagrams presented in the figures report both the number of conserved proteins common to both datasets and those unique to each, and the larger number of unique proteins including those proteins not detectably conserved across species.

GO analyses were performed using the PANTHER 18.0 (Mi et al., 2019) and GO database release 2023-10-09, comparing the protein IDs of the orthologs of proteins enriched in our CMB proximity interactome analyses against the human proteome, identifying GO annotation terms statistically enriched based on Fisher's exact test, and applying the Bonferroni correction for multiple testing. Annotation terms displayed in the figures are the top 8 terms for "cellular compartment" and "biological process" based on fold enrichment.

### Quantification and statistical analysis
#### Statistical analysis
Statistical analysis was performed in Excel and GraphPad Prism 10. Each dataset was tested for normal distribution using the D'Agostino–Pearson (omnibus K2) test. Where data were displaying a normal distribution, an unpaired, two-tailed Student's $t$ test with Welch's correction was used for comparison of two groups. When data failed the normality test for at least one of the datasets examined, an unpaired, two-tailed Mann–Whitney U test and a Kruskal–Wallis test with Dunn's multiple comparisons test were used for two groups or more than two groups, respectively. If $n$ was too small to test data for normal distribution, Student's $t$ test was used. For all experiments, the significance threshold was taken as $P < 0.05$. Significance levels are defined as follows: ****$P < 0.0001$, ***$P < 0.001$, **$P < 0.01$, *$P < 0.05$, NS, not significant. Error bars display mean with standard deviation unless otherwise stated.

### Online supplemental material
Fig. S1 shows results related to Fig. 1 (Identification of Combover as the *Drosophila* ortholog of PCM1). Fig. S2 shows results related to Fig. 2 (Further characterization of ciliary phenotypes in *Cmb* mutants). Fig. S3 shows results related to Fig. 3 (Further analysis of CMB localization). Fig. S4 shows results related to Fig. 4 (Further analysis of CMB proximity interactome). Fig. S5 shows results related to Fig. 5 (Further evidence for centriolar satellites as sites of translation in vertebrate cells). Table S1 shows orthologs of PCM1 and centrosomal/ciliary proteins across Opisthokonts, related to Fig. 1. **(A)** Orthologs of PCM1, CDK5RAP2, and CEP192 in representative species. Related to Fig. 1 B. **(B)** Orthologs of core centriolar (STIL/ANA2, SASS6/SAS-6, CENPJ/SAS-4, CEP135/BLD10) and ciliary proteins (distal appendage, transition zone, IFT and BBS components, inner and outer dynein arm components, dynein assembly factors, nexins, N-DRC, radial spoke, and central apparatus components [Dobbelaere et al., 2020, 2023]) in Nematomorpha, based on reciprocal BLAST analysis. Related to Figs. 1 B and S1 A. **(C)** GenBank accession numbers for PCM1 orthologs used to generate multiple sequence alignment in Fig. 1 C. Table S2 shows mass spectrometry data, related to Fig. 4. **(A–D)** Complete list of proteins identified by mass spectrometry in each BioID/TurboID run, including corresponding controls. **(A)** SAS-4 BioID S2 cells. **(B)** CMB TurboID S2 cells soluble/cytoplasmic fraction. **(C)** CMB TurboID S2 cells insoluble/cytoskeletal fraction. **(D)** CMB TurboID testes. Only the most relevant data rows and columns are displayed by default. The full processed MS data including excluded peptide groups, spectral counts, and differential abundance analysis using missing value imputation by random drawing from a left-censored normal distribution (ND) can be found by expanding the collapsed rows and columns. **(E)** Comparison of CMB TurboID proximity interactors with published datasets of centrosomal, ciliary and centriolar satellite proteins as well as cytosolic mRNA-associated proteins in vertebrates (Gupta et al., 2015; Youn et al., 2018; Gheiratmand et al., 2019; Quarantotti et al., 2019). Table S3 shows sm/smiFISH probe sequences, related to Table 1. Sequences of fluorescence in situ hybridization probes against the coding sequence of human *CEP290* (transcript variant 1, NCBI accession no. NM_025114), *PCNT* (transcript variant 1, NCBI accession no. NM_006031) and *RANBP10* mRNA (transcript variant 1, NCBI accession no. NM_020850), and *Drosophila Sas-4* (NCBI accession no. NM_141444).

## Data availability

Mass spectrometry data have been deposited to the ProteomeXchange Consortium via the PRIDE partner repository with the dataset identifier PXD049372. Any additional information required to reanalyze the data reported in this manuscript is available from the lead contact upon reasonable request.

## Acknowledgments

We thank members of the Dammermann and Campbell laboratories and Della David (Babraham Institute) for discussions; the Vienna *Drosophila* Resource Center, the Bloomington *Drosophila* Stock Center, Julius Brennecke and Jürgen Knoblich (IMBA), Andreas Jenny (Albert Einstein College of Medicine), Karen Oegema (University of California, San Diego), Norbert Perrimon (Harvard Medical School), and Jordan Raff (University of Oxford) and Helen White-Cooper (Cardiff University) for strains and reagents; and Balazs Erdi of the IMBA Fly House, Nicole Fellner of the VBCF Electron Microscopy Facility, Weiqiang Chen and Markus Hartl of the Max Perutz Labs Mass Spectrometry Facility, and Josef Gotzmann and Thomas Peterbauer of the Max Perutz Labs BioOptics Facility for technical assistance. Preliminary experiments for this project were performed by Astrid Steiner and Colette Emery.

This work was supported by funding from the Austrian Science Fund FWF, grants P34526-B and F8803-B, and the Austrian Research Promotion Agency FFG, grant 880579 to A. Dammermann, as well as a University of Vienna uni:doc PhD fellowship to C. Pachinger and a Max Perutz PhD fellowship of the Max Perutz Labs to J. Garcia-Baucells. Open Access funding provided by the University of Vienna.

Author contributions: C. Pachinger: conceptualization, funding acquisition, investigation, visualization, and writing—original draft, review, and editing. J. Dobbelaere: investigation, supervision, and writing—review and editing. C. Rumpf-Kienzl: investigation and writing—review and editing. S. Raina: investigation and writing—review and editing. J. Garcia-Baucells: funding acquisition, methodology, and writing—review and editing. M. Sarantseva: investigation and writing—review and editing. A. Brauneis: investigation and writing—review and editing. A. Dammermann: conceptualization, funding acquisition, project administration, supervision, visualization, and writing—original draft, review, and editing.

Disclosures: The authors declare no competing interests exist.

Submitted: 7 August 2024

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

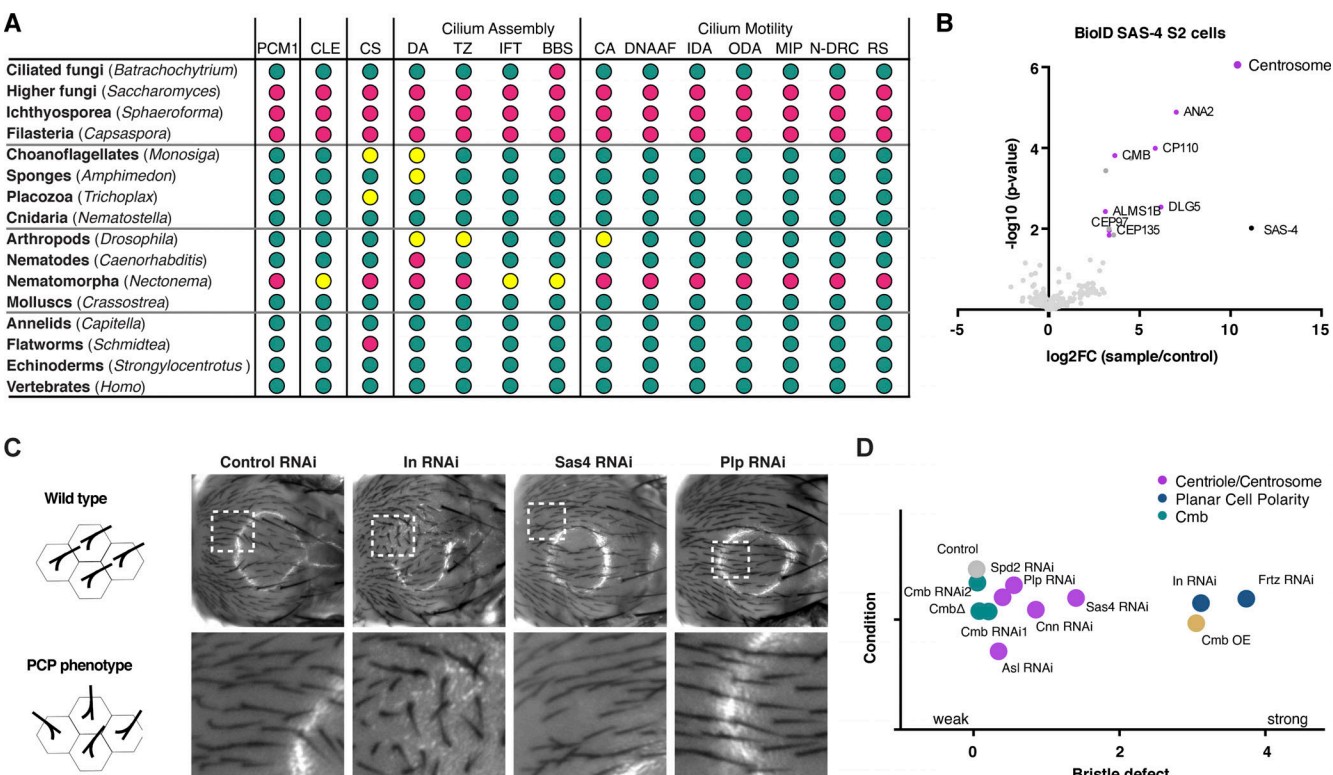

Figure S1. **Identification of Combover as the *Drosophila* ortholog of PCM1, related to** Fig. 1**. (A)** Related to Fig. 1 B. Conservation of PCM1, and core centriolar (STIL/ANA2, SASS6/SAS-6, CENPJ/SAS-4, CEP135/BLD10), centrosomal (CDK5RAP2/CNN, CEP192/SPD-2), and ciliary proteins (distal appendage, transition zone, IFT and BBS components, inner and outer dynein arm components, dynein assembly factors, nexins, N-DRC, radial spoke, and central apparatus components [Dobbelaere et al., 2023]) across opisthokonts, based on reciprocal BLAST analysis and hidden Markov model–based searches. Color code is green >2/3 of genes in indicated category present, yellow >1/3 of genes present, magenta <1/3 present. See also Table S1. **(B)** Results of LC-MS/MS analysis for direct BioID performed on the centriolar structural component SAS-4 in *Drosophila* S2 cells. Volcano plot of $-\log_{10}$ P values against $\log_2$ fold change (sample/control). Significantly enriched proteins ($\log_2$ enrichment >1, P <0.05) are indicated in dark gray, with centrosomal proteins highlighted in magenta. CMB was detected as a high-confidence interactor. See also Table S2 A. **(C)** Related to Fig. 1 D. Further characterization of PCP phenotypes in the fly notum. RNAi of PCP genes such as Inturned results in strong phenotypes, while centrosomal genes (*Sas-4* and *Plp*) show no or weak phenotypes. **(D)** Quantitation of bristle defects for selected genes. Phenotypes were scored on a scale from 0 (no phenotype) to 4 (strong phenotype), with values shifted slightly to avoid overlap. N = 10 flies per condition.

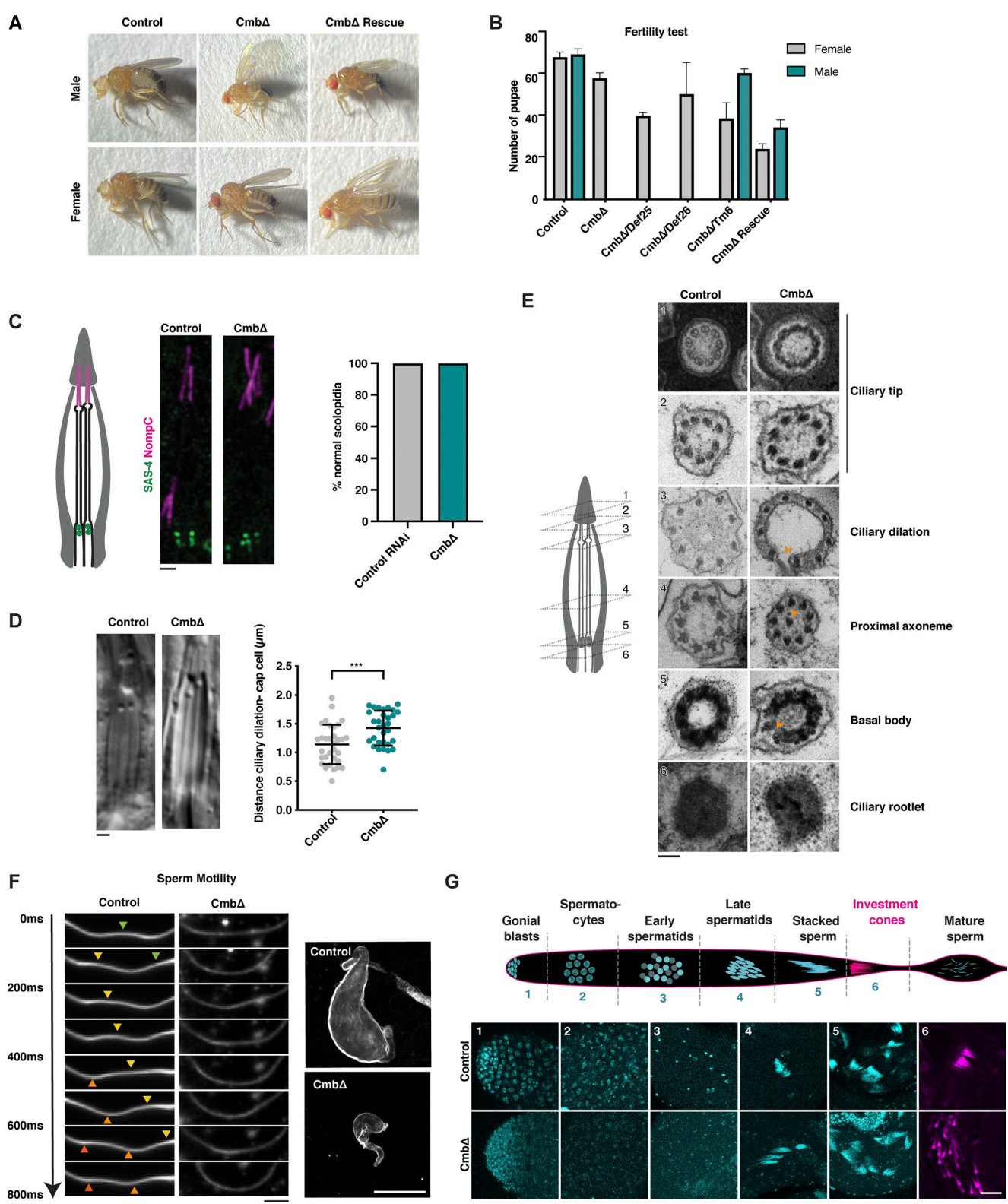

Figure S2. **Further characterization of ciliary phenotypes in _Cmb_ mutants, related to** Fig. 2. **(A)** Appearance of _Cmb_ mutant flies, as well as _Cmb_ mutants rescued by the expression of a GFP-CMB transgene. _Cmb_ mutants display abnormal wing posture, a phenotype associated with defective mechanosensation. **(B)** Fertility test performed on _Cmb_ mutant males and females, _Cmb_ mutants rescued by the expression of a GFP-CMB transgene or maintained over a balancer (Tm6), and _Cmb_ mutants placed over a deficiency that covers the _Cmb_ locus (Def 25, 26). _Cmb_ mutant males but not females exhibit fully penetrant sterility, a defect rescued by the expression of the GFP transgene. Placing the mutant over a deficiency does not impact fertility, excluding potential nonallelic effects. Error bars are the mean ± SD. _N_ = 3 single males, each crossed to four virgin females. **(C)** Schematic and immunofluorescence micrographs of scolopidia in chordotonal organ of the fly. SAS-4 and NompC were used to visualize basal body (green) and ciliary tip (magenta), respectively. Each scolopidium contains two

ciliated nerve endings ensheathed by a glial cell, with the ciliary tips attached to the cuticle via a cap cell (Kernan, 2007). No gross ciliary morphological defects are observed in *Cmb* mutants. *N* = 63 control scolopidia, 63 *Cmb* mutant. A statistical test is *t* test with Welch's correction. **(D)** DIC images of scolopidia. *Cmb* mutants display a larger distance between ciliary dilation and cap cell, indicative of ciliary positioning defects. Error bars are the mean ± SD. *N* = 32 control, 31 *Cmb* mutants. Student's *t* test with Welch's correction was used; ***P < 0.001. **(E)** Cross-sectional views of control and *Cmb* mutant scolopidia by transmission electron microscopy (TEM) from the distal tips (1) to the ciliary rootlets (6) below the basal body. Position was indicated by numbers in schematic on the left. *Cmb* mutants show minor structural defects, including broken axonemes and misplaced doublet microtubules (arrows). **(F)** Left: Analysis of flagellar movement of control and *Cmb* mutant sperm by high-speed video capture in dark-field microscopy. Sinusoidal motion can be seen in wild type. Arrowheads indicate the position of propagating peaks and troughs in image sequence. *Cmb* mutant sperm show severely compromised flagellar movement. Right: In contrast to controls, seminal vesicles of *Cmb* mutant flies are almost devoid of sperm, indicating defective movement of sperm to seminal vesicle. **(G)** Schematic and immunofluorescence images of *Drosophila* spermatogenesis. In *Cmb* mutants, the early stages of spermatogenesis appear superficially normal; however, in later stages, investment cones involved in individualizing sperm fail to form properly. Scale bars, 1 µm (C and D), 100 nm (E), 100 µm (F), 20 µm (G).

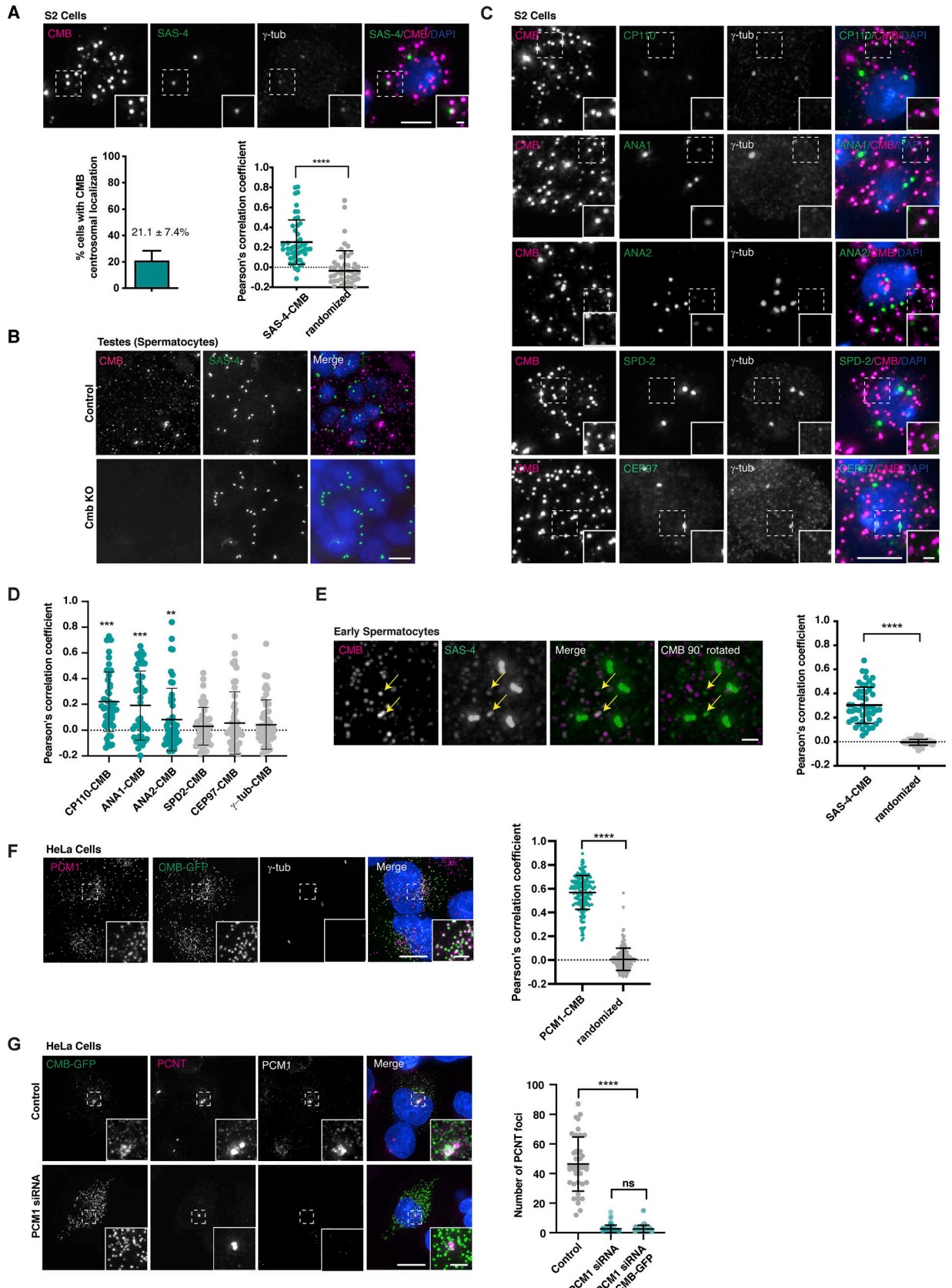

Figure S3.   **Further analysis of CMB localization, related to** Fig. 3. **(A)** Immunofluorescence micrograph showing CMB localizing to centrosomes in a subset of cells in *Drosophila* S2 cells. Quantitation of CMB localization reveals centrosome localization in ~20% of cells. Pearson's correlation coefficient analysis of centrosomal signal shows significant overlap between SAS-4 and CMB compared with randomized controls (single channel rotated by 90° with centrosome

positioned in the upper right quadrant of the square analyzed). Error bars are the mean ± SD. $N$ = 50 cells. Student's $t$ test was used; ****P < 0.0001. **(B)** Specificity of the polyclonal antibody raised against CMB confirmed by the absence of immunofluorescence signal in *Cmb* mutant testes. **(C)** Related to Fig. 3 I. Immunofluorescence micrographs showing some (CP110, ANA1, ANA2) but not all (SPD-2, γ-tubulin) centrosomal proteins colocalizing with CMB on cytoplasmic foci. **(D)** Pearson's correlation coefficient analysis of cytoplasmic protein colocalization with CMB assessed on images as in C. Error bars are the mean ± SD. $N$ = 50 cells per condition. A Mann–Whitney test was used to test statistical significance compared with randomized controls (single channel rotated 90°); ***P < 0.001, **P < 0.01. **(E)** Immunofluorescence micrographs showing SAS-4 colocalizing with Combover in the cytoplasm of primary spermatocytes in the testes. Colocalization was quantified by Pearson's correlation coefficient analysis. Error bars are the mean ± SD. $N$ = 12 animals. A $t$ test was used to assess statistical significance of colocalization compared with randomized controls (single channel rotated 90°); ****P < 0.0001. **(F)** CMB-GFP colocalizes with PCM1 when expressed in HeLa cells. Colocalization was quantified by Pearson's correlation coefficient analysis. Error bars are the mean ± SD. $N$ = 193 cells. A $t$ test was used to assess statistical significance of colocalization compared with randomized controls (single channel rotated 90°); ****P < 0.0001. **(G)** CMB-GFP expression fails to restore PCNT cytoplasmic centriolar satellite signal following depletion of endogenous PCM1 in HeLa cells. $N$ = 40 cells analyzed per condition. Mean ± SD are displayed. A Mann–Whitney test was used to assess statistical significance; ****P < 0.0001, NS, not significant. Scale bars, 5 µm (A, C, F, and G), 10 µm (B), 1 µm (E, A, C, and F, insets).

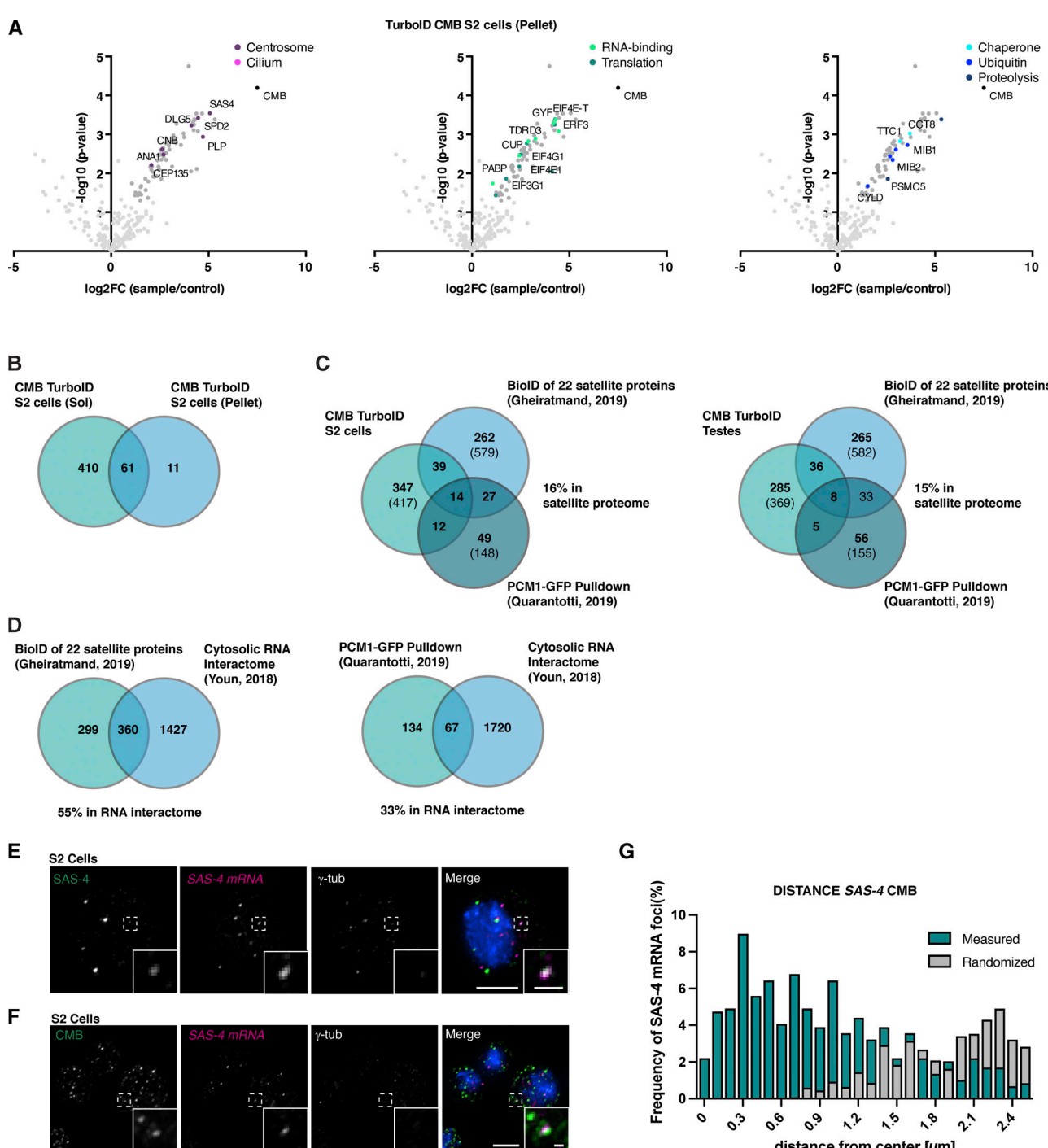

**Figure S4.    Further analysis of CMB proximity interactome, related to** Fig. 4**. (A)** Results of direct TurboID performed on CMB S2 cells (detergent-insoluble cytoskeletal fraction). LC-MS/MS analysis reveals centrosomal proteins (magenta), RNA-binding proteins (light green), and proteins involved in translation (dark green), chaperone-mediated protein folding (light blue), ubiquitination (blue), and proteolysis (dark blue). Volcano plots of $-\log_{10}$ P values against $\log_2$ fold change (sample/control). Significantly enriched proteins ($\log_2$ enrichment >1, P <0.05) are indicated in dark gray, with proteins of the above functional categories highlighted in color. See also Table S2 C. **(B)** Venn diagram revealing a significant overlap between CMB proximity interactome obtained from detergent-soluble (cytoplasmic) and detergent-insoluble (cytoskeletal) fractions of S2 cell extracts. See also Table S2 E. **(C)** Comparison of CMB S2 cell and testis TurboID interactomes with previous published datasets for centriolar satellites: the BioID of 22 satellite proteins mapped by Gheiratmand et al. (2019) and the PCM1-GFP pulldown performed by Quarantotti et al. (2019). Comparison of those proteins conserved between humans and flies. Numbers in parentheses are total number in each dataset. See also Table S2 E. **(D)** Comparison of BioID of 22 satellite proteins mapped by Gheiratmand et al. (2019) and PCM1-GFP pulldown performed by Quarantotti et al. (2019) with cytosolic RNA interactome defined by Youn et al. (2018). See also Table S2 F. **(E)** Single-molecule fluorescence hybridization (smFISH) combined with immunofluorescence microscopy shows *Sas-4* mRNA colocalizing with nascent SAS-4 protein in the cytoplasm of S2 cells. **(F and G)** smFISH combined with immunofluorescence microscopy in S2 cells. Immunofluorescence micrographs (F) and

corresponding quantitation (G). *Sas-4* mRNA localizes in the vicinity of CMB foci. To exclude random colocalization/proximity, distribution was compared with randomized controls. *N* = 109 cells. Scale bars, 5 µm (E), 10 µm (F), 1 µm (E and F, insets).

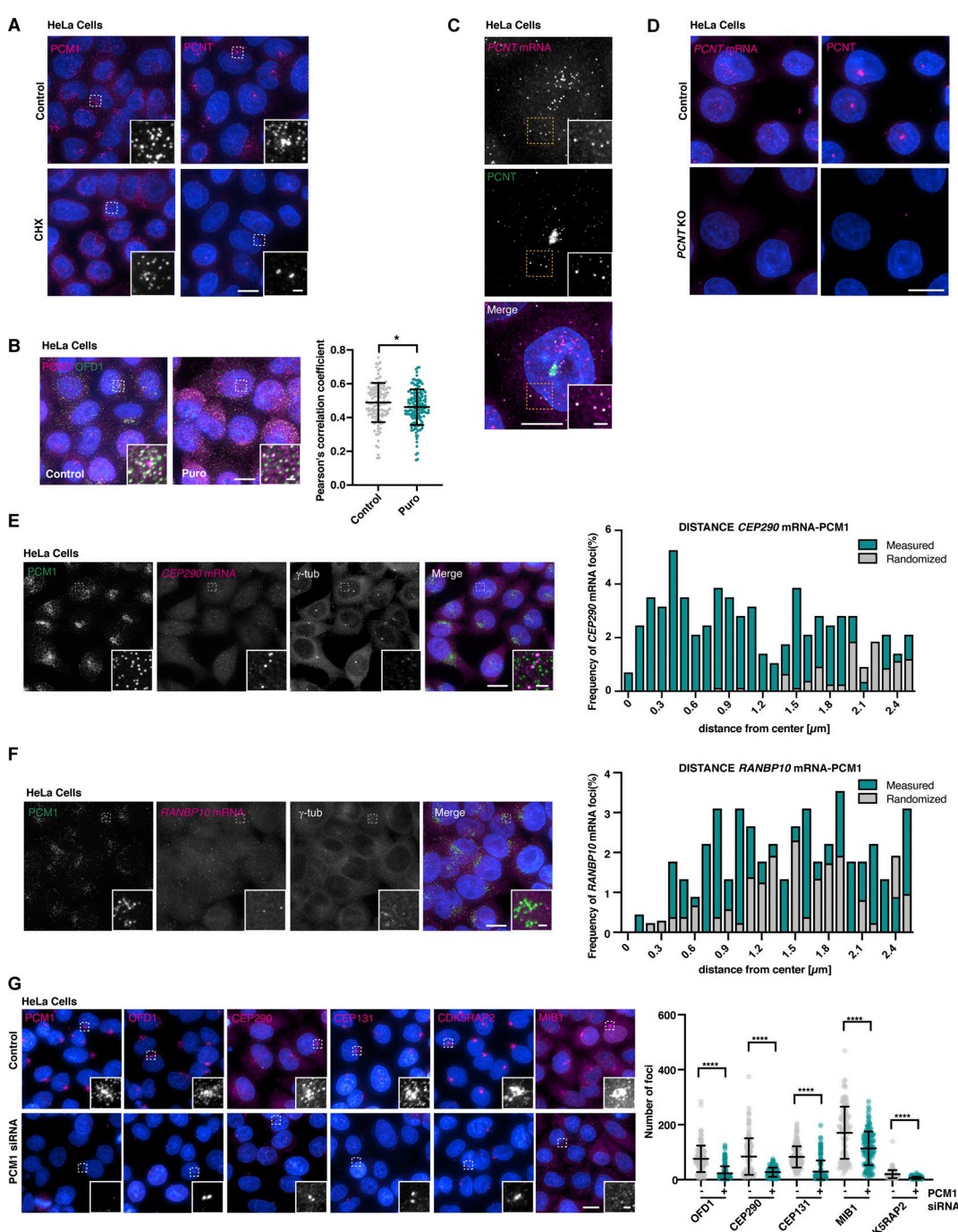

Figure S5.   **Further evidence for centriolar satellites as sites of translation in vertebrate cells, related to** Fig. 5. **(A)** Immunofluorescence micrographs of the centriolar satellite protein signal in control and cycloheximide-treated HeLa cells. The satellite client PCNT is significantly depleted of its cytoplasmic localization upon puromycin treatment, whereas its centrosome localization persists. In contrast, PCM1 signal remains, although it is now dispersed throughout the cytoplasm. **(B)** Immunofluorescence micrographs and quantitation of PCM1 and OFD1 distribution by Pearson's correlation coefficient analysis in control and puromycin-treated cells from Fig. 5 C. PCM1 and OFD1 colocalize independent of translation. N = 125 cells (control), 147 cells (puromycin treatment). Mean ± SD are indicated (Student's t test; *P < 0.05). **(C)** SmFISH combined with immunofluorescence microscopy shows PCNT mRNA colocalizing with the nascent PCNT protein in the cytoplasm. **(D)** Specificity of PCNT mRNA FISH and protein immunofluorescence signal confirmed by the absence of signal in PCNT KO cells. **(E)** Single-molecule fluorescence hybridization (smFISH) combined with immunofluorescence microscopy in HeLa cells. CEP290 mRNA localizes in the vicinity of PCM1. To exclude random colocalization/proximity, distribution was compared with randomized controls. N = 78 cells. **(F)** Single-molecule inexpensive fluorescence hybridization (smiFISH) combined with immunofluorescence microscopy in HeLa cells. Unlike PCNT and CEP290, RANBP10 mRNA does not localize proximal to PCM1. Distribution compared with randomized controls. N = 87 cells. **(G)** Immunofluorescence micrographs and quantitation of centriolar satellite signal in control and PCM1 siRNA–treated cells. Depletion of PCM1 largely eliminates cytoplasmic foci of OFD1, CEP290, CEP131, CDK5RAP2, while centrosomal signal remains. MIB1 signal is largely unaffected, although foci are now dispersed throughout the cytoplasm. Mean ± SD are indicated. N > 100 cells each condition. Statistical test to compare control and PCM1 depletions is t test with Welch's correction (MIB1), and nonparametric Mann–Whitney test (others); ****P < 0.0001. Scale bars, 10 µm (A, B, and D–G), 5 µm (C), 1 µm (insets).

Provided online are Table S1, Table S2, and Table S3. Table S1 shows orthologs of PCM1 and centrosomal/ciliary proteins across opisthokonts, related to Fig. 1. Table S2 shows mass spectrometry data, related to Fig. 4. Table S3 shows sm/smiFISH probe sequences, related to Table 1.

