## [Peer Review File · The Journal of Cell Biology]

A conserved role for centriolar satellites in translation of centrosomal and ciliary proteins

Claudia Pachinger, Jeroen Dobbelaere, Cornelia Rumpf-Kienzl, Shiviya Raina, Júlia Garcia-Baucells, Marina Sarantseva, Andrea Brauneis, and Alexander Dammermann

Corresponding Author(s): Alexander Dammermann, University of Vienna

Review Timeline:

Submission Date:	2024-08-07
Editorial Decision:	2024-09-10
Revision Received:	2025-01-30
Editorial Decision:	2025-02-28
Revision Received:	2025-03-12

Monitoring Editor: Monica Bettencourt-Dias

Scientific Editor: Tim Fessenden

Transaction Report:

DOI: <https://doi.org/10.1083/jcb.202408042>

September 10, 2024

Re: JCB manuscript #202408042

Dr. Alexander Dammermann
University of Vienna
Max Perutz Labs
Dr Bohr-Gasse 9
Vienna 1030
Austria

Dear Dr. Dammermann,

Thank you for submitting your manuscript entitled "A conserved role for centriolar satellites in translation of centrosomal and ciliary proteins". The manuscript was assessed by expert reviewers, whose comments are appended to this letter. We invite you to submit a revision if you can address the reviewers' key concerns, as outlined here.

You will see that reviewers commended the intriguing proposal that Cmb is an ortholog of PCM-1, with implications for centriolar satellites in flies. Reviewers 1 and 3 both sought greater details to distinguish between translation and transport mediated by Cmb, with validation that it regulates protein translation. Reviewers offer several suggestions for strengthening these observations from which to choose. A revision should also improve the colocalization studies as requested by Reviewer 3. If possible, testing if human PCM-1 can rescue loss of Cmb would provide important validation of Cmb's role. Finally, resolving the requests on the nuclear fallout phenotype from Reviewer 3 is optional in a revision.

GENERAL GUIDELINES:

Text limits: Character count for an Article is < 40,000, not including spaces. Count includes title page, abstract, introduction, results, discussion, and acknowledgments. Count does not include materials and methods, figure legends, references, tables, or supplemental legends.

Figures: Articles may have up to 10 main text figures. Figures must be prepared according to the policies outlined in our Instructions to Authors, under Data Presentation, <https://jcb.rupress.org/site/misc/ifora.xhtml>. All figures in accepted manuscripts will be screened prior to publication.

*****IMPORTANT:** It is JCB policy that if requested, original data images must be made available. Failure to provide original images upon request will result in unavoidable delays in publication. Please ensure that you have access to all original microscopy and blot data images before submitting your revision. ***

Supplemental information: There are strict limits on the allowable amount of supplemental data. Articles may have up to 5 supplemental figures. Up to 10 supplemental videos or flash animations are allowed. A summary of all supplemental material should appear at the end of the Materials and methods section.

Please note that JCB now requires authors to submit Source Data used to generate figures containing gels and Western blots with all revised manuscripts. This Source Data consists of fully uncropped and unprocessed images for each gel/blot displayed in the main and supplemental figures. Since your paper includes cropped gel and/or blot images, please be sure to provide one Source Data file for each figure that contains gels and/or blots along with your revised manuscript files. File names for Source Data figures should be alphanumeric without any spaces or special characters (i.e., SourceDataF#, where F# refers to the associated main figure number or SourceDataFS# for those associated with Supplementary figures). The lanes of the gels/blots should be labeled as they are in the associated figure, the place where cropping was applied should be marked (with a box), and molecular weight/size standards should be labeled wherever possible.

The typical timeframe for revisions is three to four months. While most universities and institutes have reopened labs and

allowed researchers to begin working at nearly pre-pandemic levels, we at JCB realize that the lingering effects of the COVID-19 pandemic may still be impacting some aspects of your work, including the acquisition of equipment and reagents. Therefore, if you anticipate any difficulties in meeting this aforementioned revision time limit, please contact us and we can work with you to find an appropriate time frame for resubmission. Please note that papers are generally considered through only one revision cycle, so any revised manuscript will likely be either accepted or rejected.

Thank you for this interesting contribution to Journal of Cell Biology. You can contact us at the journal office with any questions at cellbio@rockefeller.edu.

Sincerely,

Monica Bettencourt-Dias
Monitoring Editor
Journal of Cell Biology

Tim Fessenden
Scientific Editor
Journal of Cell Biology

Reviewer #1 (Comments to the Authors (Required)):

Pachinger et al. identify Comover (Cmb) as a protein interacting with *Drosophila* SAS-4 and characterize its role in ciliogenesis. The authors convincingly describe a role for Cmb in ciliary function and identify several known centrosome and ciliary proteins interacting with Cmb via TURBO-ID proximity labeling mass spectrometry. Some of these interactions were validated through colocalization analysis. Their comparative proteomics analyses will provide the field a valuable resource, while also characterizing the role of Cmb in ciliogenesis and mitotic fidelity.

Given a weak sequence homology, the authors suggest Cmb may be the *Drosophila* ortholog to PCM-1. To support this hypothesis, the authors point to the role for Cmb in ciliogenesis and highlight several shared protein interactors from their own Cmb dataset versus published PCM-1 proteomics. However, the evidence supporting the idea that Cmb is the functional ortholog of PCM-1 remains weak. For this reviewer, the most significant impact of this work is the possibility that centriolar satellites may be sites of local protein synthesis. Whether this would also be true for Cmb foci was not addressed in this work. Considering the potential conceptual advance, this submission warrants a few additional key experiments to support the proposed model and the authors' claims. In its present form, this submission is exciting and interesting, but the data do not fully support the conclusions (or the title).

Major points

1. The homology between PCM-1 and Cmb is tenuous. Did the authors try to rescue Cmb mutants with a human PCM-1 transgene? A rescue of male fertility or centriole structure, for example, would solidify the argument that the genes are functional orthologs.
2. Puro treatment in HeLa cells impaired the localization of several satellite proteins, showing translation is needed for their localization. Yet, it remains unclear if new protein synthesis is occurring at the centriolar satellites, which is a major claim of this work. Compelling evidence would be to show these proteins (by testing a subset therein) are being translated locally. While Puro-PLA is a commonly used approach, there are several other alternative assays one could use to show local protein synthesis of specified proteins. Without these data, the authors would have to revise the claims made in the text and the title.
3. The authors show a fraction of Cmb colocalizing with some cytoplasmic pools of a subset of centrosome proteins, SAS-4 in particular. It was surprising that the authors did not examine RNA localizing to Cmb foci. The claim (and title) suggesting these foci are sites of local protein synthesis remains, as yet, untested. To show functional overlap, one would ideally like to see if Cmb colocalizes with the same proteins - and RNAs - as PCM-1 and perhaps pick one to show local translation (see Major Point 2). The list need not be extensive. An obvious choice given the RNAs known to localize near centrosomes would be PCNT/PLP. A

caveat here is that while mammalian PCNT mRNA and protein localizes to centrosomes, PLP mRNA localization was not robust in S2 cells <https://www.ncbi.nlm.nih.gov/pmc/articles/PMC7921559/> . Several studies do show PLP mRNA enriched at Drosophila embryo centrosomes, however. This one also shows proximity to PLP protein: <https://pubmed.ncbi.nlm.nih.gov/35661190/> BicD or Ninein mRNAs might be alternative choices.

Minor points

1. Most of the insets in the figures (including supplemental material) lack scale bars.
2. Fig 2H, the authors present the frequency of embryos with NUF. The text suggests there is also an elevated incidence of lagging chromosomes upon Cmb LOF; however, this is not quantified.
3. It was not clear if the Cnn separation defects described in the text and presented in Fig. 2I are distinct from what was shown in 2H. When NUF occurs, a cluster of unseparated centrosomes remains at the cortex. The authors could use their existing movies and measure the time it takes for the duplicated centrosomes to separate relative to labeled nuclei. Alternatively, they could otherwise assess centrosome position relative to labeled nuclei to more clearly support this claim.
4. The visualization of interacting proteins by volcano plots is helpful and visually appealing; however, could the authors please use a different shade to distinguish the cilia vs. centrosome proteins? As presented, the two shades of pink are hard to distinguish on the graphs.
5. In Fig 4H, the LUT for the puro channel does not match the merge, which is significantly brighter. Adding arrows here and in several other panels to highlight points of overlap would be helpful.
6. When using puro to label puromycylated proteins (visualizing the nascent peptides) vs. a regime to inhibit translation, one uses very different concentrations of puromycin. It might be helpful to note this in the text for the general readership.
7. In panel 5B, gTub insets, the authors highlight two different regions of the cell: a gTub+ centrosome for the control vs. a noncentrosomal region of the cytoplasm for the puro-treatment. Why?
8. Also 5B, it was unclear how the overlap of PCM-1 vs PCNT mRNA signals were quantified among the treated vs. untreated groups, as the graph in 5B shows "measured" vs. "randomized." Which condition was measured?
9. The result in 5D is exciting. Is it feasible for the authors to show PCM-1 is required for PCNT mRNA localization while also co-staining for PCNT protein?
10. The reference to the Sepulveda et al 2018 work showing Pcnt mRNA and protein at centrosomes should be noted earlier - when the authors begin describing Fig 5B data.
11. A biochemical association, such as PCNT mRNA immunoprecipitated from PCM-1, would strengthen the authors' claim. Presumably, this association would be diminished in extracts treated with puro?
12. Related to Fig 5F, knocking out PCNT (or PLP) does impair PCM recruitment and MT nucleation, as reported in several papers: <https://pubmed.ncbi.nlm.nih.gov/18174396/> <https://pubmed.ncbi.nlm.nih.gov/15146056/> <https://www.ncbi.nlm.nih.gov/pmc/articles/PMC3246884/> <https://www.ncbi.nlm.nih.gov/pmc/articles/PMC2172389/> <https://pubmed.ncbi.nlm.nih.gov/26150390/>

It would be appropriate for the authors to acknowledge this caveat when describing their interpretation of data in panel 5F.

Reviewer #2 (Comments to the Authors (Required)):

The manuscript by Pachinger et al identifies the Drosophila ortholog of the centrosome satellite protein PCM1, called Combover (CMB), and perform in vivo experiments to investigate its loss of function phenotypes. Following this characterizing, they performed turboID to identify interacting proteins of CMB, from which they focus on translational machinery hits. The interaction data led to the hypothesis that satellites are sites of active translation of centrosome proteins. The manuscript then takes an abrupt turn to mammalian culture system, never to return to CMB. It is here where the authors attempt to show that satellites are actively translating centrosome proteins. The overarching competing models in the field are 1) satellites are "centrosome protein delivery systems" vs 2) "centrosome protein production sites", the latter being the favored model from the authors based on data presented in this manuscript. While their data does suggest a translation model, it falls short of showing centrosome protein

translation. In my opinion, the results are overinterpreted.

Main concerns

1) That satellites are sites of active translation of centrosome proteins.

- The authors use Puromycin to show that CMB foci are site of translation. This is nicely shown. Although, removing the word "remarkable" would be preferable as 15% seems low.

- Puro experiment in vertebrate cells: short incubations do not show translation at satellites. This treatment results in loss of protein at satellites, but not the centrosome. Cycloheximide shows the same result. The conclusion here is that "these proteins" found at satellites are newly formed, and I assume, they mean not a result of translocation from the cytoplasm to the satellite in wild-type conditions.

It is then that I don't follow the logic. The authors suggest that because centrosome protein levels did not increase at centrosomes following cycloheximide treatment, then this rules out protein redistribution. I assume they mean this experiment rules out the model in the field that satellites deliver centrosome proteins to centrosomes. My conclusion would be opposite, if satellites are in fact delivering protein, then puro or cyclo treatment would predictably result in no increase at the centrosome because the satellites no longer contain deliverable centrosome proteins. Additional clarification here is important.

- Now we get to the mRNA localization studies. I disagree with the conclusion that PCNT is "translated at or in the immediate vicinity of satellites". The authors show mRNA localization not translation. Importantly, the authors correctly state on page 12, 4th line from the bottom, that ".... Are potentially coordinately translated". I agree with this conclusion and strongly believe that it is the most one can say about the data. However, the authors then appear to go a couple of steps beyond this conclusion in the next sentence that claims "loss of satellite signal from centrosome clients following PCM1 depletion appears to reflect a failure to synthesize proteins on the satellites." This is an extreme leap. The conclusion from there being no signal on satellites is likely that there are no satellites following PCM1 depletion.

Bottom line for this reviewer is that no method of showing active translation of centrosome proteins at satellites was used in this study. Thus the main thrust, conclusion and model is not supported. Everything else below this point is somewhat irrelevant unless the authors can address translation directly. Meaning watching it happen.

2) The final experiment of the paper is not clear. The authors discuss the model in the literature whereby PCNT on satellites are functionally upstream of PCNT mRNA and protein delivery to the centrosome. Then propose an alternative mode, presumably based on their previous conclusion in this current work, that co-translational protein transport would drive "their" (I think they mean PCNT and PCM1) accumulation at centrosomes. What is co-translational protein transport? Then the authors seem to flip the model whereby PCNT is now upstream of PCM1 localization and accumulation at centrosomes. I do not follow.

3) The manuscript is extremely difficult to read due to the style of writing that forces the reader to fill in the blanks and guess which proteins, phenotypes, and discussion points are being referenced. A simple solution is to tell the reader, instead of having one guess or repeatedly search through each element of a figure. Here are some of the dozens of cases of this style of writing: Page 4, line 4. the sentence starting with "Thus" does not follow the previous sentence. This is likely a misuse of the word thus. Page 4, line 9. Centrosome positioning is also largely unaffected *in PCM1 mutants*. This is an example of the need for additional explanation to make reading easier.

Page 6, line 5 from bottom. Both types of defects - restating them here is preferable. Both abnormal wing posture and impaired geotaxis. Also, is the rescue experiment 'data not shown' in this case. I don't think JCB allows this.

Page 7, line 3. Which defects?

Page 9, middle of page. "These proteins" - which proteins? List in text.

Page 11, 6th line from bottom. "Cytoplasmic signal" - does this mean satellite signal?

Page 11, 3rd line from bottom. "ruling out protein redistribution" - from where to where exactly

Page 11, 3rd line from bottom. "these proteins" - which proteins?

Page 12, line 1. "similarly" - to what?

Page 12, middle. "The same was true" - what was true?

4) Did the authors attempt to investigate translation in *Drosophila*? It is not clear why this was not done in relation to CMB foci.

5) The title does not properly represent the data. As discussed above, translation of centrosomal and ciliary proteins at satellites was not shown. Furthermore, the authors do not show conservation, meaning a role for CMB.

Minor concerns

Rephrase the sentence on page 6 "mutants were also fully male (though not female) infertile."

On page 7, how was weakly motile sperm actually determined?

Page 7. Parental should be paternal.

Page 7. The word famously is highly unnecessary

Page 7. The work from Poulton and Peifer should be referenced.

Page 10, line 1 and 2. TurboID -how is "no marked difference" determined?

Page 10, second paragraph. Why are the authors surprised to see RBPs and translation factors? Hadn't previous work suggested this?

Page 12, line 4. It would be best to convert this from a conclusion to a hypothesis that centrosome and cilia proteins are

synthesized through independent mechanisms.
Page 12, bottom third. "therefore" should be "but also"

Reviewer #3 (Comments to the Authors (Required)):

In this paper, Pachinger et al., characterise the composition, behaviour and function of centriole-satellites in *Drosophila*. These structures have been extensively characterised in a variety of cells, but their function remains unclear and controversial. In part, this is because satellites have not been properly studied in model systems such as worms and flies as it has been difficult to identify clear homologues of PCM1, the key satellite-scaffolding protein. Here the authors confirm that the *Drosophila* protein Comover (Cmb), which they found in a TurboID screen with the key centriole assembly protein Sas-4/CPAP, appears to be the functional homologue of PCM1, despite only very limited sequence homology. They show that Cmb is required for proper centriole/cilia function in flies (and possibly for centrosome function as well, see below), but Cmb-satellites are not obviously concentrated around centrioles/centrosomes. Most importantly, using a BioID/MS approach they show that Cmb satellites contain several centriole/centrosome/cilia proteins, but also many proteins involved in RNA binding and protein translation. They provide evidence that Cmb/PCM1-satellites function as sites of protein translation for centriole/centrosome/cilia proteins, and propose that this may be the major conserved function of these satellite structures.

The data in this paper is generally well-presented and of high-quality. This new hypothesis about PCM1/Cmb-satellite function is intriguing and will be of considerable interest to the centriole/cilia/centrosome fields, so I am strongly in favour of publication in JCB. I do, however, have a number of suggestions that the authors should consider prior to publication.

Major points:

1. The data showing the potential co-localisation of various centriole/centrosome proteins with the Cmb condensates is crucial to the authors main conclusion, but in some places it was a bit confusing or unclear. In Fig.3I the authors show that some cytoplasmic foci of Sas-4 and Cnn colocalise with one of the many Cmb condensates in the cell. The description of how this experiment was performed (p28, para.2) could be clearer, as I wasn't sure if users defined the region of the cell to be analysed with any prior knowledge of the distribution of the cytoplasmic foci and Cmb condensates, or whether these regions were chosen at random (either by the user or computationally). This is a really important point.
2. The authors show similar co-localisation data for several other proteins (Fig.S3C), but don't show the correlation coefficient (shown for Sas-4 and Cnn in Fig.3I), so one has to read the Legend to see which proteins do and don't co-localise with Cmb (and this information does not seem to be provided for CEP97?). The list of proteins that the authors have tested that do and don't colocalise with Cmb condensates should be made clearer. The authors should comment on whether this list make sense with the idea that proteins destined to form complexes together are co-sequestered in Cmb condensates. Ideally, it would be great to see evidence for co-localisation of any such protein pairs in the same Cmb-condensate, but I appreciate this may be technically challenging so would not insist on this.
3. As the authors note, two previous studies examined the composition of centriole satellites in human cells (Quarantotti et al., 2019; Gheiratmand et al., 2019). These papers focused on centriole, centrosome and cilia proteins, but I wonder if RNA processing/protein translation proteins were also enriched in these previous studies? In theory, the authors give this information in Table S2, but I found this too difficult to wade through. The authors should address this point more directly.
4. In Figure 4H the authors should perform the same correlation experiments comparing the real images to randomised (rotated) images that they perform in Figure 3I. The significance of the correlations currently shown are hard to judge as there are so many more Puromycin dots in the 3min Puro image. On a related point, only ~15% of Cmb condensates are Puro+. Does this suggest that ~85% of Cmb condensates are not actively translating anything (in this 3min window), or is this likely to be a problem of sensitivity of detection? Do you get more Cmb+,Puro+ dots if you extend the incubation time to 10mins?
5. In Figure 2 the authors show that embryos laid by homozygous Cmb Δ -/- females exhibit nuclear fall-out defects (Fig.2H) and also centrosome separation defects (Fig.2I), and they speculate that the failure in centrosome separation may explain the mitotic defects observed in these embryos. From the images shown, it seems that the centrosome separation defects might largely occur in regions where the nuclei have already fallen from the cortex? If so, the centrosomes that remain at the cortex in these regions devoid of nuclei usually fail to separate properly (even in WT embryos), as movement around the nuclear periphery seems to be an important driver of centrosome separation. The authors might want to check this and, if correct, I think they should tone down their conclusion that Cmb mutants exhibit centrosome defects (as I think this is their only evidence for centrosome defects?). The idea that PCM1/Cmb primarily contributes to centriole/cilia function, rather than centrosome function, would fit well with most of the authors data, and perhaps is an emerging theme in the centriole satellite field.

Having said that, it is intriguing that the embryos laid by Cmb Δ -/- females can survive if they carry a WT copy of Cmb provided by

the sperm. This also argues against any centrosome defects in the syncytial embryo contributing to embryonic lethality (as the WT gene supplied by the male would be largely silent during the syncytial stages). This embryonic lethality does suggest a non-centriole/cilium function for Cmb (as I think embryos are largely devoid of cilia?), but the authors don't have much evidence that this is due to a function at centrosomes (if my point above is correct). It might be worth commenting on this.

Minor Points:

1. The description in the text of the experiment shown in Fig.5A is confusing as the authors originally state that there is "a striking loss of cytoplasmic signal for centrosomal and ciliary proteins". I believe that OFD1 is a ciliary protein, so it is confusing to readers that its cytoplasmic levels rise here. This should be reworded.
1. The authors note that a PCM1 homologue in flies was identified previously (Kuhn et al., 2014), but then imply that they had to use sophisticated comparisons to identify this homologue in their TurboID hits. I assume this is the same protein, so why didn't they already know that CMB was the fly homologue of PCM1?
2. The authors should provide more information on the Cmb mutant. After some searching I figured out that it is a partial deletion, but has there been any analysis of whether this is (or behaves as) a null? It is called Cmb Δ , implying it is a null, but worth clarifying.
3. Related to the potential role of PCM1/Cmb in centrosomes, it might be worth checking if Dictyostelium has PCM1, as this organism lacks centrioles and cilia, yet it has several proteins required to make mitotic centrosomes (e.g. homologues of Cnn/CDK5RAP2 and Spd-2/CEP192). If it doesn't, it might further support the argument that PCM1/Cmb is not required for centrosome function.
4. I don't think the authors mention whether any of the proteins that localise in Cmb-foci also form foci that don't colocalise with Cmb? If so, do the authors think these are different translation hubs, allowing the proteins to form different complexes?
5. More help is needed with Fig.S2E: what are we looking at in each section, and what defects are the orange arrows highlighting?
6. P8, para.2. "or any fly tissue" should be changed to "or in any of several fly tissues that we examined".
7. The model in Fig.5G was not very helpful. I'm not sure what it was proposing.
8. Figure 1C. It looks to me like the otherwise invariant "LR" amino acids are present in the Drosophila Cmb protein-if you just shift the position of the gap in the MSA.

RESPONSE TO REVIEWERS

We would like to thank the editor and reviewers for their time and expert analysis of our work. In preparing this revision we have sought to incorporate their suggestions as far as possible. We believe our manuscript to be substantially improved by these changes, including the addition of new data demonstrating protein synthesis at PCM1-containing satellites by Puro-PLA in vertebrate cells and the association of CMB with centrosomal mRNA by smFISH in flies. Below is our detailed point-by-point response, along with the reviewers' original comments in full in *italics*. New experimental data is highlighted in **bold**.

Reviewer #1:

Pachinger et al. identify Comover (Cmb) as a protein interacting with Drosophila SAS-4 and characterize its role in ciliogenesis. The authors convincingly describe a role for Cmb in ciliary function and identify several known centrosome and ciliary proteins interacting with Cmb via TURBO-ID proximity labeling mass spectrometry. Some of these interactions were validated through colocalization analysis. Their comparative proteomics analyses will provide the field a valuable resource, while also characterizing the role of Cmb in ciliogenesis and mitotic fidelity.

Given a weak sequence homology, the authors suggest Cmb may be the Drosophila ortholog to PCM-1. To support this hypothesis, the authors point to the role for Cmb in ciliogenesis and highlight several shared protein interactors from their own Cmb dataset versus published PCM-1 proteomics. However, the evidence supporting the idea that Cmb is the functional ortholog of PCM-1 remains weak. For this reviewer, the most significant impact of this work is the possibility that centriolar satellites may be sites of local protein synthesis. Whether this would also be true for Cmb foci was not addressed in this work. Considering the potential conceptual advance, this submission warrants a few additional key experiments to support the proposed model and the authors' claims. In its present form, this submission is exciting and interesting, but the data do not fully support the conclusions (or the title).

Major points

1. The homology between PCM-1 and Cmb is tenuous. Did the authors try to rescue Cmb mutants with a human PCM-1 transgene? A rescue of male fertility or centriole structure, for example, would solidify the argument that the genes are functional orthologs.

We readily accept that CMB is only distantly related to PCM1 at the primary sequence level. If it were more clearly conserved, it would have already been identified in prior analyses of the evolutionary distribution of centrosomal and ciliary proteins (e.g. Hodges et al, JCS 2010) rather than necessitating the development of novel and more sophisticated tools for bioinformatic analysis (Kuhn et al, PLOS Comp Biol 2014). Where primary sequence conservation is low there will always be debate whether the proteins in question are orthologs or merely related in function. Such debates are only sometimes resolved by unambiguous similarities in 3D structure or clear mechanistic parallels (as in the case of ZYG-1 and SAK/PLK4).

We are not aware of any examples of successful cross-species complementation for centrosomal or ciliary proteins between flies and humans, even for proteins much more highly conserved than PCM1, given the substantial degree of evolutionary divergence between the

two species. Nevertheless, **following the reviewer's suggestion we expressed *Drosophila* CMB in vertebrate cells** (the reverse experiment, stably expressing PCM1 in flies and introducing it into a *Cmb* mutant background would have taken substantially longer). **As shown in Fig. S3F, CMB remarkably near-perfectly colocalized with endogenous PCM1 in HeLa cells**, to a greater extent than any satellite clients tested. **However, perhaps unsurprisingly given the low degree of sequence homology it could not productively engage the translation machinery to sustain the formation of cytoplasmic PCNT foci upon RNAi-mediated depletion of PCM1 (Fig S3G).**

2. Puro treatment in HeLa cells impaired the localization of several satellite proteins, showing translation is needed for their localization. Yet, it remains unclear if new protein synthesis is occurring at the centriolar satellites, which is a major claim of this work. Compelling evidence would be to show these proteins (by testing a subset therein) are being translated locally. While Puro-PLA is a commonly used approach, there are several other alternative assays one could use to show local protein synthesis of specified proteins. Without these data, the authors would have to revise the claims made in the text and the title.

We agree that our inability to visualize nascent protein synthesis using the puromycin incorporation assay in vertebrate cells and hence detect it at satellites as we did in *Drosophila* S2 cells was a weakness of our original manuscript. We attribute this to the sheer density of active ribosomes and hence global protein synthesis in the mammalian cytoplasm, although we note that in the manuscript by Chad Pearson (Martinez *et al.*, Biorxiv 2024) the authors did observe such translation using the puromycin analog O-propargyl-puromycin (OPP).

As the reviewer notes, there are a variety of assays to detect local translation of specific proteins, such as the SunTag (Tanenbaum *et al.*, Cell 2014) and we intend to use such reporters to further characterize translation at centriolar satellites in future work. However, their use requires considerable optimization and validation to detect nascent protein and ensure signal specificity. **For this revision, we therefore turned to the puromycin proximity ligation assay (Puro-PLA) mentioned by the reviewer, which combines puromycin incorporation with proximity ligation to make it specific for a particular protein of interest. Using PCNT as our model satellite client, we now show that PCNT protein is synthesized in close proximity to PCM1 foci (Fig. 5 A, B).** This, combined with our prior work showing that cytoplasmic signal for a variety of other satellite clients is sensitive to inhibition of protein synthesis (i.e. all such signal represents newly synthesized protein), strongly supports our contention that satellites not only associate with centrosomal and ciliary mRNAs but also promote their translation.

*3. The authors show a fraction of Cmb colocalizing with some cytoplasmic pools of a subset of centrosome proteins, SAS-4 in particular. It was surprising that the authors did not examine RNA localizing to Cmb foci. The claim (and title) suggesting these foci are sites of local protein synthesis remains, as yet, untested. To show functional overlap, one would ideally like to see if Cmb colocalizes with the same proteins - and RNAs - as PCM-1 and perhaps pick one to show local translation (see Major Point 2). The list need not be extensive. An obvious choice given the RNAs known to localize near centrosomes would be PCNT/PLP. A caveat here is that while mammalian PCNT mRNA and protein localizes to centrosomes, PLP mRNA localization was not robust in S2 cells <https://www.ncbi.nlm.nih.gov/pmc/articles/PMC7921559/>. Several studies do show PLP mRNA enriched at *Drosophila* embryo centrosomes, however. This one also shows proximity to PLP protein: <https://pubmed.ncbi.nlm.nih.gov/35661190/> BicD or Ninein mRNAs might be alternative choices.*

Having identified CMB as the functional ortholog of PCM1 and discovering proteins involved in translation/translation-coupled quality control as a common denominator between the (proximity) interactomes of PCM1 and CMB, our focus was on exploring this potential link to translation in vertebrate cells, not *Drosophila*, on the basis that this is where the majority of work on satellites has been carried out and where evidence for such a role would have the most impact.

To strengthen our contention that CMB performs a similar role in *Drosophila*, we now show by smFISH that CMB associates with the mRNA encoding SAS-4, the protein most prominently localized to CMB satellite foci in flies (Fig. S4E-G). We would like to note that the concentration of mRNAs in the vicinity of centrosomes, a consequence of co-translational targeting, is not a common feature in flies. Such targeting may occur for certain satellite clients in vertebrates, such as PCNT, but this is unrelated to satellite function in the process of translation itself. Hence, neither CMB nor SAS-4 mRNA concentrates in the vicinity of centrosomes. We expect this to be true also for other satellite clients in the fly.

Minor points

1. *Most of the insets in the figures (including supplemental material) lack scale bars.*

Scale bars have now been added to all insets.

2. *Fig 2H, the authors present the frequency of embryos with NUF. The text suggests there is also an elevated incidence of lagging chromosomes upon Cmb LOF; however, this is not quantified.*

We have re-examined our movies as also suggested by reviewer 3, and nuclear fallout invariably followed chromosome missegregation. Those phenotypes are therefore likely causally linked. We have changed our presentation of these results accordingly.

3. *It was not clear if the Cnn separation defects described in the text and presented in Fig. 2I are distinct from what was shown in 2H. When NUF occurs, a cluster of unseparated centrosomes remains at the cortex. The authors could use their existing movies and measure the time it takes for the duplicated centrosomes to separate relative to labeled nuclei. Alternatively, they could otherwise assess centrosome position relative to labeled nuclei to more clearly support this claim.*

This has been slightly more difficult to assess given that failure of centrosome separation is more readily apparent using a PCM marker such as CNN and those sequences did not include a histone countermarker. However, **the reviewer appears to be correct that failure of centrosome separation follows nuclear fallout, which places chromosome missegregation as the most upstream event.** Why this occurs we can only speculate. Our analysis revealed centriole duplication and recruitment of the scaffolding component CNN to be normal. However, this does not preclude defects in the recruitment of other PCM components, such as D-PLP. As the reviewer notes above, the mRNA encoding this protein has been found to be weakly enriched at centrosomes during early embryogenesis and perturbing its expression in mutants of the cytoplasmic polyadenylation element binding (CPEB) protein ORB results in chromosome missegregation and nuclear fallout, similar to what we see for CMB (Fang and Lerit, Development 2022). We have been unable to assess D-PLP recruitment in *Cmb* mutants, but its dependence on PCM1 for expression in vertebrates is certainly intriguing.

4. *The visualization of interacting proteins by volcano plots is helpful and visually appealing;*

however, could the authors please use a different shade to distinguish the cilia vs. centrosome proteins? As presented, the two shades of pink are hard to distinguish on the graphs.

We have changed the color scheme for centrosomal proteins to a darker shade of purple. Our main aim in the volcano plots in Figures 4 and S4 was to highlight the presence of centrosomal and ciliary proteins, RNA-binding and translation-related proteins, and proteins involved in proteostasis and protein quality control. The distinction within each major category, such as between a centrosomal/centriolar and ciliary protein is often somewhat arbitrary and maybe not worth emphasizing.

5. In Fig 4H, the LUT for the puro channel does not match the merge, which is significantly brighter. Adding arrows here and in several other panels to highlight points of overlap would be helpful.

The pink puro channel in the merge images was intentionally boosted for all three conditions in order to avoid that signal being swamped by the brighter green CMB signal. However, we agree that the brightness of the single color black and white images could also have been increased, which we have now done. We have refrained from adding arrows to the figure to highlight additional points of coincidence of CMB and Puro signals beyond the two foci in the inset to avoid obscuring other foci in the main image.

6. When using puro to label puromycylated proteins (visualizing the nascent peptides) vs. a regime to inhibit translation, one uses very different concentrations of puromycin. It might be helpful to note this in the text for the general readership.

We now note this in the text.

7. In panel 5B, gTub insets, the authors highlight two different regions of the cell: a gTub+ centrosome for the control vs. a noncentrosomal region of the cytoplasm for the puro-treatment. Why?

PCM1 and PCNT mRNA both disperse throughout the cytoplasm upon puromycin treatment. We therefore picked a representative area. However, we could also have selected a region encompassing the centrosome. We now do this for consistency.

8. Also 5B, it was unclear how the overlap of PCM-1 vs PCNT mRNA signals were quantified among the treated vs. untreated groups, as the graph in 5B shows "measured" vs. "randomized." Which condition was measured?

Distances between PCM1 foci and the nearest PCNT mRNA particle were measured for both 'measured' and 'randomized' conditions. However, for the randomized condition, the PCNT mRNA signal was computationally reshuffled within the image mask for each cell before quantitation. The figure legend now refers the reader to the Methods where this is explained in detail.

9. The result in 5D is exciting. Is it feasible for the authors to show PCM-1 is required for PCNT mRNA localization while also co-staining for PCNT protein?

Technically, yes, although we are not sure what this would add. Cytoplasmic PCNT protein foci are entirely dependent on PCM1 and hence absent in PCM1 RNAi conditions (see Fig. 6A in revised manuscript). Those PCNT mRNA foci in both interphase (Fig. 6C) and mitosis (Fig. 6B) now found dispersed across the cytoplasm are therefore not associated with PCNT protein.

10. The reference to the Sepulveda *et al* 2018 work showing *Pcnt* mRNA and protein at centrosomes should be noted earlier - when the authors begin describing Fig 5B data.

The work on co-translational targeting of centrosomal proteins by Sepulveda *et al.* and others was clearly critical to our understanding of PCM1 localization and dynamics. However, it only makes sense to discuss this work in that context (i.e. at the beginning of what is now presented in Fig. 6), not when showing that PCM1 associates with centrosomal/ciliary mRNA.

11. A biochemical association, such as PCNT mRNA immunoprecipitated from PCM-1, would strengthen the authors' claim. Presumably, this association would be diminished in extracts treated with puromycin?

We do intend to characterize the RNA interactome of PCM1 by RIP-Seq to explore satellite composition and identify potential targeting motifs in satellite clients. However, such experiments are not trivial and beyond the scope of the current manuscript. Regarding puromycin sensitivity of the PCM1-RNA interaction, as we show in Figure 5B, *PCNT* mRNA continues to associate with PCM1 satellite foci in the presence of protein synthesis inhibitor. We therefore do not expect to see any reduction in RNA association.

12. Related to Fig 5F, knocking out PCNT (or PLP) does impair PCM recruitment and MT nucleation, as reported in several papers:

<https://pubmed.ncbi.nlm.nih.gov/18174396/> <https://pubmed.ncbi.nlm.nih.gov/15146056/>
<https://www.ncbi.nlm.nih.gov/pmc/articles/PMC3246884/> <https://www.ncbi.nlm.nih.gov/pmc/articles/PMC2172389/> <https://pubmed.ncbi.nlm.nih.gov/26150390/>

It would be appropriate for the authors to acknowledge this caveat when describing their interpretation of data in panel 5F.

We are aware that *PCNT* depletion perturbs mitotic centrosome organization and this could indirectly lead to a reduction of PCM1 at mitotic spindle poles. We now note this caveat in our discussion of these data. However, the effect is rather minor: quantitative analyses of centrosomal gamma-tubulin consistently estimate levels to be reduced to ~50% of controls following *PCNT* RNAi (see Haren *et al.*, PLOS One 2009; Lawo *et al.*, Nat Cell Biol 2012). The same is true in the *PCNT* mutant used in this study (Watanabe *et al.*, JCB 2020, see also Fig. 6D) - a reduction, certainly, but not a total loss of enrichment as we see for PCM1. Indeed, one of the critical findings of Watanabe *et al* was that mitotic spindle assembly and chromosome segregation are not compromised even upon double depletion of both *PCNT* and *CDK5RAP2*, with centrosomes still capable of sustaining mitosis. An indirect effect on PCM1 via impaired microtubule organization is therefore unlikely.

Reviewer #2:

The manuscript by Pachinger *et al* identifies the *Drosophila* ortholog of the centrosome satellite protein PCM1, called Comover (CMB), and perform *in vivo* experiments to investigate its loss of function phenotypes. Following this characterizing, they performed *turboID* to identify interacting proteins of CMB, from which they focus on translational machinery hits. The interaction data led to the hypothesis that satellites are sites of active translation of centrosome proteins. The manuscript then takes an abrupt turn to mammalian culture system, never to return to CMB. It is here where the authors attempt to show that satellites are actively translating centrosome proteins. The overarching competing models in the field are 1) satellites are "centrosome protein delivery systems" vs 2) "centrosome protein

production sites", the latter being the favored model from the authors based on data presented in this manuscript. While their data does suggest a translation model, it falls short of showing centrosome protein translation. In my opinion, the results are overinterpreted.

Main concerns

1) That satellites are sites of active translation of centrosome proteins.

- The authors use Puromycin to show that CMB foci are site of translation. This is nicely shown. Although, removing the word "remarkable" would be preferable as 15% seems low.

We have removed the offending word. However, we still find it remarkable that a puromycin incorporation assay not specific for centrosomal/ciliary proteins reveals any distinctive signal at satellites considering the sheer density of ribosomes in the cytoplasm and the overall level of global protein synthesis. To be clear, the remaining 85% of satellite may still sustain translation, but not at a level that rises above the cytoplasmic background.

- Puro experiment in vertebrate cells: short incubations do not show translation at satellites. This treatment results in loss of protein at satellites, but not the centrosome. Cycloheximide shows the same result. The conclusion here is that "these proteins" found at satellites are newly formed, and I assume, they mean not a result of translocation from the cytoplasm to the satellite in wild-type conditions. It is then that I don't follow the logic. The authors suggest that because centrosome protein levels did not increasing at centrosomes following cycloheximide treatment, then this rules out protein redistribution. I assume they mean this experiment rules out the model in the field that satellites deliver centrosome proteins to centrosomes. My conclusion would be opposite, if satellites are in fact delivering protein, then puro or cyclo treatment would predictable result in no increase at the centrosome because the satellites no longer contain deliverable centrosome proteins. Additional clarification here is important.

The reviewer is already one step ahead. With the experiment in Fig. 5C we sought to exclude another alternative explanation, that satellite signal is lost because it is redistributed to centrosomes (which would be predicted to lead to an increase in centrosomal signal). That it does not increase unambiguously identifies the satellite client protein population as newly synthesized and not a pool that normally shuttles to and from the centrosome. What this experiment does not do is distinguish between satellites acting as sites of translation or as waystations in the transport of nascent protein to centrosomes (in both cases as the reviewer correctly notes there would not be a change or – long term – a reduction in centrosomal signal).

- Now we get to the mRNA localization studies. I disagree with the conclusion that PCNT is "translated at or in the immediate vicinity of satellites". The authors show mRNA localization not translation. Importantly, the authors correctly state on page 12, 4th line from the bottom, that " Are potentially coordinately translated". I agree with this conclusion and strongly believe that it the most one can say about the data. However, the authors then appear to go a couple of steps beyond this conclusion in the next sentence that claims "loss of satellite signal form centrosome clients following PCMI depletion appears to reflect a failure to synthesize proteins on the satellites." This is an extreme leap. The conclusion from there being no signal on satellites is likely that there are no satellites following PCMI depletion. Bottom line for this reviewer is that no method of showing active translation of centrosome proteins at satellites was used in this study. Thus the main thrust, conclusion and model is not supported. Everything else below this point is somewhat irrelevant unless the authors can address translation directly. Meaning watching it happen.

The reviewer is correct that in the original manuscript we did not show nascent protein at satellites in vertebrates, as we did using the puromycin incorporation assay in flies. However,

based on the previous point the reviewer appears to agree with our interpretation of our inhibition of protein synthesis experiment identifying the cytoplasmic signal of satellite clients such as PCNT as newly synthesized. If this population is newly synthesized and satellites are associated with mRNA encoding these proteins, the most straightforward explanation (consistent also with the identification of components of the translation machinery associated with CMB and also PCM1) is that translation occurred at satellites.

We agree with the reviewer that visualizing protein synthesis also in vertebrate cells would be desirable. There are a variety of assays to detect local translation of specific proteins, such as the SunTag (Tanenbaum *et al.*, Cell 2014) and we intend to use such reporters to further characterize translation at centriolar satellites in future work. However, their use requires considerable optimization and validation to detect nascent protein and ensure signal specificity. **For this revision, we therefore turned to the puromycin proximity ligation assay (Puro-PLA) suggested by reviewer 1 (Major point 2), which combines puromycin incorporation with proximity ligation to make it specific for a particular protein of interest. Using PCNT as our model satellite client, we now show that PCNT protein is synthesized in close proximity to PCM1 foci (Fig. 5A, B).** This strongly supports our contention that satellites not only associate with centrosomal and ciliary mRNAs but also promote their translation.

2) The final experiment of the paper is not clear. The authors discuss the model in the literature whereby PCM1 on satellites are functionally upstream of PCNT mRNA and protein delivery to the centrosome. Then propose an alternative mode, presumably based on their previous conclusion in this current work, that co-translational protein transport would drive "their" (I think they mean PCNT and PCM1) accumulation at centrosomes. What is co-translational protein transport? Then the authors seem to flip the model whereby PCNT is now upstream of PCM1 localization and accumulation at centrosomes. I do not follow.

This is exactly what we are saying. Co-translational protein transport is a phenomenon that has been extensively documented for centrosomal proteins including PCNT in vertebrates (Sepulveda *et al.*, eLife 2018; Safieddine *et al.*, Nat Commun 2021; reviewed in Lerit, Mol Biol Cell 2022), with the N-terminus of the nascent protein directing the entire mRNA-protein complex to centrosomes. The benefits of such co-translational targeting are difficult to determine, but have been proposed to include the timely delivery of proteins to sustain mitosis. In the original trafficking model of satellite function, PCM1 could be an active participant in that transport, linking nascent PCNT and other cargo to microtubule motors. In our revised translation model, satellites are only passively involved in that they are engaged in the synthesis of those clients, which then drag the satellite particle to centrosomes. Both models are consistent with the observed defects in centrosomal enrichment of PCNT and *PCNT* mRNA following PCM1 depletion. Where they differ is in their prediction of the consequences of loss of PCNT. If PCNT is a cargo of a satellite trafficking module, its loss shouldn't impair the continued shuttling of satellites and hence PCM1 concentration at mitotic centrosomes. If, however, the translation of PCNT is what primarily leads to the redistribution of satellites, then loss of PCNT should impair PCM1 accumulation. This latter outcome is what we found. With PCM1 involved in translation of centrosomal proteins including PCNT, PCM1 and satellites are unambiguously 'upstream' of PCNT. Their localization and dynamics, however, can be affected by the co-translational transport of specific clients like PCNT (hence in another sense also 'downstream'). However, since *Drosophila* shows no evidence of co-translational transport influencing satellite distribution, we do not believe this passive movement is critical to their function in translation.

3) *The manuscript is extremely difficult to read due to the style of writing that forces the reader to fill in the blanks and guess which proteins, phenotypes, and discussion points are being reference. A simple solution is to tell the reader, instead of having one guess or repeatedly search through each element of a figure. Here are some of the dozens of cases of this style of writing:*

Page 4, line 4. the sentence starting with "Thus" does not follow the previous sentence. This is likely a misuse of the word thus.

*Page 4, line 9. Centrosome positioning is also largely unaffected *in PCM1 mutants*. This is an example of the need for additional explanation to make reading easier.*

Page 6, line 5 from bottom. Both types of defects - restating them here is preferable. Both abnormal wind posture and impaired geotaxis. Also, is the rescue experiment 'data not shown' in this case. I don't this JCB allows this.

Page 7, line 3. Which defects?

Page 9, middle of page. "These proteins" - which proteins? List in text.

Page 11, 6th line from bottom. "Cytoplasmic signal" - does this mean satellite signal?

Page 11, 3rd line from bottom. "ruling out protein redistribution" - from where to where exactly

Page 11, 3rd line from bottom. "these proteins" - which proteins?

Page 12, line 1. "similarly" - to what?

Page 12, middle. "The same was true" - what was true?

This manuscript was initially conceived as a short-format report, which necessitated eliminating any redundancies in the main text, leading to the abbreviated style the reviewer is criticizing. Given that JCB has more relaxed character limits we have expanded those and other passages in the text to make the manuscript more readable.

Regarding the reviewer's query, the 'cytoplasmic signal' we refer to for the puromycin incorporation assay on page 11 is any punctate puromycin signal, whether on satellites or elsewhere. What we instead found using this assay was diffuse cytoplasmic signal at any puromycin concentration and incubation timepoint tested. This precluded any analysis of the spatial distribution of global protein synthesis as we had performed in *Drosophila* S2 cells. We attribute this to the sheer density of active ribosomes in the cytoplasm of vertebrate cultured cells. As discussed above we now circumvent this limitation by monitoring the translation of one specific satellite client, PCNT, using the Puro-PLA assay.

4) *Did the authors attempt to investigate translation in Drosophila? It is not clear why this was not done in relation to CMB foci.*

Having identified CMB as the functional ortholog of PCM1 and discovering a potential link to translation, our focus was on exploring this link in vertebrate cells, not *Drosophila*, on the basis that this is where the majority of work on satellites has been carried out and where evidence for such a role would have the most impact. If the reviewer is referring to confirming the association of nascent protein with satellites, this is where we previously had the stronger evidence in *Drosophila* using the puromycin incorporation assay. As for satellite foci also harboring centrosomal/ciliary mRNAs, the reviewer is correct that we had previously only shown this in vertebrate cells since smFISH particularly when combined with immunofluorescence staining requires optimization in any new cell line. **To strengthen our contention that CMB and satellites perform a similar role in *Drosophila* as in vertebrates, we now show by smFISH that CMB associates with the mRNA encoding SAS-4, the protein most prominently localized to CMB satellite foci in flies (Fig. S4E-G).**

5) *The title does not properly represent the data. As discussed above, translation of centrosomal and ciliary proteins at satellites was not show. Furthermore, the authors do not show conservation, meaning a role for CMB.*

We respectfully disagree. Puromycin incorporation assays with or without proximity ligation (Puro-PLA) and single molecule RNA fluorescence in situ hybridization (smFISH) are standard tools to visualize sites of protein synthesis and have been used as evidence of local translation in other recent studies, e.g. Park *et al.*, *Dev Cell* 2023 (local translation at mammalian midbodies). While there are numerous biochemical and microscopy-based tools to further characterize translation and we intend to use these tools in follow up work, we believe our current manuscript providing the first evidence of such local translation occurring at centriolar satellites should be held to the same standard. Regarding demonstrating a conserved function of satellites in translation, as detailed elsewhere we now show evidence for mRNA association (smFISH) and protein synthesis (puromycin incorporation/Puro-PLA) in both *Drosophila* and vertebrate cells. We therefore believe the title to fairly represent our data.

Minor concerns

Rephrase the sentence on page 6 "mutants were also fully male (though not female) infertile."

We believe the meaning of this sentence to be clear. Mutants are male but not female infertile.

On page 7, how was weakly motile sperm actually determined?

As detailed in the Methods section, sperm motility was examined by live imaging of sperm stored in the seminal vesicle of 3-day old males. As can be seen in the still images in Fig. S2F, control sperm upon dissection into Schneider medium display vigorous sinusoidal movement. In contrast, residual *Cmb* mutant sperm were barely motile.

Page 7. Parental should be paternal.

No. Paternal effect lethality would imply lethality upon introduction of mutant sperm into a wild-type oocyte. In the case of *Cmb* both maternal and paternal contribution have to be eliminated in order to observe embryonic lethality (i.e. no maternal or zygotic contribution of CMB).

Page 7. The word famously is highly unnecessary

The offending word has been removed.

Page 7. The work from Poulton and Peifer should be referenced.

This reference has been included.

Page 10, line 1 and 2. TurboID -how is "no marked difference" determined?

This statement is based on the fact that 61 of 72 significantly enriched proteins (85%) found by mass spectrometry in the S2 cell insoluble fraction were also detected in the soluble fraction (Fig. S4B), a remarkable degree of overlap.

Page 10, second paragraph. Why are the authors surprised to see RBPs and translation factors? Hadn't previous work suggested this?

No. Other than the accompanying manuscript by Chad Pearson (Martinez *et al.*, *Biorxiv* 2024) to our knowledge this work is the first study linking centriolar satellites to RNA or translation, in any experimental model.

Page 12, line 4. It would be best to convert this from a conclusion to a hypothesis that centrosome and cilia proteins are synthesized through independent mechanisms.

We disagree. What our data shows is that cytoplasmic foci of MIB1 and OFD1 persist upon inhibition of protein synthesis for 1 hour. What we can conclude from this is that the population of these proteins on satellites is not newly synthesized. We have no data on where their synthesis occurs.

Page 12, bottom third. "therefore" should be "but also

Assuming the reviewer is referring to the following sentence “The close proximity of *PCNT* mRNA to PCMI remained in cells treated with puromycin and therefore devoid of cytoplasmic PCNT protein, indicating that..”, puromycin treated cells as shown in Fig. 5A lack cytoplasmic PCNT protein signal. ‘Therefore’ to us would appear to be the correct word to describe the consequences of inhibitor treatment on the presence or absence of PCNT protein.

Reviewer #3:

In this paper, Pachinger et al., characterise the composition, behaviour and function of centriole-satellites in Drosophila. These structures have been extensively characterised in a variety of cells, but their function remains unclear and controversial. In part, this is because satellites have not been properly studied in model systems such as worms and flies as it has been difficult to identify clear homologues of PCMI, the key satellite-scaffolding protein. Here the authors confirm that the Drosophila protein Comover (Cmb), which they found in a TurboID screen with the key centriole assembly protein Sas-4/CPAP, appears to be the functional homologue of PCMI, despite only very limited sequence homology. They show that Cmb is required for proper centriole/cilia function in flies (and possibly for centrosome function as well, see below), but Cmb-satellites are not obviously concentrated around centrioles/centrosomes. Most importantly, using a BioID/MS approach they show that Cmb satellites contain several centriole/centrosome/cilia proteins, but also many proteins involved in RNA binding and protein translation. They provide evidence that Cmb/PCMI-satellites function as sites of protein translation for centriole/centrosome/cilia proteins, and propose that this may be the major conserved function of these satellite structures.

The data in this paper is generally well-presented and of high-quality. This new hypothesis about PCMI/Cmb-satellite function is intriguing and will be of considerable interest to the centriole/cilia/centrosome fields, so I am strongly in favour of publication in JCB. I do, however, have a number of suggestions that the authors should consider prior to publication.

Major points:

1. The data showing the potential co-localisation of various centriole/centrosome proteins with the Cmb condensates is crucial to the authors main conclusion, but in some places it was a bit confusing or unclear. In Fig.3I the authors show that some cytoplasmic foci of Sas-4 and Cnn colocalise with one of the many Cmb condensates in the cell. The description of how this experiment was performed (p28, para.2) could be clearer, as I wasn't sure if users defined the region of the cell to be analysed with any prior knowledge of the distribution of the cytoplasmic foci and Cmb condensates, or whether these regions were chosen at random (either by the user or computationally). This is a really important point.

Regions for quantitation were chosen at random from within the cytoplasm away from the nucleus and centrosomes (marked by γ -tubulin in Fig. 3I and S3C, D), without any prior assessment of the coincidence of CMB foci with SAS-4 or other centrosomal proteins. Centrosomes had to be excluded given the localization of CMB to centrosomes in ~20% of

cells (Fig. S3A). Capturing the remaining cytoplasm in its entirety was precluded by the presence of supernumerary centrosomes in >50% of *Drosophila* S2 cells (Kwon *et al.*, *Genes Dev* 2008), which do not cluster together except in mitosis. We now clarify this in the Methods.

2. The authors show similar co-localisation data for several other proteins (Fig.S3C), but don't show the correlation coefficient (shown for Sas-4 and Cnn in Fig.3I), so one has to read the Legend to see which proteins do and don't co-localise with Cmb (and this information does not seem to be provided for CEP97?). The list of proteins that the authors have tested that do and don't colocalise with Cmb condensates should be made clearer. The authors should comment on whether this list make sense with the idea that proteins destined to form complexes together are co-sequestered in Cmb condensates. Ideally, it would be great to see evidence for co-localisation of any such protein pairs in the same Cmb-condensate, but I appreciate this may be technically challenging so would not insist on this.

We do provide a quantitation of the correlation coefficient for all proteins shown in Fig. S3C in the following panel (Fig. S3D), where we use cyan to highlight those proteins that in addition to SAS-4 and CNN shown in the main figures significantly co-localized with CMB: CP110, ANA1 and ANA2. Not significantly associated with CMB were CEP97, SPD-2 and γ -tubulin. We comment on the apparent lack of γ -tubulin localization, since this appears to be a common feature of satellites in flies and vertebrates. In principle, the presence of SAS-4 and CNN on satellites would be consistent with the S-CAP complexes previously identified by Gopalakrishnan *et al.*, *Nat Commun* 2011. SAS-4 has also been reported to interact with CP110, another satellite component (Schmidt *et al.*, *Curr Biol* 2009). Reagent limitations have precluded us from a more comprehensive analysis, so we hesitate to speculate on whether S-CAP complexes and satellites are indeed related. Furthermore, since a lack of clear cytoplasmic foci may be due to reagent limitations, we prefer not to draw strong conclusions regarding the presence or absence of specific components.

As to the reviewer's final point, we agree that examining co-localization of clients belonging to the same macromolecular complex on the same satellite, either at the protein or mRNA level, will be highly informative and we have been investing considerable effort into establishing dual color smFISH to assess this in a comprehensive manner. However, we are not yet in a position to make a definitive statement in that regard.

*3. As the authors note, two previous studies examined the composition of centriole satellites in human cells (Quarantotti *et al.*, 2019; Gheiratmand *et al.*, 2019). These papers focused on centriole, centrosome and cilia proteins, but I wonder if RNA processing/protein translation proteins were also enriched in these previous studies? In theory, the authors give this information in Table S2, but I found this too difficult to wade through. The authors should address this point more directly.*

Yes, they were. **We now include an analysis of the overlap between the satellite proteomes previously defined by Quarantotti *et al.* and Gheiratmand *et al.* and the cytosolic RNA interactome defined by Youn *et al.*, *Mol Cell* 2018 (Fig. S4D), and if anything the overlap is even greater: 55% for the satellite proteome of Gheiratmand and 33% for the PCM1 pulldown of Quarantotti.** This is all the more remarkable since many translation-related proteins were filtered out as background contaminants in the study of Gheiratmand (see list in Gupta *et al.*, *Cell* 2015, Supplemental Table 4). As we state in the text, PCM1 has also been previously identified in numerous studies aimed at identifying RNA-binding proteins (see RBP2GO database, Caudron-Herger *et al.*, *NAR* 2021). The signs were therefore already there from previous work.

4. In Figure 4H the authors should perform the same correlation experiments comparing the real images to randomised (rotated) images that they perform in Figure 3I. The significance of the correlations currently shown are hard to judge as there are so many more Puromycin dots in the 3min Puro image. On a related point, only ~15% of Cmb condensates are Puro+. Does this suggest that ~85% of Cmb condensates are not actively translating anything (in this 3min window), or is this likely to be a problem of sensitivity of detection? Do you get more Cmb+,Puro+ dots if you extend the incubation time to 10mins?

We had included randomized controls in this panel initially, but then removed them to avoid overcrowding the figure. We agree that particularly for the puromycin condition this is an important control to show. **As can be seen in the revised Fig. 4H, the correlation of CMB foci with puromycin label is non-random and significant.** As for the 85% of CMB foci without apparent puromycin signal, we attribute this at least in part to the high level of global cytoplasmic protein synthesis such that any translation at those foci is not detectable above that background. Longer incubations with puromycin exacerbate the problem by further increasing background, with the additional complication of nascent peptides diffusing further from their original site of synthesis.

5. In Figure 2 the authors show that embryos laid by homozygous *Cmb* Δ -/- females exhibit nuclear fall-out defects (Fig.2H) and also centrosome separation defects (Fig.2I), and they speculate that the failure in centrosome separation may explain the mitotic defects observed in these embryos. From the images shown, it seems that the centrosome separation defects might largely occur in regions where the nuclei have already fallen from the cortex? If so, the centrosomes that remain at the cortex in these regions devoid of nuclei usually fail to separate properly (even in WT embryos), as movement around the nuclear periphery seems to be an important driver of centrosome separation. The authors might want to check this and, if correct, I think they should tone down their conclusion that *Cmb* mutants exhibit centrosome defects (as I think this is their only evidence for centrosome defects?). The idea that *PCMI*/*Cmb* primarily contributes to centriole/cilia function, rather than centrosome function, would fit well with most of the authors data, and perhaps is an emerging theme in the centriole satellite field.

Having said that, it is intriguing that the embryos laid by *Cmb* Δ -/- females can survive if they carry a WT copy of *Cmb* provided by the sperm. This also argues against any centrosome defects in the syncytial embryo contributing to embryonic lethality (as the WT gene supplied by the male would be largely silent during the syncytial stages). This embryonic lethality does suggest a non-centriole/cilium function for *Cmb* (as I think embryos are largely devoid of cilia?), but the authors don't have much evidence that this is due to a function at centrosomes (if my point above is correct). It might be worth commenting on this.

We re-examined our movies and the reviewer is correct that the order of events appears to be chromosome missegregation, followed by nuclear fallout and subsequent failure to separate centrosomes. We have revised our discussion of those phenotypes accordingly.

The question remains why CMB is required for proper spindle function. Certainly, only centrosomal but not ciliary defects can explain a cell division phenotype. Yet, centriole duplication and PCM recruitment both appeared to be normal. However, we note that for the latter we have only assessed the scaffolding component CNN, which does not preclude defects in the recruitment of other PCM components, such as the ortholog of PCNT, D-PLP. This protein is of particular interest since its cognate mRNA has been found to be enriched at centrosomes during early embryogenesis and perturbing its expression in mutants of the cytoplasmic polyadenylation element binding (CPEB) protein ORB results in chromosome

missegregation and nuclear fallout, similar to what we see for *Cmb* (Fang and Lerit, Development 2022). We have been unable to assess D-PLP recruitment, but its dependence on PCMI for cytoplasmic expression in vertebrates is certainly intriguing.

It is curious that paternally supplied CMB should be sufficient to sustain embryonic development in embryos laid by homozygous mutant mothers considering that zygotic gene expression only ramps up after the cellularization at nuclear cycle 14 (Farrell and O'Farrell, Annu Rev Genet 2014) and no gross morphological defects were observed in any chromosomal deficiency until that stage (Merrill *et al.*, Development 1998). However, it is known that many zygotic genes are already transcribed at earlier stages. Expression from the paternal copy of *Cmb* may be essential for translation of maternally deposited mRNAs in the absence of maternally contributed CMB, which would be consistent with the above work.

Minor Points:

1. The description in the text of the experiment shown in Fig. 5A is confusing as the authors originally state that there is "a striking loss of cytoplasmic signal for centrosomal and ciliary proteins". I believe that OFD1 is a ciliary protein, so it is confusing to readers that its cytoplasmic levels rise here. This should be reworded.

OFD1 is an outlier in that like MIB1 but unlike other centrosomal and ciliary proteins it has been linked to non-centrosomal/ciliary processes, including translation and protein quality control (Iaconis *et al.*, Sci Rep 2017; Morleo *et al.* EMBOJ 2021). It remains unclear whether these functions are separate from its originally ascribed function in centriole length control (Ferrante *et al.*, Nat Genet 2006) or whether like PCMI OFD1 is only indirectly involved in centriole/ciliary function. However, our statement is certainly true for the majority of centrosomal and ciliary proteins localized to satellites. We have modified the text accordingly.

1. The authors note that a PCMI homologue in flies was identified previously (Kuhn et al., 2014), but then imply that they had to use sophisticated comparisons to identify this homologue in their TurboID hits. I assume this is the same protein, so why didn't they already know that CMB was the fly homologue of PCMI?

We found this protein originally when annotating our SAS-4 BioID proximity interactome. Buried within the Supplemental material of Kuhn *et al.* (in Table S1) the authors identified Comover (then named CG10732) and Y56A3A.7 as the putative *Drosophila* and *C. elegans* orthologs of PCMI. Since this finding was not remarked upon within the main body of the paper, it failed to catch the attention of the centrosome community at the time it was published in 2014, and certainly ours. Kuhn *et al.* used sophisticated tools for extracting information from coiled coil regions of proteins such as PCMI. We instead relied on reciprocal BLASTP best hit analysis using less divergent species as intermediates to confirm the identification of those orthologs and further explore the phylogenetic distribution of PCMI within opisthokonts.

2. The authors should provide more information on the Cmb mutant. After some searching I figured out that it is a partial deletion, but has there been any analysis of whether this is (or behaves as) a null? It is called CmbΔ, implying it is a null, but worth clarifying.

The *Cmb* mutant used in this study was originally described in Fagan *et al.*, PLOS One 2014 and generated via homologous recombination, replacing parts of exons 3 (including the start codon for the major protein isoform) and 4 with the white gene. The authors characterize this mutant as a protein null, with no detectable expression by immunoblotting, a result we confirmed by immunofluorescence using our own polyclonal antibody in a range of tissues including the syncytial embryo and testes (the latter shown in Fig. S3B). We also did not

observe any obvious phenotypic differences when placing the mutant over a deficiency (e.g. Fig. S2B). We therefore consider this mutant to behave as a null.

3. Related to the potential role of PCM1/Cmb in centrosomes, it might be worth checking if Dictyostelium has PCM1, as this organism lacks centrioles and cilia, yet it has several proteins required to make mitotic centrosomes (e.g. homologues of Cnn/CDK5RAP2 and Spd-2/CEP192). If it doesn't, it might further support the argument that PCM1/Cmb is not required for centrosome function.

It would indeed be interesting to see if PCM1 is conserved in a species that lost cilia but retained centrosomes. Dictyostelium is such a species, although the relationship between the acentriolar centrosome of Dictyostelium and the centriole-based centrosome of metazoans remains somewhat unclear. However, we were unable to detect any clear orthologs of PCM1 outside of opisthokonts, including in Amoebozoa to which slime molds belong, which may reflect either a more recent evolutionary origin of the protein or its poor conservation at the primary sequence level. We had initially hoped to use the metazoan phylum Nematomorpha instead on the basis of their reported loss of ciliary genes (Cunha *et al.*, *Curr Biol* 2023). However, here on closer inspection we found ciliary genes to be not entirely lost, while PCM1, Cnn/CDK5RAP2 and SPD-2/CEP192 were undetectable, making it difficult to draw any conclusions. While the evolutionary inheritance pattern is therefore consistent with a role in the formation of centriole-based structures and PCM1 is clearly conserved in the planarian *Schmidtea mediterranea*, which lacks centrosomes but retains centrioles and cilia (Azimzadeh *et al.*, *Nature* 2012), we have no example to assess the reverse.

4. I don't think the authors mention whether any of the proteins that localise in Cmb-foci also form foci that don't colocalise with Cmb? If so, do the authors think these are different translation hubs, allowing the proteins to form different complexes?

No centrosomal/ciliary client protein perfectly co-localizes with CMB, or indeed with PCM1 in vertebrates (see Fig. 7, Gheiratmand *et al.*, *EMBOJ* 2019). However, at least in vertebrates all punctate cytoplasmic signal is translation- (Fig. 5C) and PCM1-dependent (Fig. 6A and S5G). We therefore do not believe that there are other dedicated translation hubs for centrosomal/ciliary proteins besides CMB/PCM1-containing satellites. Instead, we suspect that those foci not colocalizing with CMB/PCM1 in control cells represent fully formed protein particles that have dissociated from their original site of assembly.

5. More help is needed with Fig.S2E: what are we looking at in each section, and what defects are the orange arrows highlighting?

Sections are now labeled and the nature of the ciliary defects indicated by the arrows described in the figure legend.

6. P8, para.2. "or any fly tissue" should be changed to "or in any of several fly tissues that we examined".

This has been changed.

7. The model in Fig.5G was not very helpful. I'm not sure what it was proposing.

What we were trying to highlight in our proposed model of satellite function in the original Fig 5G in contrast to the original transport model in Fig. 1A is the lack of dynamics as a core feature of satellites and the translation of mRNAs encoding centrosomal/ciliary client proteins. We have now added arrows to the original model in Fig. 1A to indicate movement and more

labels to the revised model in Fig. 6E to clarify the nature of satellite clients as nascent protein translated on satellites. We hope the schematics are now more clear.

8. Figure 1C. It looks to me like the otherwise invariant "LR" amino acids are present in the Drosophila Cmb protein-if you just shift the position of the gap in the MSA.

The reviewer may be correct. We did not try to manually refine the multiple sequence alignment. It is worth noting that CMB displays more striking homology to orthologs in other insect species in other regions of the protein. However, this C-terminus appears to be the most conserved amongst PCM1 orthologs overall.

February 28, 2025

RE: JCB Manuscript #202408042R

Alexander Dammermann
University of Vienna

Dear Dr. Dammermann:

Thank you for submitting your revised manuscript entitled "A conserved role for centriolar satellites in translation of centrosomal and ciliary proteins". As you will see, reviewers unanimously support publication of this work. We would be happy to publish your paper in JCB pending final revisions necessary to meet our formatting guidelines (see details below). We also invite you to consider the minor points noted by Reviewers 1 and 3.

Your manuscript was originally co-submitted with a companion article from Chad Pearson's group and was initially reviewed together with that manuscript. At this point we have not received a revised manuscript from Chad Pearson. Please let us know if you would prefer to hold your manuscript and wait for the companion, or if you would prefer to proceed towards publication without it.

A. MANUSCRIPT ORGANIZATION AND FORMATTING:

Full guidelines are available on our Instructions for Authors page, <http://jcb.rupress.org/submission-guidelines#revised>. Submission of a paper that does not conform to JCB guidelines will delay the acceptance of your manuscript.

1) Text limits: Character count for Articles is < 40,000, not including spaces. Count includes abstract, introduction, results, discussion, and acknowledgments. Count does not include title page, figure legends, materials and methods, references, tables, or supplemental legends.

2) Figures limits: Articles may have up to 10 main figures and 5 supplemental figures/tables.

3) Figure formatting: Scale bars must be present on all microscopy images, including inset magnifications. Molecular weight or nucleic acid size markers must be included on all gel electrophoresis. Please avoid pairing red and green for images and graphs to ensure legibility for color-blind readers. If red and green are paired for images, please ensure that the particular red and green hues used in micrographs are distinctive with any of the colorblind types. If not, please modify colors accordingly or provide separate images of the individual channels.

4) Statistical analysis: Error bars on graphic representations of numerical data must be clearly described in the figure legend. The number of independent data points (n) represented in a graph must be indicated in the legend. Statistical methods should be explained in full in the materials and methods. For figures presenting pooled data the statistical measure should be defined in the figure legends. Please also be sure to indicate the statistical tests used in each of your experiments (either in the figure legend itself or in a separate methods section) as well as the parameters of the test (for example, if you ran a t-test, please indicate if it was one- or two-sided, etc.). Also, if you used parametric tests, please indicate if the data distribution was tested for normality (and if so, how). If not, you must state something to the effect that "Data distribution was assumed to be normal but this was not formally tested."

** For all bar plots, or other plots where individual data points are not shown, please describe the error bars in figure legends (SD or SEM).

5) Abstract and title: The abstract should be no longer than 160 words and should communicate the significance of the paper for a general audience. The title should be less than 100 characters including spaces. Make the title concise but accessible to a general readership.

6) Materials and methods: Should be comprehensive and not simply reference a previous publication for details on how an experiment was performed. Please provide full descriptions in the text for readers who may not have access to referenced manuscripts. We also provide a report from SciScore and an associate score, which we encourage you to use as a means of evaluating and improving the methods section.

7) Please be sure to provide the sequences for all of your primers/oligos, plasmids, and RNAi constructs in the materials and methods. You must also indicate in the methods the source, species, and catalog numbers (where appropriate) for all of your antibodies. Please also indicate the acquisition and quantification methods for immunoblotting/western blots.

8) Microscope image acquisition: The following information must be provided about the acquisition and processing of images:

- Make and model of microscope
- Type, magnification, and numerical aperture of the objective lenses
- Temperature
- Imaging medium
- Fluorochromes
- Camera make and model
- Acquisition software
- Any software used for image processing subsequent to data acquisition. Please include details and types of operations involved (e.g., type of deconvolution, 3D reconstitutions, surface or volume rendering, gamma adjustments, etc.).

10) Supplemental materials: There are strict limits on the allowable amount of supplemental data. Articles may have up to 5 supplemental figures. Please also note that tables, like figures, should be provided as individual, editable files. A summary of all supplemental material should appear at the end of the Materials and methods section.

13) ORCID IDs: ORCID IDs are unique identifiers allowing researchers to create a record of their various scholarly contributions in a single place. At resubmission of your final files, please provide an ORCID ID for all authors.

15) A data availability statement is required for all research article submissions. The statement should address all data underlying the research presented in the manuscript. Please visit the JCB instructions for authors for guidelines and examples of statements at (<https://rupress.org/jcb/pages/editorial-policies#data-availability-statement>).

Please note that JCB requires authors to submit Source Data used to generate figures containing gels and Western blots with all revised manuscripts. This Source Data consists of fully uncropped and unprocessed images for each gel/blot displayed in the main and supplemental figures. Since your paper includes cropped gel and/or blot images, please be sure to provide one Source Data file for each figure that contains gels and/or blots along with your revised manuscript files. File names for Source Data figures should be alphanumeric without any spaces or special characters (i.e., SourceDataF#, where F# refers to the associated main figure number or SourceDataFS# for those associated with Supplementary figures). The lanes of the gels/blots should be labeled as they are in the associated figure, the place where cropping was applied should be marked (with a box), and molecular weight/size standards should be labeled wherever possible. Source Data files will be directly linked to specific figures in the published article.

WHEN APPROPRIATE: The source code for all custom computational methods published in JCB must be made freely available as supplemental material hosted at www.jcb.org. Please contact the JCB Editorial Office to find out how to submit your custom macros, code for custom algorithms, etc. Generally, these are provided as raw code in a .txt file or as other file types in a .zip file. Please also include a one-sentence summary of each file in the Online Supplemental Material paragraph of your manuscript.

Journal of Cell Biology now requires a data availability statement for all research article submissions. These statements will be published in the article directly above the Acknowledgments. The statement should address all data underlying the research presented in the manuscript. Please visit the JCB instructions for authors for guidelines and examples of statements at (<https://rupress.org/jcb/pages/editorial-policies#data-availability-statement>).

B. FINAL FILES:

Please upload the following materials to our online submission system. These items are required prior to acceptance. If you

have any questions, contact JCB's Managing Editor, Lindsey Hollander (lhollander@rockefeller.edu).

Thank you for your attention to these final processing requirements. Please revise and format the manuscript and upload materials within 7 days. If you need an extension for whatever reason, please let us know and we can work with you to determine a suitable revision period.

Thank you for this interesting contribution, we look forward to publishing your paper in Journal of Cell Biology.

Sincerely,

Monica Bettencourt-Dias
Monitoring Editor
Journal of Cell Biology

Tim Fessenden
Scientific Editor
Journal of Cell Biology

Reviewer #1 (Comments to the Authors (Required)):

In this revised manuscript, Pachinger et al. provided more mechanistic examination of the contributions of centriole satellites to local translation. In my prior review, I challenged the authors to more convincingly demonstrate a role for centriole satellites, and CMB in particular, in local protein synthesis. For the most part, the authors addressed all three of the major concerns raised in the initial review. To address the functional homology of Cmb, the authors now show Cmb localizes to vertebrate satellites. While this assay is not a functional rescue experiment, the fact that Cmb can localize to orthologous vertebrate structures is interesting and exciting. I am also very pleased to see the authors included the Puro-PLA experiment showing PCNT is translated in very close proximity to PCM1. Finally, the authors provide some much-needed evidence that local translation occurs near Cmb puncta through their new puromycylation data. I appreciated the authors included in Fig 4H the necessary controls and quantification to show the signals are convincing and the data rigorous. They also include a supplemental image showing some CMB puncta overlap with SAS-4 mRNA via smFISH.

In addition to these critical points, the authors also seem to have addressed most of my minor points, and I commend the authors on their efforts. The revision also improved the clarity and cohesion of the manuscript. I support publication of this work in JCB, and I think it is an exciting advance for the field.

minor points

1. The authors include in 5B the randomized measurements as control for specificity of the Puro-PLA (PCNT) experiment, which is helpful. For future PLA experiments, I would encourage the authors to also include the similar CHX control, as they show for their earlier puromycylation expt (Fig 4H)
2. The LUTs for two of the single channel insets presented in the supplement look off. These are Fig S3C (CMB channel from the Ana2 expt) and Fig S5C (PCNT inset). For those insets, the single channel has significantly less background fluorescence than the uncropped image.
3. On p. 10, the authors report the identification of centrosome and cilia proteins from their proteomics of CMB interactors was "expected" (4th line from bottom). Is this because they first identified CMB from their SAS-4 datasets? I'm not sure I agree it was an expected result, and I also think it undersells the advance of the argument they are building, which is that CMB seems to be important for ciliary functions (NGA, male sterility, axonome defects).
4. On p. 14, 4th line down: would the authors please edit "that enrichment" to "PCNT mRNA enrichment" or "localization," as the term "that" is so vague that the impact of this important finding becomes muddled.

Reviewer #2 (Comments to the Authors (Required)):

I have now read the updated version of the manuscript by Pachinger, and their reviewer response letter. Overall paper is much improved from the standpoint of readability and clarity, and also as it relates to supporting the main claim of the manuscript of translation at centriole satellites. The authors have added a new key experiment using an accepted technique in the field (Puro-PLA) that indicated sites of translation. The authors quantification of this experiment supports their model that centriole satellites actively translate protein.

My recommendation to JCB is to consider this manuscript for publication without delay.

Reviewer #3 (Comments to the Authors (Required)):

In their revised manuscript, Pachinger et al., generally do a good job of answering my main concerns, and some nice additional experiments have been included that strengthen some of their conclusions. I therefore remain strongly in support of publication in JCB.

On a very minor note, I still don't understand why the authors persist in claiming that Cmb mutants have centrosome defects, as I don't see any convincing evidence for this here, and the cmb phenotype (as well as the analysis presented in Figure 1B) suggest that Cmb is involved in centriole/cilia function rather than centrosome function. I also think this fits better with most of the recent data on satellite function. I am very happy for the authors to keep their interpretation, and there is no need to address this point in any response; I'm just slightly surprised by it.

RESPONSE TO REVIEWERS

We are pleased to hear we successfully addressed the reviewers' major concerns in the previous round of revision. We believe our manuscript to be substantially improved by those changes and we would like to thank the editor and reviewers for their time. Below our response to the reviewers' remaining minor points, with the original comments reproduced in full in *italics*.

Reviewer #1:

In this revised manuscript, Pachinger et al. provided more mechanistic examination of the contributions of centriole satellites to local translation. In my prior review, I challenged the authors to more convincingly demonstrate a role for centriole satellites, and CMB in particular, in local protein synthesis. For the most part, the authors addressed all three of the major concerns raised in the initial review. To address the functional homology of Cmb, the authors now show Cmb localizes to vertebrate satellites. While this assay is not a functional rescue experiment, the fact that Cmb can localize to orthologous vertebrate structures is interesting and exciting. I am also very pleased to see the authors included the Puro-PLA experiment showing PCNT is translated in very close proximity to PCMI. Finally, the authors provide some much-needed evidence that local translation occurs near Cmb puncta through their new puromycylation data. I appreciated the authors included in Fig 4H the necessary controls and quantification to show the signals are convincing and the data rigorous. They also include a supplemental image showing some CMB puncta overlap with SAS-4 mRNA via smFISH.

In addition to these critical points, the authors also seem to have addressed most of my minor points, and I commend the authors on their efforts. The revision also improved the clarity and cohesion of the manuscript. I support publication of this work in JCB, and I think it is an exciting advance for the field.

Minor points

1. The authors include in 5B the randomized measurements as control for specificity of the Puro-PLA (PCNT) experiment, which is helpful. For future PLA experiments, I would encourage the authors to also include the similar CHX control, as they show for their earlier puromycylation expt (Fig 4H).

Thank you for the suggestion. We will bear this in mind.

2. The LUTs for two of the single channel insets presented in the supplement look off. These are Fig S3C (CMB channel from the Ana2 expt) and Fig S5C (PCNT inset). For those insets, the single channel has significantly less background fluorescence than the uncropped image.

Fig. S3C has been corrected (insets had been scaled independently of the main image). Fig. S5C does not have this issue.

3. On p. 10, the authors report the identification of centrosome and cilia proteins from their proteomics of CMB interactors was "expected" (4th line from bottom). Is this because they first identified CMB from their SAS-4 datasets? I'm not sure I agree it was an expected result, and I also think it undersells the advance of the argument they are building, which is that CMB seems to be important for ciliary functions (NGA, male sterility, axonome defects).

To us the identification of centrosomal and ciliary proximity interactors for CMB was indeed to be expected, given that we are dealing with the *Drosophila* ortholog of PCMI, for which

such interactors have been extensively characterized in previous studies (Gupta *et al.*, Cell 2015; Gheiratmand *et al.*, EMBOJ 2019; Quarantotti *et al.*, EMBOJ 2019). This class of interactors also did not help us in getting to the molecular mechanism underlying centriolar satellite function (ie translation). Indeed, we suspect that many of these interactions are indirect, with CMB associating with the cognate mRNA not the protein itself. We now clarify this in the text.

4. On p. 14, 4th line down: would the authors please edit "that enrichment" to "PCNT mRNA enrichment" or "localization," as the term "that" is so vague that the impact of this important finding becomes muddled.

This has been changed.

Reviewer #2:

I have now read the updated version of the manuscript by Pachinger, and their reviewer response letter. Overall paper is much improved from the standpoint of readability and clarity, and also as it relates to supporting the main claim of the manuscript of translation at centriole satellites. The authors have added a new key experiment using an accepted technique in the field (Puro-PLA) that indicated sites of translation. The authors quantification of this experiment supports their model that centriole satellites actively translate protein.

My recommendation to JCB is to consider this manuscript for publication without delay.

Reviewer #3:

*In their revised manuscript, Pachinger *et al.*, generally do a good job of answering my main concerns, and some nice additional experiments have been included that strengthen some of their conclusions. I therefore remain strongly in support of publication in JCB.*

On a very minor note, I still don't understand why the authors persist in claiming that Cmb mutants have centrosome defects, as I don't see any convincing evidence for this here, and the cmb phenotype (as well as the analysis presented in Figure 1B) suggest that Cmb is involved in centriole/cilia function rather than centrosome function. I also think this fits better with most of the recent data on satellite function. I am very happy for the authors to keep their interpretation, and there is no need to address this point in any response; I'm just slightly surprised by it.

As we wrote in the previous rebuttal (Reviewer #3, major point 5), we agree that have not observed any obvious defects in PCM scaffold recruitment in *Cmb* mutants. However, the chromosome segregation and embryonic lethality phenotype in those mutants cannot be explained by centriolar or ciliary defects, but would be in line with what would be expected for misregulation of a PCM client protein. We note that PCM proteins are also represented amongst the CMB proximity interactome, in addition to centriolar and ciliary proteins. That satellite perturbations in flies would also affect centrosomes is therefore not too far-fetched.